# Asynchronous Distributed Bilevel Optimization

**Yang Jiao**
Tongji University

**Kai Yang**[*]
Tongji University

**Tiancheng Wu**
Tongji University

**Dongjin Song**
University of Connecticut

**Chengtao Jian**
Tongji University

## Abstract

Bilevel optimization plays an essential role in many machine learning tasks, ranging from hyperparameter optimization to meta-learning. Existing studies on bilevel optimization, however, focus on either centralized or synchronous distributed setting. The centralized bilevel optimization approaches require collecting a massive amount of data to a single server, which inevitably incur significant communication expenses and may give rise to data privacy risks. Synchronous distributed bilevel optimization algorithms, on the other hand, often face the straggler problem and will immediately stop working if a few workers fail to respond. As a remedy, we propose **A**synchronous **D**istributed **B**ilevel **O**ptimization (ADBO) algorithm. The proposed ADBO can tackle bilevel optimization problems with both nonconvex upper-level and lower-level objective functions, and its convergence is theoretically guaranteed. Furthermore, it is revealed through theoretical analysis that the iteration complexity of ADBO to obtain the $\epsilon$-stationary point is upper bounded by $\mathcal{O}(\frac{1}{\epsilon^2})$. Thorough empirical studies on public datasets have been conducted to elucidate the effectiveness and efficiency of the proposed ADBO.

## 1 Introduction

Recently, bilevel optimization has emerged due to its popularity in various machine learning applications, *e.g.*, hyperparameter optimization (Khanduri et al., 2021; Liu et al., 2021a), meta-learning (Likhosherstov et al., 2021; Ji et al., 2020), reinforcement learning (Hong et al., 2020; Zhou & Liu, 2022), and neural architecture search (Jiang et al., 2020; Jiao et al., 2022b). In bilevel optimization, one optimization problem is embedded or nested with another. Specifically, the outer optimization problem is called the upper-level optimization problem and the inner optimization problem is called the lower-level optimization problem. A general form of the bilevel optimization problem can be written as,

$$
\begin{aligned}
\min \quad & F(\boldsymbol{x}, \boldsymbol{y}) \\
\text{s.t.} \quad & \boldsymbol{y} = \arg\min_{\boldsymbol{y}'} f(\boldsymbol{x}, \boldsymbol{y}') \\
\text{var.} \quad & \boldsymbol{x}, \boldsymbol{y},
\end{aligned}
\tag{1}
$$

where $F$ and $f$ denote the upper-level and lower-level objective functions, respectively. $\boldsymbol{x} \in \mathbb{R}^n$ and $\boldsymbol{y} \in \mathbb{R}^m$ are variables. Bilevel optimization can be treated as a special case of constrained optimization since the lower-level optimization problem can be viewed as a constraint to the upper-level optimization problem (Sinha et al., 2017).

The proliferation of smartphones and Internet of Things (IoT) devices has generated a plethora of data in various real-world applications. Centralized bilevel optimization approaches require collecting a massive amount of data from distributed edge devices and passing them to a centralized server for model training. These methods, however, may give rise to data privacy risks (Subramanya & Riggio, 2021) and encounter communication bottlenecks (Subramanya & Riggio, 2021). To tackle these challenges, recently, distributed algorithms have been developed to solve the decentralized

---

[*]Corresponding author.

bilevel optimization problems (Yang et al., 2022; Chen et al., 2022b; Lu et al., 2022). Tarzanagh et al. (2022) and Li et al. (2022) study the bilevel optimization problems under a federated setting. Specifically, the distributed bilevel optimization problem can be given by

$$
\begin{aligned}
\min\quad & F(\boldsymbol{x}, \boldsymbol{y}) = \sum_{i=1}^{N} G_i(\boldsymbol{x}, \boldsymbol{y}) \\
\text{s.t.}\quad & \boldsymbol{y} = \arg\min_{\boldsymbol{y}'} f(\boldsymbol{x}, \boldsymbol{y}') = \sum_{i=1}^{N} g_i(\boldsymbol{x}, \boldsymbol{y}') \\
\text{var.}\quad & \boldsymbol{x}, \boldsymbol{y},
\end{aligned}
\tag{2}
$$

where $N$ is the number of workers (devices), $G_i$ and $g_i$ denote the local upper-level and lower-level objective functions, respectively. Although existing approaches have shown their success in resolving distributed bilevel optimization problems, they only focus on the synchronous distributed setting. Synchronous distributed methods may encounter the straggler problem (Jiang et al., 2021) and its speed is limited by the worker with maximum delay (Chang et al., 2016). Moreover, synchronous distributed method will immediately stop working if a few workers fail to respond (Zhang & Kwok, 2014) (which is common in large-scale distributed systems). The aforementioned issues give rise to the following question:

*Can we design an asynchronous distributed algorithm for bilevel optimization?*

To this end, we develop an **A**synchronous **D**istributed **B**ilevel **O**ptimization (ADBO) algorithm which is a single-loop algorithm and computationally efficient. The proposed ADBO regards the lower-level optimization problem as a constraint to the upper-level optimization problem, and utilizes cutting planes to approximate this constraint. Then, the approximate problem is solved in an ***asynchronous distributed manner*** by the proposed ADBO. We prove that even if both the upper-level and lower-level objectives are *nonconvex*, the proposed ADBO is guaranteed to converge. The iteration complexity of ADBO is also theoretically derived. To facilitate the comparison, we not only present a centralized bilevel optimization algorithm in Appendix A, but also compare the convergence results of ADBO to state-of-the-art bilevel optimization algorithms with both centralized and distributed settings in Table 1.

**Contributions.** Our contributions can be summarized as:

1. We propose a novel algorithm, ADBO, to solve the bilevel optimization problem in an ***asynchronous distributed manner***. ADBO is a single-loop algorithm and is computationally efficient. To the best of our knowledge, it is the first work in tackling asynchronous distributed bilevel optimization problem.

2. We demonstrate that the proposed ADBO can be applied to bilevel optimization with *nonconvex* upper-level and lower-level objectives *with constraints*. We also theoretically derive that the iteration complexity for the proposed ADBO to obtain the $\epsilon$-stationary point is upper bounded by $\mathcal{O}(\frac{1}{\epsilon^2})$.

3. Our thorough empirical studies justify the superiority of the proposed ADBO over the existing state-of-the-art methods.

## 2    RELATED WORK

**Bilevel optimization:** The bilevel optimization problem was firstly introduced by Bracken & McGill (1973). In recent years, many approaches have been developed to solve this problem and they can be divided into three categories (Gould et al., 2016). The first type of approaches assume there is an analytical solution to the lower-level optimization problem (*i.e.*, $\phi(\boldsymbol{x}) = \arg\min_{\boldsymbol{y}'} f(\boldsymbol{x}, \boldsymbol{y}')$) (Zhang et al., 2021). In this case, the bilevel optimization problem can be simplified to a single-level optimization problem (*i.e.*, $\min_{\boldsymbol{x}} F(\boldsymbol{x}, \phi(\boldsymbol{x}))$). Nevertheless, finding the analytical solution for the lower-level optimization problem is often very difficult, if not impossible. The second type of approaches replace the lower-level optimization problem with the sufficient conditions for optimality (*e.g.*, KKT conditions) (Biswas & Hoyle, 2019; Sinha et al., 2017). Then, the bilevel program can be reformulated as a single-level constrained optimization problem. However, the resulting problem could be hard to solve since it often involves a large number of constraints (Ji et al., 2021; Gould et al., 2016). The third type of approaches are gradient-based methods (Ghadimi & Wang, 2018; Hong et al., 2020; Liao et al., 2018) that compute the hypergradient (or the estimation of hypergradient), *i.e.*, $\frac{\partial F(\boldsymbol{x},\boldsymbol{y})}{\partial \boldsymbol{x}} + \frac{\partial F(\boldsymbol{x},\boldsymbol{y})}{\partial \boldsymbol{y}} \frac{\partial \boldsymbol{y}}{\partial \boldsymbol{x}}$, and use gradient descent to solve the bilevel optimization

problems. Most of the existing bilevel optimziation methods focus on centralized settings and require collecting a massive amount of data from distributed edge devices (workers). This may give rise to data privacy risks (Subramanya & Riggio, 2021) and encounter communication bottlenecks (Subramanya & Riggio, 2021).

**Asynchronous distributed optimization:** To alleviate the aforementioned issues in the centralized setting, various distributed optimization methods can be employed. Distributed optimization methods can be generally divided into synchronous distributed methods and asynchronous distributed methods (Assran et al., 2020). For synchronous distributed methods (Boyd et al., 2011), the master needs to wait for the updates from all workers before it proceeds to the next iteration. Therefore, it may suffer from the straggler problem and the speed is limited by the worker with maximum delay (Chang et al., 2016). There are several advanced techniques have been proposed to make the synchronous algorithm more efficient, such as large batch size, warmup and so on (Goyal et al., 2017; You et al., 2019; Huo et al., 2021; Liu & Mozafari, 2022; Wang et al., 2020). For asynchronous distributed methods (Chen et al., 2020; Matamoros, 2017), the master can update its variables once it receives updates from $S$ workers, *i.e.*, active workers ($1 \le S \le N$, where $N$ is the number of all workers). The asynchronous distributed algorithm is strongly preferred for large scale distributed systems in practice since it does not suffer from the straggler problem (Jiang et al., 2021). Asynchronous distributed methods (Wu et al., 2017; Liu et al., 2017) have been employed for many real-world applications, such as Google's DistBelief system (Dean et al., 2012), the training of 10 million YouTube videos (Le, 2013), federated learning for edge computing (Lu et al., 2019; Liu et al., 2021c). Since the action orders of each worker are different in the asynchronous distributed setting, which will result in complex interaction dynamics (Jiang et al., 2021), the theoretical analysis for asynchronous distributed algorithms is usually more challenging than that of the synchronous distributed algorithms. In summary, the synchronous and asynchronous algorithm have different application scenarios. When the delay of each worker is not much different, the synchronous algorithm suits better. While there are stragglers in the distributed system, the asynchronous algorithm is more preferred. So far, existing works for distributed bilevel optimization only focus on the synchronous setting (Tarzanagh et al., 2022; Li et al., 2022; Chen et al., 2022b), how to design an asynchronous algorithm for distributed bilevel optimization remains under-explored. To the best of our knowledge, this is the first work that designs an asynchronous algorithm for distributed bilevel optimization.

## 3 ASYNCHRONOUS DISTRIBUTED BILEVEL OPTIMIZATION

In this section, we propose **A**synchronous **D**istributed **B**ilevel **O**ptimization (ADBO) to solve the distributed bilevel optimization problem in an asynchronous manner. First, we reformulate problem in Eq. (2) as a consensus problem (Matamoros, 2017; Chang et al., 2016),

$$
\begin{aligned}
\min \quad & F(\{\boldsymbol{x}_i\}, \{\boldsymbol{y}_i\}, \boldsymbol{v}, \boldsymbol{z}) = \sum_{i=1}^{N} G_i(\boldsymbol{x}_i, \boldsymbol{y}_i) \\
\text{s.t.} \quad & \boldsymbol{x}_i = \boldsymbol{v}, i = 1, \cdots, N \\
& \{\boldsymbol{y}_i\}, \boldsymbol{z} = \argmin_{\{\boldsymbol{y}_i'\}, \boldsymbol{z}'} f(\boldsymbol{v}, \{\boldsymbol{y}_i'\}, \boldsymbol{z}') = \sum_{i=1}^{N} g_i(\boldsymbol{v}, \boldsymbol{y}_i') \\
& \qquad\qquad \boldsymbol{y}_i' = \boldsymbol{z}', i = 1, \cdots, N \\
\text{var.} \quad & \{\boldsymbol{x}_i\}, \{\boldsymbol{y}_i\}, \boldsymbol{v}, \boldsymbol{z},
\end{aligned}
\tag{3}
$$

where $\boldsymbol{x}_i \in \mathbb{R}^n$ and $\boldsymbol{y}_i \in \mathbb{R}^m$ are local variables in $i^{\text{th}}$ worker, $\boldsymbol{v} \in \mathbb{R}^n$ and $\boldsymbol{z} \in \mathbb{R}^m$ are the consensus variables in the master node. The reformulation given in Eq. (3) is a consensus problem which allows to develop distributed training algorithms for bilevel optimization based on the parameter server architecture (Assran et al., 2020). As shown in Figure 13, in parameter server architecture, the communication is centralized around the master, and workers pull the consensus variables $\boldsymbol{v}, \boldsymbol{z}$ from and send their local variables $\boldsymbol{x}_i, \boldsymbol{y}_i$ to the master. Parameter server training is a well-known data-parallel approach for scaling up machine learning model training on a multitude of machines (Verbraeken et al., 2020). Most of the existing bilevel optimization works in machine learning only consider the bilevel programs without upper-level and lower-level constraints (Franceschi et al., 2018; Yang et al., 2021; Chen et al., 2022a) or bilevel programs with only upper-level (or lower-level) constraint (Zhang et al., 2022; Mehra & Hamm, 2021). On the contrary, we focus on the bilevel programs (*i.e.*, Eq. (3)) with *both* lower-level and upper-level constraints, which is more challenging. By defining $\phi(\boldsymbol{v}) = \argmin_{\{\boldsymbol{y}_i'\}, \boldsymbol{z}'} \{ \sum_{i=1}^{N} g_i(\boldsymbol{v}, \boldsymbol{y}_i') : \boldsymbol{y}_i' = \boldsymbol{z}', i = 1, \cdots, N \}$

and $h(\boldsymbol{v}, \{\boldsymbol{y}_i\}, \boldsymbol{z}) = \left\| \begin{bmatrix} \{\boldsymbol{y}_i\} \\ \boldsymbol{z} \end{bmatrix} - \phi(\boldsymbol{v}) \right\|^2$, we can reformulate problem in Eq. (3) as:

$$
\begin{aligned}
\min \quad & F(\{\boldsymbol{x}_i\}, \{\boldsymbol{y}_i\}, \boldsymbol{v}, \boldsymbol{z}) = \sum_{i=1}^{N} G_i(\boldsymbol{x}_i, \boldsymbol{y}_i) \\
\text{s.t.} \quad & \boldsymbol{x}_i = \boldsymbol{v}, i = 1, \cdots, N \\
& h(\boldsymbol{v}, \{\boldsymbol{y}_i\}, \boldsymbol{z}) = 0 \\
\text{var.} \quad & \{\boldsymbol{x}_i\}, \{\boldsymbol{y}_i\}, \boldsymbol{v}, \boldsymbol{z}.
\end{aligned}
\tag{4}
$$

To better clarify how ADBO works, we sketch the procedure of ADBO. Firstly, ADBO computes the estimate of the solution to lower-level optimization problem. Then, inspired by cutting plane method, a set of cutting planes is utilized to approximate the feasible region of the upper-level bilevel optimization problem. Finally, the asynchronous algorithm for solving the resulting problem and how to update cutting planes are proposed. The remaining contents are divided into four parts, *i.e.*, estimate of solution to lower-level optimization problem, polyhedral approximation, asynchronous algorithm, updating cutting planes.

### 3.1 Estimate of Solution to Lower-level Optimization Problem

A consensus problem, *i.e.*, the lower-level optimization problem in Eq. (3), needs to be solved in a distributed manner if an exact $\phi(\boldsymbol{v})$ is desired. Following existing works (Li et al., 2022; Gould et al., 2016; Yang et al., 2021) for bilevel optimization, instead of pursuing the exact $\phi(\boldsymbol{v})$, an estimate of $\phi(\boldsymbol{v})$ could be utilized. For this purpose, we first obtain the first-order Taylor approximation of $g_i(\boldsymbol{v}, \{\boldsymbol{y}_i'\})$ with respect to $\boldsymbol{v}$, *i.e.*, for a given point $\overline{\boldsymbol{v}}$, $\widetilde{g}_i(\boldsymbol{v}, \{\boldsymbol{y}_i'\}) = g_i(\overline{\boldsymbol{v}}, \{\boldsymbol{y}_i'\}) + \nabla_{\boldsymbol{v}} g_i(\overline{\boldsymbol{v}}, \{\boldsymbol{y}_i'\})^\top (\boldsymbol{v} - \overline{\boldsymbol{v}})$. Then, similar to many works that use $K$ steps of gradient descent (GD) to approximate the optimal solution of lower-level optimization problem (Ji et al., 2021; Yang et al., 2021; Liu et al., 2021b), we utilize the results after $K$ communication rounds between workers and master to approximate $\phi(\boldsymbol{v})$. Specifically, given $\widetilde{g}_i(\boldsymbol{v}, \{\boldsymbol{y}_i'\})$, the augmented Lagrangian function of the lower-level optimization problem in Eq. (3) can be expressed as,

$$
g_p(\boldsymbol{v}, \{\boldsymbol{y}_i'\}, \boldsymbol{z}', \{\boldsymbol{\varphi}_i\}) = \sum_{i=1}^{N} \left( \widetilde{g}_i(\boldsymbol{v}, \boldsymbol{y}_i') + \boldsymbol{\varphi}_i^\top (\boldsymbol{y}_i' - \boldsymbol{z}') + \frac{\mu}{2} \|\boldsymbol{y}_i' - \boldsymbol{z}'\|^2 \right),
\tag{5}
$$

where $\boldsymbol{\varphi}_i \in \mathbb{R}^m$ is the dual variable, and $\mu > 0$ is a penalty parameter. In $(k+1)^{\text{th}}$ iteration, we have,

(1) Workers update their local variables as follows,

$$
\boldsymbol{y}_{i,k+1}' = \boldsymbol{y}_{i,k}' - \eta_{\boldsymbol{y}} \nabla_{\boldsymbol{y}_i} g_p(\boldsymbol{v}, \{\boldsymbol{y}_{i,k}'\}, \boldsymbol{z}_k', \{\boldsymbol{\varphi}_{i,k}\}),
\tag{6}
$$

where $\eta_{\boldsymbol{y}}$ is the step-size. Then, workers transmit the local variables $\boldsymbol{y}_{i,k+1}'$ to the master.

(2) After receiving updates from workers, the master updates variables as follows,

$$
\boldsymbol{z}_{k+1}' = \boldsymbol{z}_k' - \eta_{\boldsymbol{z}} \nabla_{\boldsymbol{z}} g_p(\boldsymbol{v}, \{\boldsymbol{y}_{i,k}'\}, \boldsymbol{z}_k', \{\boldsymbol{\varphi}_{i,k}\}),
\tag{7}
$$

$$
\boldsymbol{\varphi}_{i,k+1} = \boldsymbol{\varphi}_{i,k} + \eta_{\boldsymbol{\varphi}} \nabla_{\boldsymbol{\varphi}_i} g_p(\boldsymbol{v}, \{\boldsymbol{y}_{i,k+1}'\}, \boldsymbol{z}_{k+1}', \{\boldsymbol{\varphi}_{i,k}\}),
\tag{8}
$$

where $\eta_{\boldsymbol{z}}$ and $\eta_{\boldsymbol{\varphi}}$ are step-sizes. Next, the master broadcasts $\boldsymbol{z}_{k+1}'$ and $\boldsymbol{\varphi}_{i,k+1}$ to workers.

As mentioned above, we utilize the results after $K$ communication rounds to approximate $\phi(\boldsymbol{v})$, *i.e.*,

$$
\phi(\boldsymbol{v}) = \begin{bmatrix} \{\boldsymbol{y}_{i,0}' - \sum_{k=0}^{K-1} \eta_{\boldsymbol{y}} \nabla_{\boldsymbol{y}_i} g_p(\boldsymbol{v}, \{\boldsymbol{y}_{i,k}'\}, \boldsymbol{z}_k', \{\boldsymbol{\varphi}_{i,k}\})\} \\ \boldsymbol{z}_0' - \sum_{k=0}^{K-1} \eta_{\boldsymbol{z}} \nabla_{\boldsymbol{z}} g_p(\boldsymbol{v}, \{\boldsymbol{y}_{i,k}'\}, \boldsymbol{z}_k', \{\boldsymbol{\varphi}_{i,k}\}) \end{bmatrix}.
\tag{9}
$$

### 3.2 Polyhedral Approximation

Considering $\phi(\boldsymbol{v})$ in Eq. (9), the relaxed problem with respect to the problem in Eq. (4) is,

$$
\begin{aligned}
\min \quad & F(\{\boldsymbol{x}_i\}, \{\boldsymbol{y}_i\}, \boldsymbol{v}, \boldsymbol{z}) = \sum_{i=1}^{N} G_i(\boldsymbol{x}_i, \boldsymbol{y}_i) \\
\text{s.t.} \quad & \boldsymbol{x}_i = \boldsymbol{v}, i = 1, \cdots, N \\
& h(\boldsymbol{v}, \{\boldsymbol{y}_i\}, \boldsymbol{z}) \leq \varepsilon \\
\text{var.} \quad & \{\boldsymbol{x}_i\}, \{\boldsymbol{y}_i\}, \boldsymbol{v}, \boldsymbol{z},
\end{aligned}
\tag{10}
$$

where $\varepsilon > 0$ is a constant. Assuming that $h(\boldsymbol{v}, \{\boldsymbol{y}_i\}, \boldsymbol{z})$ is a convex function with respect to $(\boldsymbol{v}, \{\boldsymbol{y}_i\}, \boldsymbol{z})$, which is always satisfied when we set $K = 1$ in Eq. (10) according to the operations that preserve convexity (Boyd et al., 2004). Since the sublevel set of a convex function is convex (Boyd et al., 2004), the feasible set with respect to constraint $h(\boldsymbol{v}, \{\boldsymbol{y}_i\}, \boldsymbol{z}) \leq \varepsilon$ is a convex set. In this paper, inspired by the cutting plane method (Boyd & Vandenberghe, 2007; Michalka, 2013; Franc et al., 2011; Yang et al., 2014), a set of cutting planes is utilized to approximate the feasible region with respect to constraint $h(\boldsymbol{v}, \{\boldsymbol{y}_i\}, \boldsymbol{z}) \leq \varepsilon$ in Eq. (10). The set of cutting planes forms a polytope, let $\boldsymbol{\mathcal{P}}^t$ denote the polytope in $(t+1)^{\text{th}}$ iteration, which can be expressed as,

$$\boldsymbol{\mathcal{P}}^t = \{\boldsymbol{a}_l^\top \boldsymbol{v} + \sum_{i=1}^N \boldsymbol{b}_{i,l}^\top \boldsymbol{y}_i + \boldsymbol{c}_l^\top \boldsymbol{z} + \kappa_l \leq 0, \, l = 1, \cdots, |\boldsymbol{\mathcal{P}}^t|\}, \tag{11}$$

where $\boldsymbol{a}_l \in \mathbb{R}^n$, $\boldsymbol{b}_{i,l} \in \mathbb{R}^m$, $\boldsymbol{c}_l \in \mathbb{R}^m$ and $\kappa_l \in \mathbb{R}^1$ are the parameters in $l^{\text{th}}$ cutting plane, and $|\boldsymbol{\mathcal{P}}^t|$ denotes the number of cutting planes in $\boldsymbol{\mathcal{P}}^t$. Thus, the approximate problem in $(t+1)^{\text{th}}$ iteration can be expressed as follows,

$$
\begin{aligned}
\min \quad & F(\{\boldsymbol{x}_i\}, \{\boldsymbol{y}_i\}, \boldsymbol{v}, \boldsymbol{z}) = \sum_{i=1}^N G_i(\boldsymbol{x}_i, \boldsymbol{y}_i) \\
\text{s.t.} \quad & \boldsymbol{x}_i = \boldsymbol{v}, i = 1, \cdots, N \\
& \boldsymbol{a}_l^\top \boldsymbol{v} + \sum_{i=1}^N \boldsymbol{b}_{i,l}^\top \boldsymbol{y}_i + \boldsymbol{c}_l^\top \boldsymbol{z} + \kappa_l \leq 0, l = 1, \cdots, |\boldsymbol{\mathcal{P}}^t| \\
\text{var.} \quad & \{\boldsymbol{x}_i\}, \{\boldsymbol{y}_i\}, \boldsymbol{v}, \boldsymbol{z},
\end{aligned}
\tag{12}
$$

The cutting planes will be updated to refine the approximation, details are given in Section 3.4.

### 3.3 Asynchronous Algorithm

In the proposed ADBO, we solve the distributed bilevel optimization problem in an asynchronous manner. The Lagrangian function of Eq. (12) can be written as:

$$L_p = \sum_{i=1}^N G_i(\boldsymbol{x}_i, \boldsymbol{y}_i) + \sum_{l=1}^{|\boldsymbol{\mathcal{P}}^t|} \lambda_l \left(\boldsymbol{a}_l^\top \boldsymbol{v} + \sum_{i=1}^N \boldsymbol{b}_{i,l}^\top \boldsymbol{y}_i + \boldsymbol{c}_l^\top \boldsymbol{z} + \kappa_l\right) + \sum_{i=1}^N \boldsymbol{\theta}_i^\top (\boldsymbol{x}_i - \boldsymbol{v}), \tag{13}$$

where $\lambda_l \in \mathbb{R}^1$, $\boldsymbol{\theta}_i \in \mathbb{R}^n$ are dual variables, $L_p$ is simplified form of $L_p(\{\boldsymbol{x}_i\}, \{\boldsymbol{y}_i\}, \boldsymbol{v}, \boldsymbol{z}, \{\lambda_l\}, \{\boldsymbol{\theta}_i\})$. The regularized version (Xu et al., 2020) of Eq. (13) is employed to update all variables as follows,

$$\widetilde{L}_p(\{\boldsymbol{x}_i\}, \{\boldsymbol{y}_i\}, \boldsymbol{v}, \boldsymbol{z}, \{\lambda_l\}, \{\boldsymbol{\theta}_i\}) = L_p - \sum_{l=1}^{|\boldsymbol{\mathcal{P}}^t|} \frac{c_1^t}{2} ||\lambda_l||^2 - \sum_{i=1}^N \frac{c_2^t}{2} ||\boldsymbol{\theta}_i||^2, \tag{14}$$

where $c_1^t$ and $c_2^t$ denote the regularization terms in $(t+1)^{\text{th}}$ iteration. In each iteration, we set that $|\boldsymbol{\mathcal{P}}^t| \leq M, \forall t$. $c_1^t = \frac{1}{\eta_\lambda (t+1)^{\frac{1}{4}}} \geq \underline{c}_1$, $c_2^t = \frac{1}{\eta_\theta (t+1)^{\frac{1}{4}}} \geq \underline{c}_2$ are two nonnegative non-increasing sequences, where $\eta_\lambda$ and $\eta_\theta$ are positive constants, and constants $\underline{c}_1$, $\underline{c}_2$ meet that $0 < \underline{c}_1 \leq 1/\eta_\lambda c$, $0 < \underline{c}_2 \leq 1/\eta_\theta c$, $c = ((4M\alpha_3/\eta_\lambda{}^2 + 4N\alpha_4/\eta_\theta{}^2)^2 1/\epsilon^2 + 1)^{\frac{1}{4}}$ ($\epsilon$, $\alpha_3$, $\alpha_4$ are introduced in Section 4).

Following (Zhang & Kwok, 2014), to alleviate the staleness issue in ADBO, we set that master updates its variables once it receives updates from $S$ active workers at every iteration and every worker has to communicate with the master at least once every $\tau$ iterations. In $(t+1)^{\text{th}}$ iteration, let $\boldsymbol{\mathcal{Q}}^{t+1}$ denote the index set of active workers, the proposed algorithm proceeds as follows,

(1) *Active workers* update the local variables as follows,

$$\boldsymbol{x}_i^{t+1} = \begin{cases} \boldsymbol{x}_i^t - \eta_{\boldsymbol{x}} \nabla_{\boldsymbol{x}_i} \widetilde{L}_p(\{\boldsymbol{x}_i^{\hat{t}_i}\}, \{\boldsymbol{y}_i^{\hat{t}_i}\}, \boldsymbol{v}^{\hat{t}_i}, \boldsymbol{z}^{\hat{t}_i}, \{\lambda_l^{\hat{t}_i}\}, \{\boldsymbol{\theta}_i^{\hat{t}_i}\}), i \in \boldsymbol{\mathcal{Q}}^{t+1} \\ \boldsymbol{x}_i^t, i \notin \boldsymbol{\mathcal{Q}}^{t+1} \end{cases}, \tag{15}$$

$$\boldsymbol{y}_i^{t+1} = \begin{cases} \boldsymbol{y}_i^t - \eta_{\boldsymbol{y}} \nabla_{\boldsymbol{y}_i} \widetilde{L}_p(\{\boldsymbol{x}_i^{\hat{t}_i}\}, \{\boldsymbol{y}_i^{\hat{t}_i}\}, \boldsymbol{v}^{\hat{t}_i}, \boldsymbol{z}^{\hat{t}_i}, \{\lambda_l^{\hat{t}_i}\}, \{\boldsymbol{\theta}_i^{\hat{t}_i}\}), i \in \boldsymbol{\mathcal{Q}}^{t+1} \\ \boldsymbol{y}_i^t, i \notin \boldsymbol{\mathcal{Q}}^{t+1} \end{cases}, \tag{16}$$

where $\hat{t}_i$ denotes the last iteration during which worker $i$ was active, $\eta_{\boldsymbol{x}}$ and $\eta_{\boldsymbol{y}}$ are step-sizes. Then, the active workers transmit the local variables $\boldsymbol{x}_i^{t+1}$ and $\boldsymbol{y}_i^{t+1}$ to the master.

(2) After receiving the updates from active workers, the *master* updates the variables as follows,

$$\boldsymbol{v}^{t+1} = \boldsymbol{v}^t - \eta_{\boldsymbol{v}} \nabla_{\boldsymbol{v}} \widetilde{L}_p(\{\boldsymbol{x}_i^{t+1}\}, \{\boldsymbol{y}_i^{t+1}\}, \boldsymbol{v}^t, \boldsymbol{z}^t, \{\lambda_l^t\}, \{\boldsymbol{\theta}_i^t\}), \tag{17}$$

$$\boldsymbol{z}^{t+1} = \boldsymbol{z}^t - \eta_{\boldsymbol{z}} \nabla_{\boldsymbol{z}} \widetilde{L}_p(\{\boldsymbol{x}_i^{t+1}\}, \{\boldsymbol{y}_i^{t+1}\}, \boldsymbol{v}^{t+1}, \boldsymbol{z}^t, \{\lambda_l^t\}, \{\boldsymbol{\theta}_i^t\}), \tag{18}$$

$$\lambda_l^{t+1} = \lambda_l^t + \eta_\lambda \nabla_{\lambda_l} \widetilde{L}_p(\{\boldsymbol{x}_i^{t+1}\}, \{\boldsymbol{y}_i^{t+1}\}, \boldsymbol{v}^{t+1}, \boldsymbol{z}^{t+1}, \{\lambda_l^t\}, \{\boldsymbol{\theta}_i^t\}), \tag{19}$$

$$\boldsymbol{\theta}_i^{t+1} = \begin{cases} \boldsymbol{\theta}_i^t + \eta_{\boldsymbol{\theta}} \nabla_{\boldsymbol{\theta}_i} \widetilde{L}_p(\{\boldsymbol{x}_i^{t+1}\}, \{\boldsymbol{y}_i^{t+1}\}, \boldsymbol{v}^{t+1}, \boldsymbol{z}^{t+1}, \{\lambda_l^{t+1}\}, \{\boldsymbol{\theta}_i^t\}), i \in \boldsymbol{\mathcal{Q}}^{t+1} \\ \boldsymbol{\theta}_i^t, i \notin \boldsymbol{\mathcal{Q}}^{t+1} \end{cases}, \tag{20}$$

where $\eta_{\boldsymbol{v}}, \eta_{\boldsymbol{z}}, \eta_\lambda$ and $\eta_{\boldsymbol{\theta}}$ are step-sizes. Next, the master broadcasts $\boldsymbol{v}^{t+1}, \boldsymbol{z}^{t+1}, \boldsymbol{\theta}_i^{t+1}$ and $\{\lambda_l^{t+1}\}$ to worker $i, i \in \boldsymbol{\mathcal{Q}}^{t+1}$ (*i.e.*, active workers). Details are summarized in Algorithm 1.

### 3.4 Updating Cutting Planes

Every $k_{\text{pre}}$ iterations ($k_{\text{pre}} > 0$ is a pre-set constant, which can be controlled flexibly), the cutting planes are updated based on the following two steps (a) and (b) when $t < T_1$:

**(a) Removing the inactive cutting planes,**

$$\boldsymbol{\mathcal{P}}^{t+1} = \begin{cases} \text{Drop}(\boldsymbol{\mathcal{P}}^t, cp_l), \text{if } \lambda_l^{t+1} \text{ and } \lambda_l^t = 0 \\ \boldsymbol{\mathcal{P}}^t, \text{otherwise} \end{cases}, \tag{21}$$

where $cp_l$ represents the $l^{\text{th}}$ cutting plane in $\boldsymbol{\mathcal{P}}^t$ and $\text{Drop}(\boldsymbol{\mathcal{P}}^t, cp_l)$ represents the $l^{\text{th}}$ cutting plane $cp_l$ is removed from $\boldsymbol{\mathcal{P}}^t$. The dual variable set $\{\lambda^{t+1}\}$ will be updated as follows,

$$\{\lambda^{t+1}\} = \begin{cases} \text{Drop}(\{\lambda^t\}, \lambda_l), \text{if } \lambda_l^{t+1} \text{ and } \lambda_l^t = 0 \\ \{\lambda^t\}, \text{otherwise} \end{cases}, \tag{22}$$

where $\{\lambda^{t+1}\}$ and $\{\lambda^t\}$ represent the dual variable set in $(t+1)^{\text{th}}$ and $t^{\text{th}}$ iterations, respectively. $\text{Drop}(\{\lambda^t\}, \lambda_l)$ represents that $\lambda_l$ is removed from the dual variable set $\{\lambda^t\}$.

**(b) Adding new cutting planes.** Firstly, we investigate whether $(\boldsymbol{v}^{t+1}, \{\boldsymbol{y}_i^{t+1}\}, \boldsymbol{z}^{t+1})$ is feasible for the constraint $h(\boldsymbol{v}, \{\boldsymbol{y}_i\}, \boldsymbol{z}) \le \varepsilon$. We can obtain $h(\boldsymbol{v}^{t+1}, \{\boldsymbol{y}_i^{t+1}\}, \boldsymbol{z}^{t+1})$ according to $\phi(\boldsymbol{v}^{t+1})$ in Eq. (9). If $(\boldsymbol{v}^{t+1}, \{\boldsymbol{y}_i^{t+1}\}, \boldsymbol{z}^{t+1})$ is not a feasible solution to the original problem (Eq. (10)), new cutting plane $cp_{new}^{t+1}$ will be generated to separate the point $(\boldsymbol{v}^{t+1}, \{\boldsymbol{y}_i^{t+1}\}, \boldsymbol{z}^{t+1})$ from the feasible region of constraint $h(\boldsymbol{v}, \{\boldsymbol{y}_i\}, \boldsymbol{z}) \le \varepsilon$. Thus, the *valid* cutting plane (Boyd & Vandenberghe, 2007) $\boldsymbol{a}_l^\top \boldsymbol{v} + \sum_{i=1}^N \boldsymbol{b}_{i,l}^\top \boldsymbol{y}_i + \boldsymbol{c}_l^\top \boldsymbol{z} + \kappa_l \le 0$ must satisfy that,

$$\begin{cases} \boldsymbol{a}_l^\top \boldsymbol{v} + \sum_{i=1}^N \boldsymbol{b}_{i,l}^\top \boldsymbol{y}_i + \boldsymbol{c}_l^\top \boldsymbol{z} + \kappa_l \le 0, \forall (\boldsymbol{v}, \{\boldsymbol{y}_i\}, \boldsymbol{z}) \text{ satisfies } h(\boldsymbol{v}, \{\boldsymbol{y}_i\}, \boldsymbol{z}) \le \varepsilon \\ \boldsymbol{a}_l^\top \boldsymbol{v}^{t+1} + \sum_{i=1}^N \boldsymbol{b}_{i,l}^\top \boldsymbol{y}_i^{t+1} + \boldsymbol{c}_l^\top \boldsymbol{z}^{t+1} + \kappa_l > 0 \end{cases}. \tag{23}$$

Since $h(\boldsymbol{v}, \{\boldsymbol{y}_i\}, \boldsymbol{z})$ is a convex function, we have that,

$$h(\boldsymbol{v}, \{\boldsymbol{y}_i\}, \boldsymbol{z}) \ge h(\boldsymbol{v}^{t+1}, \{\boldsymbol{y}_i^{t+1}\}, \boldsymbol{z}^{t+1}) + \begin{bmatrix} \frac{\partial h(\boldsymbol{v}^{t+1}, \{\boldsymbol{y}_i^{t+1}\}, \boldsymbol{z}^{t+1})}{\partial \boldsymbol{v}} \\ \{\frac{\partial h(\boldsymbol{v}^{t+1}, \{\boldsymbol{y}_i^{t+1}\}, \boldsymbol{z}^{t+1})}{\partial \boldsymbol{y}_i}\} \\ \frac{\partial h(\boldsymbol{v}^{t+1}, \{\boldsymbol{y}_i^{t+1}\}, \boldsymbol{z}^{t+1})}{\partial \boldsymbol{z}} \end{bmatrix}^\top \left( \begin{bmatrix} \boldsymbol{v} \\ \{\boldsymbol{y}_i\} \\ \boldsymbol{z} \end{bmatrix} - \begin{bmatrix} \boldsymbol{v}^{t+1} \\ \{\boldsymbol{y}_i^{t+1}\} \\ \boldsymbol{z}^{t+1} \end{bmatrix} \right). \tag{24}$$

Combining Eq. (24) with Eq. (23), we have that a valid cutting plane (with respect to point $(\boldsymbol{v}^{t+1}, \{\boldsymbol{y}_i^{t+1}\}, \boldsymbol{z}^{t+1})$) can be expressed as,

$$h(\boldsymbol{v}^{t+1}, \{\boldsymbol{y}_i^{t+1}\}, \boldsymbol{z}^{t+1}) + \begin{bmatrix} \frac{\partial h(\boldsymbol{v}^{t+1}, \{\boldsymbol{y}_i^{t+1}\}, \boldsymbol{z}^{t+1})}{\partial \boldsymbol{v}} \\ \{\frac{\partial h(\boldsymbol{v}^{t+1}, \{\boldsymbol{y}_i^{t+1}\}, \boldsymbol{z}^{t+1})}{\partial \boldsymbol{y}_i}\} \\ \frac{\partial h(\boldsymbol{v}^{t+1}, \{\boldsymbol{y}_i^{t+1}\}, \boldsymbol{z}^{t+1})}{\partial \boldsymbol{z}} \end{bmatrix}^\top \left( \begin{bmatrix} \boldsymbol{v} \\ \{\boldsymbol{y}_i\} \\ \boldsymbol{z} \end{bmatrix} - \begin{bmatrix} \boldsymbol{v}^{t+1} \\ \{\boldsymbol{y}_i^{t+1}\} \\ \boldsymbol{z}^{t+1} \end{bmatrix} \right) \le \varepsilon. \tag{25}$$

For brevity, we utilize $cp_{new}^{t+1}$ to denote the new added cutting plane (*i.e.*, Eq. (25)). Thus the polytope $\boldsymbol{\mathcal{P}}^{t+1}$ will be updated as follows,

$$\boldsymbol{\mathcal{P}}^{t+1} = \begin{cases} \text{Add}(\boldsymbol{\mathcal{P}}^{t+1}, cp_{new}^{t+1}), \text{if } h(\boldsymbol{v}^{t+1}, \{\boldsymbol{y}_i^{t+1}\}, \boldsymbol{z}^{t+1}) > \varepsilon \\ \boldsymbol{\mathcal{P}}^{t+1}, \text{otherwise} \end{cases}, \tag{26}$$

where $\text{Add}(\boldsymbol{\mathcal{P}}^{t+1}, cp_{new}^{t+1})$ represents that new cutting plane $cp_{new}^{t+1}$ is added to polytope $\boldsymbol{\mathcal{P}}^{t+1}$. The dual variable set $\{\lambda^{t+1}\}$ is updated as follows,

$$\{\lambda^{t+1}\} = \begin{cases} \text{Add}(\{\lambda^{t+1}\}, \lambda_{|\boldsymbol{\mathcal{P}}^{t+1}|}^{t+1}), \text{if } h(\boldsymbol{v}^{t+1}, \{\boldsymbol{y}_i^{t+1}\}, \boldsymbol{z}^{t+1}) > \varepsilon \\ \{\lambda^{t+1}\}, \text{otherwise} \end{cases}, \tag{27}$$

where $\text{Add}(\{\lambda^{t+1}\}, \lambda_{|\boldsymbol{\mathcal{P}}^{t+1}|}^{t+1})$ represents that dual variable $\lambda_{|\boldsymbol{\mathcal{P}}^{t+1}|}^{t+1}$ is added to the dual variable set $\{\lambda^{t+1}\}$. Finally, master broadcasts the updated $\boldsymbol{\mathcal{P}}^{t+1}$ and $\{\lambda^{t+1}\}$ to all workers. The details of the proposed algorithm are summarized in Algorithm 1.

---

**Algorithm 1** ADBO: Asynchronous Distributed Bilevel Optimization

---

**Initialization:** master iteration $t = 0$, variables $\{\boldsymbol{x}_i^0\}$, $\{\boldsymbol{y}_i^0\}$, $\boldsymbol{v}^0$, $\boldsymbol{z}^0$, $\{\lambda_l^0\}$, $\{\boldsymbol{\theta}_i^0\}$ and polytope $\mathcal{P}^0$.
**repeat**
    **for** *active worker* **do**
        updates variables $\boldsymbol{x}_i^{t+1}$, $\boldsymbol{y}_i^{t+1}$ according to Eq. (15) and (16);
    **end for**
    Active workers transmit their local variables to master;
    **for** *master* **do**
        updates variables $\boldsymbol{v}^{t+1}$, $\boldsymbol{z}^{t+1}$, $\{\lambda_l^{t+1}\}$, $\{\boldsymbol{\theta}_i^{t+1}\}$ according to Eq. (17), (18), (19) and (20);
    **end for**
    master broadcasts variables to active workers;
    **if** $(t + 1)$ mod $k_{\text{pre}} == 0$ and $t < T_1$ **then**
        master computes $\phi(\boldsymbol{v}^{t+1})$ according to Eq. (9);
        master updates $\mathcal{P}^{t+1}$ and $\{\lambda^{t+1}\}$ according to Eq. (21), (22), (26) and (27);
        master broadcasts $\mathcal{P}^{t+1}$ and $\{\lambda^{t+1}\}$ to all workers;
    **end if**
    $t = t + 1$;
**until** termination.

---

## 4 DISCUSSION

**Theorem 1** *(**Convergence**) As the cutting plane continues to be added to the polytope, the optimal objective value of approximate problem in Eq. (12) converges monotonically.*

The proof of Theorem 1 is presented in Appendix C.

**Definition 1** *(**Stationarity gap**) Following (Xu et al., 2020; Lu et al., 2020; Jiao et al., 2022a), the stationarity gap of our problem at $t^{th}$ iteration is defined as:*

$$
\nabla G^t = \begin{bmatrix}
\{\nabla_{\boldsymbol{x}_i} L_p(\{\boldsymbol{x}_i^t\}, \{\boldsymbol{y}_i^t\}, \boldsymbol{v}^t, \boldsymbol{z}^t, \{\lambda_l^t\}, \{\boldsymbol{\theta}_i^t\})\} \\
\{\nabla_{\boldsymbol{y}_i} L_p(\{\boldsymbol{x}_i^t\}, \{\boldsymbol{y}_i^t\}, \boldsymbol{v}^t, \boldsymbol{z}^t, \{\lambda_l^t\}, \{\boldsymbol{\theta}_i^t\})\} \\
\nabla_{\boldsymbol{v}} L_p(\{\boldsymbol{x}_i^t\}, \{\boldsymbol{y}_i^t\}, \boldsymbol{v}^t, \boldsymbol{z}^t, \{\lambda_l^t\}, \{\boldsymbol{\theta}_i^t\}) \\
\nabla_{\boldsymbol{z}} L_p(\{\boldsymbol{x}_i^t\}, \{\boldsymbol{y}_i^t\}, \boldsymbol{v}^t, \boldsymbol{z}^t, \{\lambda_l^t\}, \{\boldsymbol{\theta}_i^t\}) \\
\{\nabla_{\lambda_l} L_p(\{\boldsymbol{x}_i^t\}, \{\boldsymbol{y}_i^t\}, \boldsymbol{v}^t, \boldsymbol{z}^t, \{\lambda_l^t\}, \{\boldsymbol{\theta}_i^t\})\} \\
\{\nabla_{\boldsymbol{\theta}_i} L_p(\{\boldsymbol{x}_i^t\}, \{\boldsymbol{y}_i^t\}, \boldsymbol{v}^t, \boldsymbol{z}^t, \{\lambda_l^t\}, \{\boldsymbol{\theta}_i^t\})\}
\end{bmatrix}. \tag{28}
$$

**Definition 2** *(**$\epsilon$-stationary point**) $(\{\boldsymbol{x}_i^t\}, \{\boldsymbol{y}_i^t\}, \boldsymbol{v}^t, \boldsymbol{z}^t, \{\lambda_l^t\}, \{\boldsymbol{\theta}_i^t\})$ is an $\epsilon$-stationary point ($\epsilon \geq 0$) of a differentiable function $L_p$, if $||\nabla G^t||^2 \leq \epsilon$. $T(\epsilon)$ is the first iteration index such that $||\nabla G^t||^2 \leq \epsilon$, i.e., $T(\epsilon) = \min\{t \mid ||\nabla G^t||^2 \leq \epsilon\}$.*

**Assumption 1** *(**Smoothness/Gradient Lipschitz**) Following (Ji et al., 2021), we assume that $L_p$ has Lipschitz continuous gradients, i.e., for any $\boldsymbol{\omega}, \boldsymbol{\omega}'$, we assume that there exists $L > 0$ satisfying that,*

$$
||\nabla L_p(\boldsymbol{\omega}) - \nabla L_p(\boldsymbol{\omega}')|| \leq L||\boldsymbol{\omega} - \boldsymbol{\omega}'||, \tag{29}
$$

**Assumption 2** *(**Boundedness**) Following (Qian et al., 2019), we assume that variables are bounded, i.e., $||\boldsymbol{x}_i||^2 \leq \alpha_1, ||\boldsymbol{v}||^2 \leq \alpha_1, ||\boldsymbol{y}_i||^2 \leq \alpha_2, ||\boldsymbol{z}||^2 \leq \alpha_2, ||\lambda_l||^2 \leq \alpha_3, ||\boldsymbol{\theta}_i||^2 \leq \alpha_4$. And we assume that before obtaining the $\epsilon$-stationary point (i.e., $t \leq T(\epsilon) - 1$), the variables in master satisfy that $||\boldsymbol{v}^{t+1} - \boldsymbol{v}^t||^2 + ||\boldsymbol{z}^{t+1} - \boldsymbol{z}^t||^2 + \sum_l ||\lambda_l^{t+1} - \lambda_l^t||^2 \geq \vartheta$, where $\vartheta > 0$ is a relative small constant. The change of the variables in master is upper bounded within $\tau$ iterations:*

$$
||\boldsymbol{v}^t - \boldsymbol{v}^{t-k}||^2 \leq \tau k_1 \vartheta, \quad ||\boldsymbol{z}^t - \boldsymbol{z}^{t-k}||^2 \leq \tau k_1 \vartheta, \quad \sum_l ||\lambda_l^t - \lambda_l^{t-k}||^2 \leq \tau k_1 \vartheta, \forall 1 \leq k \leq \tau, \tag{30}
$$

*where $k_1 > 0$ is a constant.*

**Theorem 2** *(**Iteration complexity**) Suppose Assumption 1 and 2 hold, we set the step-sizes as $\eta_{\boldsymbol{x}} = \eta_{\boldsymbol{y}} = \eta_{\boldsymbol{v}} = \eta_{\boldsymbol{z}} = \frac{2}{L + \eta_\lambda M L^2 + \eta_{\boldsymbol{\theta}} N L^2 + 8(\frac{M\gamma L^2}{\eta_\lambda c_1^2} + \frac{N\gamma L^2}{\eta_{\boldsymbol{\theta}} c_2^2})}$, $\eta_\lambda < \min\{\frac{2}{L + 2c_1^0}, \frac{1}{30\tau k_1 N L^2}\}$ and $\eta_{\boldsymbol{\theta}} \leq \frac{2}{L + 2c_2^0}$. For a given $\epsilon$, we have:*

$$
T(\epsilon) \sim \mathcal{O}(\max\{(\frac{4M\alpha_3}{\eta_\lambda^2} + \frac{4N\alpha_4}{\eta_{\boldsymbol{\theta}}^2})^2 \frac{1}{\epsilon^2}, (\frac{4(d_7 + \frac{\eta_{\boldsymbol{\theta}}(N-S)L^2}{2})(\bar{d} + k_d\tau(\tau - 1))d_6}{\epsilon} + (T_1 + 2)^{\frac{1}{2}})^2\}), \tag{31}
$$

*where $\alpha_3$, $\alpha_4$, $\gamma$, $k_d$, $T_1$, $\bar{d}$, $d_6$ and $d_7$ are constants. The detailed proof is given in Appendix B.*

## 5 EXPERIMENT

In this section, experiments[1] are conducted on two hyperparameter optimization tasks (*i.e.*, data hyper-cleaning task and regularization coefficient optimization task) in the distributed setting to evaluate the performance of the proposed ADBO. The proposed ADBO is compared with the state-of-the-art distributed bilevel optimization method FEDNEST (Tarzanagh et al., 2022). In data hyper-cleaning task, experiments are carried out on MNIST (LeCun et al., 1998) and Fashion MNIST (Xiao et al., 2017) datasets. In coefficient optimization task, following (Chen et al., 2022a), experiments are conducted on Covertype (Blackard & Dean, 1999) and IJCNN1 (Prokhorov, 2001) datasets.

### 5.1 DATA HYPER-CLEANING

Following (Ji et al., 2021; Yang et al., 2021), we compare the performance of the proposed ADBO and distributed bilevel optimization method FEDNEST on the distributed data hyper-cleaning task (Chen et al., 2022b) on MNIST and Fashion MNIST datasets. Data hyper-cleaning involves training a classifier in a contaminated environment where each training data label is changed to a random class number with a probability (*i.e.*, the corruption rate). In the experiment, the distributed data hyper-cleaning problem is considered, whose formulation can be expressed as,

$$
\begin{aligned}
\min\ & F(\boldsymbol{\psi}, \boldsymbol{w}) = \sum_{i=1}^{N} \frac{1}{|\mathcal{D}_i^{\mathrm{val}}|} \sum_{(\mathbf{x}_j, y_j) \in \mathcal{D}_i^{\mathrm{val}}} \mathcal{L}(\mathbf{x}_j^\top \boldsymbol{w}, y_j) \\
\text{s.t.}\ & \boldsymbol{w} = \arg\min_{\boldsymbol{w}'} f(\boldsymbol{\psi}, \boldsymbol{w}') = \sum_{i=1}^{N} \frac{1}{|\mathcal{D}_i^{\mathrm{tr}}|} \sum_{(\mathbf{x}_j, y_j) \in \mathcal{D}_i^{\mathrm{tr}}} \sigma(\psi_j) \mathcal{L}(\mathbf{x}_j^\top \boldsymbol{w}', y_j) + C_r \|\boldsymbol{w}'\|^2 \\
\text{var.}\ & \boldsymbol{\psi}, \boldsymbol{w},
\end{aligned}
\tag{32}
$$

where $\mathcal{D}_i^{\mathrm{tr}}$ and $\mathcal{D}_i^{\mathrm{val}}$ denote the training and validation datasets on $i^{\mathrm{th}}$ worker, respectively. $(\mathbf{x}_j, y_j)$ denote the $j^{\mathrm{th}}$ data and label. $\sigma(.)$ is the sigmoid function, $\mathcal{L}$ is the cross-entropy loss, $C_r$ is a regularization parameter and $N$ is the number of workers in the distributed system. In MNIST and Fashion MNIST datasets, we set $N = 18$, $S = 9$ and $\tau = 15$. According to Cohen et al. (2021), we assume that the communication delay of each worker obeys the heavy-tailed distribution. The proposed ADBO is compared with the state-of-the-art distributed bilevel optimization method FEDNEST and SDBO (Synchronous Distributed Bilevel Optimization, *i.e.*, ADBO without asynchronous setting). The test accuracy versus time is shown in Figure 1, and the test loss versus time is shown in Figure 2. We can observe that the proposed ADBO is the most efficient algorithm since 1) the asynchronous setting is considered in ADBO, the master can update its variables once it receives updates from $S$ active workers instead of all workers; and 2) ADBO is a single-loop algorithm and only gradient descent/ascent is required at each iteration, thus ADBO is computationally more efficient.

### 5.2 REGULARIZATION COEFFICIENT OPTIMIZATION

Following (Chen et al., 2022a), we compare the proposed ADBO with baseline algorithms FEDNEST and SDBO on the regularization coefficient optimization task with Covertype and IJCNN1 datasets. The distributed regularization coefficient optimization problem is given by,

$$
\begin{aligned}
\min\ & F(\boldsymbol{\psi}, \boldsymbol{w}) = \sum_{i=1}^{N} \frac{1}{|\mathcal{D}_i^{\mathrm{val}}|} \sum_{(\mathbf{x}_j, y_j) \in \mathcal{D}_i^{\mathrm{val}}} \mathcal{L}(\mathbf{x}_j^\top \boldsymbol{w}, y_j) \\
\text{s.t.}\ & \boldsymbol{w} = \arg\min_{\boldsymbol{w}'} f(\boldsymbol{\psi}, \boldsymbol{w}') = \sum_{i=1}^{N} \frac{1}{|\mathcal{D}_i^{\mathrm{tr}}|} \sum_{(\mathbf{x}_j, y_j) \in \mathcal{D}_i^{\mathrm{tr}}} \mathcal{L}(\mathbf{x}_j^\top \boldsymbol{w}', y_j) + \sum_{j=1}^{n} \psi_j(w_j')^2 \\
\text{var.}\ & \boldsymbol{\psi}, \boldsymbol{w},
\end{aligned}
\tag{33}
$$

where $\boldsymbol{\psi} \in \mathbb{R}^n$, $\boldsymbol{w} \in \mathbb{R}^n$ and $\mathcal{L}$ respectively denote the regularization coefficient, model parameter, and logistic loss, and $\boldsymbol{w}' = [w_1', \dots, w_n']$. In Covertype and IJCNN1 datasets, we set $N = 18$, $S = 9$, $\tau = 15$ and $N = 24$, $S = 12$, $\tau = 15$, respectively. We also assume that the delay of each worker obeys the heavy-tailed distribution. Firstly, we compare the performance of the proposed ADBO, SDBO and FEDNEST in terms of test accuracy and test loss on Covertype and IJCNN1 datasets,

---

[1]Codes are available in `https://github.com/ICLR23Submission6251/adbo`.

which are shown in Figure 3 and 4. It is seen that the proposed ADBO is more efficient because of the same two reasons we gave in Section 5.1.

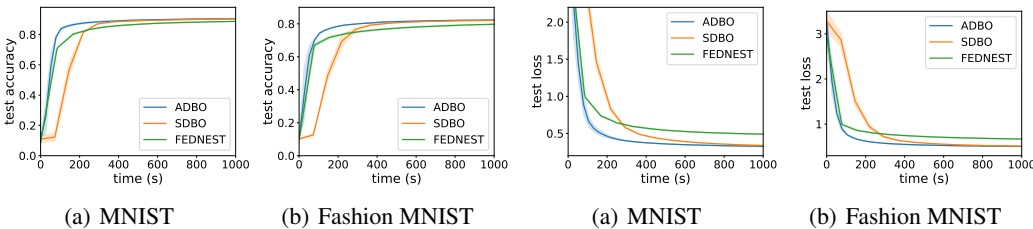

(a) MNIST      (b) Fashion MNIST      (a) MNIST      (b) Fashion MNIST

Figure 1: Test accuracy vs time on (a) MNIST and (b) Fashion MNIST datasets.

Figure 2: Test loss vs time on (a) MNIST and (b) Fashion MNIST datasets.

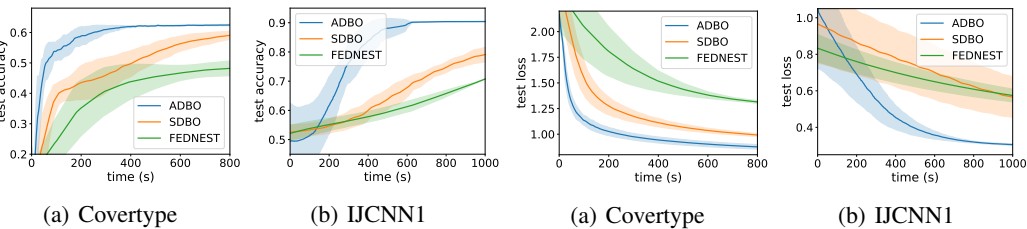

(a) Covertype      (b) IJCNN1      (a) Covertype      (b) IJCNN1

Figure 3: Test accuracy vs time on (a) Covertype and (b) IJCNN1 datasets.

Figure 4: Test loss vs time on (a) Covertype and (b) IJCNN1 datasets.

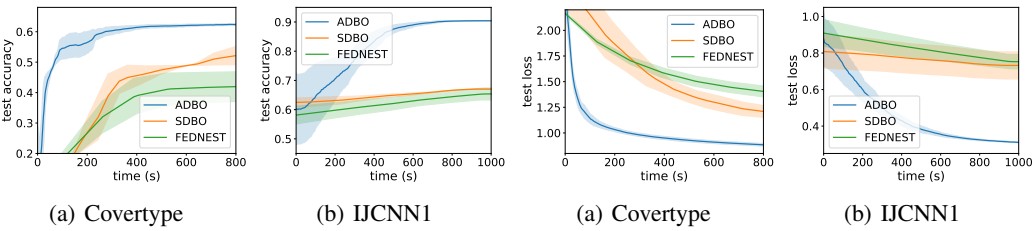

(a) Covertype      (b) IJCNN1      (a) Covertype      (b) IJCNN1

Figure 5: Test accuracy vs time on (a) Covertype and (b) IJCNN1 datasets when there are stragglers in distributed system.

Figure 6: Test loss vs time on (a) Covertype and (b) IJCNN1 datasets when there are stragglers in distributed system.

We also consider the straggler problem, *i.e.*, there exist workers with high delays (stragglers) in the distributed system. In this case, the efficiency of the bilevel optimization method with the synchronous distributed setting will be affected heavily. In the experiment, we assume there are three stragglers in the distributed system, and the mean of (communication + computation) delay of stragglers is four times the delay of normal workers. The results on Covertype and IJCNN1 datasets are reported in Figure 5 and 6. It is seen that the efficiency of the synchronous distributed algorithms (FEDNEST and SDBO) will be significantly affected, while the proposed ADBO does not suffer from the straggler problem since it is an asynchronous method and is able to only consider active workers.

# 6 CONCLUSION

Existing bilevel optimization works focus either on the centralized or synchronous distributed setting, which will give rise to data privacy risks and suffer from the straggler problem. As a remedy, we propose ADBO in this paper to solve the bilevel optimization problem in an asynchronous distributed manner. To our best knowledge, this is the first work that devises the asynchronous distributed algorithm for bilevel optimization. We demonstrate that the proposed ADBO can effectively tackle bilevel optimization problems with both nonconvex upper-level and lower-level objective functions. Theoretical analysis has also been conducted to analyze the convergence properties and iteration complexity of ADBO. Extensive empirical studies on real-world datasets demonstrate the efficiency and effectiveness of the proposed ADBO.

ACKNOWLEDGMENTS

The work of Yang Jiao, Kai Yang and Chengtao Jian was supported in part by the National Natural Science Foundation of China under Grant 61771013, in part by the Shenzhen Institute of Artificial Intelligence and Robotics for Society (AIRS), in part by the Fundamental Research Funds for the Central Universities of China, and in part by the Fundamental Research Funds of Shanghai Jiading District. We thank Haibo Zhao for providing the experiment results in meta-learning.

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

## A  CUTTING PLANE METHOD FOR BILEVEL OPTIMIZATION

In this section, a cutting plane method, named CPBO, is proposed for bileve optimization. Defining $\phi(\boldsymbol{x}) = \arg\min_{\boldsymbol{y}'} f(\boldsymbol{x}, \boldsymbol{y}')$ and $h(\boldsymbol{x}, \boldsymbol{y}) = ||\boldsymbol{y} - \phi(\boldsymbol{x})||^2$, we can reformulate problem in Eq. (1) as:

$$
\begin{aligned}
\min \quad & F(\boldsymbol{x}, \boldsymbol{y}) \\
\text{s.t.} \quad & h(\boldsymbol{x}, \boldsymbol{y}) = 0 \\
\text{var.} \quad & \boldsymbol{x}, \boldsymbol{y}.
\end{aligned}
\tag{34}
$$

Following the previous works (Li et al., 2022; Gould et al., 2016; Yang et al., 2021) in bilevel optimization, it is not necessary to get the exact $\phi(\boldsymbol{x})$, and the approximate $\phi(\boldsymbol{x})$ is given as follows. Firstly, as many work do (Ji et al., 2021; Yang et al., 2021), we utilize the $K$ steps of gradient descent (GD) to approximate $\phi(\boldsymbol{x})$. And the first-order Taylor approximation of $f(\boldsymbol{x}, \boldsymbol{y}')$ with respect to $\boldsymbol{x}$ is considered, *i.e.*, for a given point $\overline{\boldsymbol{x}}$, $\widetilde{f}(\boldsymbol{x}, \boldsymbol{y}') = f(\overline{\boldsymbol{x}}, \boldsymbol{y}') + \nabla_{\boldsymbol{x}} f(\overline{\boldsymbol{x}}, \boldsymbol{y}')^\top (\boldsymbol{x} - \overline{\boldsymbol{x}})$. Thus, we have,

$$
\phi(\boldsymbol{x}) = \boldsymbol{y}_0' - \sum_{k=0}^{K-1} \eta \nabla_{\boldsymbol{y}} \widetilde{f}(\boldsymbol{x}, \boldsymbol{y}_k'),
\tag{35}
$$

where $\eta$ is the step-size. Considering the estimated $\phi(\boldsymbol{x})$ in Eq. (35), the relaxed problem with respect to problem in Eq. (34) is considered as follows,

$$
\begin{aligned}
\min \quad & F(\boldsymbol{x}, \boldsymbol{y}) \\
\text{s.t.} \quad & h(\boldsymbol{x}, \boldsymbol{y}) \leq \varepsilon \\
\text{var.} \quad & \boldsymbol{x}, \boldsymbol{y}.
\end{aligned}
\tag{36}
$$

Assuming that $h(\boldsymbol{x}, \boldsymbol{y})$ is a convex function with respect to $(\boldsymbol{x}, \boldsymbol{y})$, which is always satisfied when we set $K = 1$ in Eq. (35) according to the operations that preserve convexity (Boyd et al., 2004). Since the sublevel set of a convex function is convex, we have that the feasible set of $(\boldsymbol{x}, \boldsymbol{y})$, *i.e.*,

$$
\boldsymbol{Z}^{relax} = \{(\boldsymbol{x}, \boldsymbol{y}) \in \mathbb{R}^n \times \mathbb{R}^m | h(\boldsymbol{x}, \boldsymbol{y}) \leq \epsilon\},
\tag{37}
$$

is a convex set. We utilize a set of cutting plane constraints (*i.e.*, linear constraints) to approximate the feasible set $\boldsymbol{Z}^{relax}$. The set of cutting plane constraints forms a polytope, which can be expressed as follows,

$$
\boldsymbol{\mathcal{P}} = \{(\boldsymbol{x}, \boldsymbol{y}) \in \mathbb{R}^n \times \mathbb{R}^m | \boldsymbol{a}_l^\top \boldsymbol{x} + \boldsymbol{b}_l^\top \boldsymbol{y} + \kappa_l \leq 0, \ l = 1, \cdots, L\},
\tag{38}
$$

where $\boldsymbol{a}_l \in \mathbb{R}^n$, $\boldsymbol{b}_l \in \mathbb{R}^m$ and $\kappa_l \in \mathbb{R}^1$ are parameters in $l^{\text{th}}$ cutting plane, and $L$ represents the number of cutting planes in $\boldsymbol{\mathcal{P}}$. Considering the approximate problem, which can be expressed as follows,

$$
\begin{aligned}
\min \quad & F(\boldsymbol{x}, \boldsymbol{y}) \\
\text{s.t.} \quad & \boldsymbol{a}_l^\top \boldsymbol{x} + \boldsymbol{b}_l^\top \boldsymbol{y} + \kappa_l \leq 0, \ l = 1, \cdots, |\boldsymbol{\mathcal{P}}^t| \\
\text{var.} \quad & \boldsymbol{x}, \boldsymbol{y},
\end{aligned}
\tag{39}
$$

where $\boldsymbol{\mathcal{P}}^t$ is the polytope in $(t+1)^{\text{th}}$ iteration, and $|\boldsymbol{\mathcal{P}}^t|$ denotes the number of cutting planes in $\boldsymbol{\mathcal{P}}^t$. Then, the Lagrangian function of Eq. (39) can be written as,

$$
L_p(\boldsymbol{x}, \boldsymbol{y}, \{\lambda_l\}) = F(\boldsymbol{x}, \boldsymbol{y}) + \sum_{l=1}^{|\boldsymbol{\mathcal{P}}^t|} \lambda_l (\boldsymbol{a}_l^\top \boldsymbol{x} + \boldsymbol{b}_l^\top \boldsymbol{y} + \kappa_l),
\tag{40}
$$

where $\lambda_l$ is the dual variable. The proposed algorithm proceeds as follows in $(t+1)^{\text{th}}$ iteration:

If $t < T_1$, the variables are updated as follows,

$$
\boldsymbol{x}^{t+1} = \boldsymbol{x}^t - \eta_{\boldsymbol{x}} \nabla_{\boldsymbol{x}} L_p(\boldsymbol{x}^t, \boldsymbol{y}^t, \{\lambda_l^t\}),
\tag{41}
$$

$$
\boldsymbol{y}^{t+1} = \boldsymbol{y}^t - \eta_{\boldsymbol{y}} \nabla_{\boldsymbol{y}} L_p(\boldsymbol{x}^{t+1}, \boldsymbol{y}^t, \{\lambda_l^t\}),
\tag{42}
$$

$$
\lambda_l^{t+1} = \lambda_l^t + \eta_{\lambda_l} \nabla_{\lambda_l} L_p(\boldsymbol{x}^{t+1}, \boldsymbol{y}^{t+1}, \{\lambda_l^t\}), \ l = 1, \cdots, |\boldsymbol{\mathcal{P}}^t|,
\tag{43}
$$

where $\eta_{\boldsymbol{x}}$, $\eta_{\boldsymbol{y}}$ and $\eta_{\lambda_l}$ are the step-sizes.

And every $k_{\text{pre}}$ iterations ($k_{\text{pre}} > 0$ is a pre-set constant, which can be controlled flexibly) the cutting planes will be updated based on the following two steps:

Table 1: Convergence results of bilevel optimization algorithms (with centralized and distributed setting).

| Method | Centralized | Synchronous (Distributed) | Asynchronous (Distributed) |
|---|---|---|---|
| AID-BiO (Ghadimi & Wang, 2018) | $\mathcal{O}(\frac{1}{\epsilon^{1.25}})$ | NA | NA |
| AID-BiO (Ji et al., 2021) | $\mathcal{O}(\frac{1}{\epsilon^1})$ | NA | NA |
| ITD-BiO (Ji et al., 2021) | $\mathcal{O}(\frac{1}{\epsilon^1})$ | NA | NA |
| STABLE (Chen et al., 2022a) | $\mathcal{O}(\frac{1}{\epsilon^2})^1$ | NA | NA |
| stocBio (Ji et al., 2021) | $\mathcal{O}(\frac{1}{\epsilon^2})^1$ | NA | NA |
| VRBO (Yang et al., 2021) | $\mathcal{O}(\frac{1}{\epsilon^{1.5}})^1$ | NA | NA |
| FEDNEST (Tarzanagh et al., 2022) | NA | $\mathcal{O}(\frac{1}{\epsilon^2})^1$ | NA |
| SPDB (Lu et al., 2022) | NA | $\mathcal{O}(\frac{1}{\epsilon^2})^1$ | NA |
| DSBO (Yang et al., 2022) | NA | $\mathcal{O}(\frac{1}{\epsilon^2})^1$ | NA |
| **Proposed Method** | $\mathcal{O}(\frac{1}{\epsilon^1})$ | NA | $\mathcal{O}(\frac{1}{\epsilon^2})$ |

[1] Stochastic optimization algorithm.

(a) Removing the inactive cutting planes, that is,

$$\boldsymbol{\mathcal{P}}^{t+1} = \begin{cases} \mathrm{Drop}(\boldsymbol{\mathcal{P}}^t, cp_l), \text{if } \lambda_l^{t+1} \text{ and } \lambda_l^t = 0 \\ \boldsymbol{\mathcal{P}}^t, \text{otherwise} \end{cases}, \tag{44}$$

where $cp_l$ represents the $l^{\text{th}}$ cutting plane in $\boldsymbol{\mathcal{P}}^t$, and $\mathrm{Drop}(\boldsymbol{\mathcal{P}}^t, cp_l)$ represents removing the $l^{\text{th}}$ cutting plane $cp_l$ from $\boldsymbol{\mathcal{P}}^t$. And the dual variable set $\{\lambda^t\}$ will be updated as follows,

$$\{\lambda^{t+1}\} = \begin{cases} \mathrm{Drop}(\{\lambda^t\}, \lambda_l^t), \text{if } \lambda_l^{t+1} \text{ and } \lambda_l^t = 0 \\ \{\lambda^t\}, \text{otherwise} \end{cases}, \tag{45}$$

where $\{\lambda^{t+1}\}$ and $\{\lambda^t\}$ respectively represent the dual variable set in $(t+1)^{\text{th}}$ and $t^{\text{th}}$ iteration. And $\mathrm{Drop}(\{\lambda^t\}, \lambda_l^t)$ represents that $\lambda_l^t$ is removed from the dual variable set $\{\lambda^t\}$.

(b) Adding new cutting planes. Firstly, we investigate whether $(\boldsymbol{x}^{t+1}, \boldsymbol{y}^{t+1})$ is a feasible solution to the original problem in Eq. (36). If $(\boldsymbol{x}^{t+1}, \boldsymbol{y}^{t+1})$ is not a feasible solution to the original problem, that is $h(\boldsymbol{x}^{t+1}, \boldsymbol{y}^{t+1}) > \varepsilon$, new cutting plane is generated to separate the point $(\boldsymbol{x}^{t+1}, \boldsymbol{y}^{t+1})$ from $\boldsymbol{Z}^{relax}$, that is, the *valid* cutting plane $\boldsymbol{a}_l^\top \boldsymbol{x} + \boldsymbol{b}_l^\top \boldsymbol{y} + \kappa_l \leq 0$ must satisfy that,

$$\begin{cases} \boldsymbol{a}_l^\top \boldsymbol{x} + \boldsymbol{b}_l^\top \boldsymbol{y} + \kappa_l \leq 0, \forall (\boldsymbol{x}, \boldsymbol{y}) \in \boldsymbol{Z}^{relax} \\ \boldsymbol{a}_l^\top \boldsymbol{x}^{t+1} + \boldsymbol{b}_l^\top \boldsymbol{y}^{t+1} + \kappa_l > 0 \end{cases}. \tag{46}$$

Since $h(\boldsymbol{x}, \boldsymbol{y})$ is a convex function, we have that,

$$h(\boldsymbol{x}, \boldsymbol{y}) \geq h(\boldsymbol{x}^{t+1}, \boldsymbol{y}^{t+1}) + \begin{bmatrix} \frac{\partial h(\boldsymbol{x}^{t+1}, \boldsymbol{y}^{t+1})}{\partial \boldsymbol{x}} \\ \frac{\partial h(\boldsymbol{x}^{t+1}, \boldsymbol{y}^{t+1})}{\partial \boldsymbol{y}} \end{bmatrix}^\top \left( \begin{bmatrix} \boldsymbol{x} \\ \boldsymbol{y} \end{bmatrix} - \begin{bmatrix} \boldsymbol{x}^{t+1} \\ \boldsymbol{y}^{t+1} \end{bmatrix} \right). \tag{47}$$

According to Eq. (47), $h(\boldsymbol{x}^{t+1}, \boldsymbol{y}^{t+1}) + \begin{bmatrix} \frac{\partial h(\boldsymbol{x}^{t+1}, \boldsymbol{y}^{t+1})}{\partial \boldsymbol{x}} \\ \frac{\partial h(\boldsymbol{x}^{t+1}, \boldsymbol{y}^{t+1})}{\partial \boldsymbol{y}} \end{bmatrix}^\top \left( \begin{bmatrix} \boldsymbol{x} \\ \boldsymbol{y} \end{bmatrix} - \begin{bmatrix} \boldsymbol{x}^{t+1} \\ \boldsymbol{y}^{t+1} \end{bmatrix} \right) \leq \varepsilon$ is a valid

cutting plane at point $(\boldsymbol{x}^{t+1}, \boldsymbol{y}^{t+1})$ which satisfies Eq. (46). For brevity, we utilize $cp_{new}^{t+1}$ to denote this cutting plane. Thus, we have that,

$$\boldsymbol{\mathcal{P}}^{t+1} = \begin{cases} \mathrm{Add}(\boldsymbol{\mathcal{P}}^{t+1}, cp_{new}^{t+1}), \text{if } h(\boldsymbol{x}^{t+1}, \boldsymbol{y}^{t+1}) > \varepsilon \\ \boldsymbol{\mathcal{P}}^{t+1}, \text{if } h(\boldsymbol{x}^{t+1}, \boldsymbol{y}^{t+1}) \leq \varepsilon \end{cases}, \tag{48}$$

where $\mathrm{Add}(\boldsymbol{\mathcal{P}}^{t+1}, cp_{new}^{t+1})$ represents that new cutting plane $cp_{new}^{t+1}$ is added to polytope $\boldsymbol{\mathcal{P}}^{t+1}$. And the dual variable set is updated as follows,

$$\{\lambda^{t+1}\} = \begin{cases} \mathrm{Add}(\{\lambda^{t+1}\}, \lambda_{|\boldsymbol{\mathcal{P}}^{t+1}|}^{t+1}), \text{if } h(\boldsymbol{x}^{t+1}, \boldsymbol{y}^{t+1}) > \varepsilon \\ \{\lambda^{t+1}\}, \text{if } h(\boldsymbol{x}^{t+1}, \boldsymbol{y}^{t+1}) \leq \varepsilon \end{cases}, \tag{49}$$

---

**Algorithm 2** CPBO: Cutting Plane Method for Bilevel Optimization

---

**Initialization:** iteration $t = 0$, variables $\boldsymbol{x}^0$, $\boldsymbol{y}^0$, $\{\lambda_l^0\}$ and polytope $\boldsymbol{\mathcal{P}}^0$.
**repeat**
    **if** $t < T_1$ **then**
        updating variables $\boldsymbol{x}^{t+1}$, $\boldsymbol{y}^{t+1}$ and $\lambda_l^{t+1}$ according to Eq. (41), (42) and (43);
        **if** $(t+1) \bmod k_{\text{pre}} == 0$ **then**
            updating the polytope $\boldsymbol{\mathcal{P}}^{t+1}$ according to Eq. (44) and (48);
            updating the dual variable set $\{\lambda^{t+1}\}$ according to Eq. (45) and (49);
        **end if**
    **else**
        updating variables $\boldsymbol{x}^{t+1}$ and $\boldsymbol{y}^{t+1}$ according to Eq. (50) and (51);
    **end if**
    $t = t + 1$;
**until** termination.

---

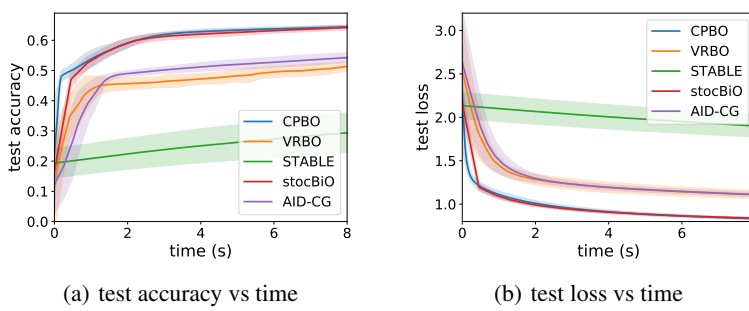

(a) test accuracy vs time        (b) test loss vs time

Figure 7: Comparison of (a) test accuracy vs time, (b) test loss vs time on Covertype dataset.

where $\text{Add}(\{\lambda^{t+1}\}, \lambda^{t+1}_{|\boldsymbol{\mathcal{P}}^{t+1}|})$ represents that new dual variable $\lambda^{t+1}_{|\boldsymbol{\mathcal{P}}^{t+1}|}$ is added to $\{\lambda^{t+1}\}$.

Else if $t \geq T_1$, the polytope $\boldsymbol{\mathcal{P}}^{T_1}$ and dual variables will be fixed. Variables $\boldsymbol{x}, \boldsymbol{y}$ will be updated as follows,

$$\boldsymbol{x}^{t+1} = \boldsymbol{x}^t - \eta_{\boldsymbol{x}} \nabla_{\boldsymbol{x}} \hat{L}_p(\boldsymbol{x}^t, \boldsymbol{y}^t), \tag{50}$$

$$\boldsymbol{y}^{t+1} = \boldsymbol{y}^t - \eta_{\boldsymbol{y}} \nabla_{\boldsymbol{y}} \hat{L}_p(\boldsymbol{x}^{t+1}, \boldsymbol{y}^t), \tag{51}$$

where $\hat{L}_p(\boldsymbol{x}, \boldsymbol{y}) = F(\boldsymbol{x}, \boldsymbol{y}) + \sum_{l=1}^{|\boldsymbol{\mathcal{P}}^{T_1}|} \lambda_l [\max\{0, \boldsymbol{a}_l^\top \boldsymbol{x} + \boldsymbol{b}_l^\top \boldsymbol{y} + \kappa_l\}]^2$. And details of the proposed algorithm are summarized in Algorithm 2. The comparison about the convergence results between the proposed method and state-of-the-art methods are summarized in Table 1.

## A.1 EXPERIMENT

To evaluate the performance of the proposed CPBO, experiments are carried out on two applications: 1) hyperparameter optimization, 2) meta-learning. In hyperparameter optimization, we compare CPBO with baseline algorithms stocBio (Ji et al., 2021), STABLE (Chen et al., 2022a), VRBO (Yang et al., 2021)), and AID-CG (Grazzi et al., 2020) on the regularization coefficient optimization task (Chen et al., 2022a) with Covertype (Blackard & Dean, 1999) and IJCNN1 (Prokhorov, 2001) datasets. We compare the performance of the proposed CPBO with all competing algorithms in terms of both the test accuracy and the test loss, which are shown in Figure 7 and 8. In meta-learning, we focus on the bilevel optimization problem in (Rajeswaran et al., 2019). And we compare the proposed CPBO with baseline algorithms MAML (Finn et al., 2017), iMAML (Rajeswaran et al., 2019), and ANIL (Raghu et al., 2019) on Omniglot (Lake et al., 2015) and CIFAR-FS (Bertinetto et al., 2018) datasets. And the comparison between the proposed method with the baseline algorithms are shown in Figure 9 and 10. It is seen that the proposed CPBO can achieve relatively fast convergence rate among all competing algorithms since 1) the iteration complexity of the proposed method is not high; 2) every step in CPBO is computationally efficient.

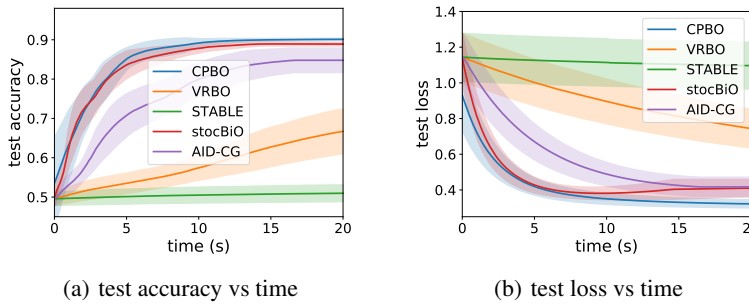

(a) test accuracy vs time          (b) test loss vs time

Figure 8: Comparison of (a) test accuracy vs time, (b) test loss vs time on IJCNN1 dataset.

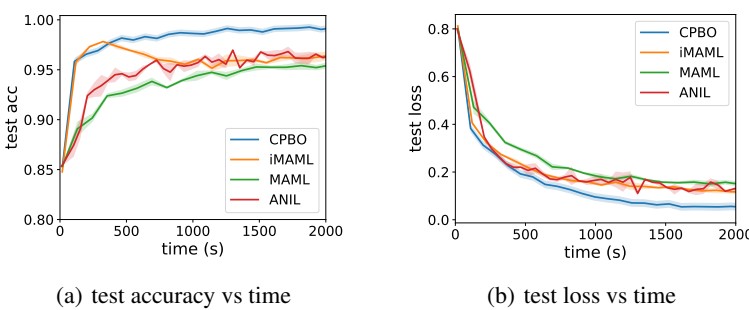

(a) test accuracy vs time          (b) test loss vs time

Figure 9: Comparison of (a) test accuracy vs time, (b) test loss vs time on Omniglot dataset.

## A.2 DISCUSSION

**Definition A.1** $(\boldsymbol{x}, \boldsymbol{y})$ *is an $\epsilon$-stationary point of a differentiable function $\hat{L}_p$, if $||\nabla_{\boldsymbol{x}}\hat{L}_p(\boldsymbol{x}, \boldsymbol{y})||^2 + ||\nabla_{\boldsymbol{y}}\hat{L}_p(\boldsymbol{x}, \boldsymbol{y})||^2 \leq \epsilon.$*

**Assumption A.1** *(**Smoothness/Gradient Lipschitz**) Following (Ji et al., 2021), we assume that $\hat{L}_p$ has Lipschitz continuous gradients, i.e., for any $\boldsymbol{\omega}, \boldsymbol{\omega}'$, we assume that there exists $L > 0$ satisfying that,*

$$||\nabla\hat{L}_p(\boldsymbol{\omega}) - \nabla\hat{L}_p(\boldsymbol{\omega}')|| \leq L||\boldsymbol{\omega} - \boldsymbol{\omega}'||. \tag{52}$$

**Assumption A.2** *(**Boundedness**) Following (Qian et al., 2019), we assume that variables have boundedness, i.e., $||\boldsymbol{x}||^2 \leq \beta_1, ||\boldsymbol{y}||^2 \leq \beta_2.$*

**Theorem 3** *(**Iteration Complexity**) Under Assumption A.1, A.2, and setting the step-sizes as $\eta_{\boldsymbol{x}} < \frac{2}{L}, \eta_{\boldsymbol{y}} < \frac{2}{L}$, the iteration complexity (also the gradient complexity) of the proposed algorithm to obtain $\epsilon$-stationary point is bounded by $\mathcal{O}(\frac{1}{\epsilon})$.*

***Proof of Theorem 3:***

According to Assumption A.1 and Eq. (50), when $t \geq T_1$, we have,

$$\begin{aligned}
\hat{L}_p(\boldsymbol{x}^{t+1}, \boldsymbol{y}^t) &\leq \hat{L}_p(\boldsymbol{x}^t, \boldsymbol{y}^t) + \left\langle \nabla_{\boldsymbol{x}}\hat{L}_p(\boldsymbol{x}^t, \boldsymbol{y}^t), \boldsymbol{x}^{t+1} - \boldsymbol{x}^t \right\rangle + \frac{L}{2}||\boldsymbol{x}^{t+1} - \boldsymbol{x}^t||^2 \\
&\leq \hat{L}_p(\boldsymbol{x}^t, \boldsymbol{y}^t) - \eta_{\boldsymbol{x}}||\nabla_{\boldsymbol{x}}\hat{L}_p(\boldsymbol{x}^t, \boldsymbol{y}^t)||^2 + \frac{L\eta_{\boldsymbol{x}}^2}{2}||\nabla_{\boldsymbol{x}}\hat{L}_p(\boldsymbol{x}^t, \boldsymbol{y}^t)||^2.
\end{aligned} \tag{53}$$

Similarly, according to Assumption A.1 and Eq. (51), we have,

$$\begin{aligned}
\hat{L}_p(\boldsymbol{x}^{t+1}, \boldsymbol{y}^{t+1}) &\leq \hat{L}_p(\boldsymbol{x}^{t+1}, \boldsymbol{y}^t) + \left\langle \nabla_{\boldsymbol{y}}\hat{L}_p(\boldsymbol{x}^{t+1}, \boldsymbol{y}^t), \boldsymbol{y}^{t+1} - \boldsymbol{y}^t \right\rangle + \frac{L}{2}||\boldsymbol{y}^{t+1} - \boldsymbol{y}^t||^2 \\
&\leq \hat{L}_p(\boldsymbol{x}^{t+1}, \boldsymbol{y}^t) - \eta_{\boldsymbol{y}}||\nabla_{\boldsymbol{y}}\hat{L}_p(\boldsymbol{x}^{t+1}, \boldsymbol{y}^t)||^2 + \frac{L\eta_{\boldsymbol{y}}^2}{2}||\nabla_{\boldsymbol{y}}\hat{L}_p(\boldsymbol{x}^{t+1}, \boldsymbol{y}^t)||^2.
\end{aligned} \tag{54}$$

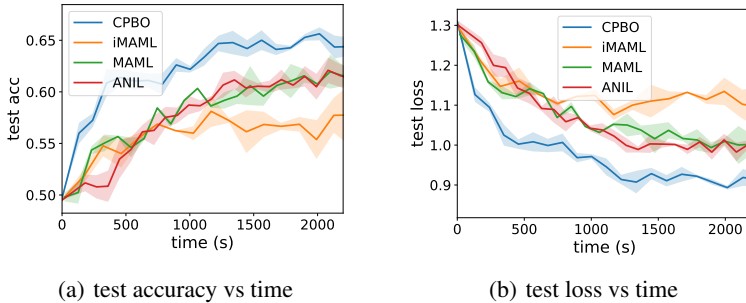

(a) test accuracy vs time       (b) test loss vs time

Figure 10: Comparison of (a) test accuracy vs time, (b) test loss vs time on CIFAR-FS dataset.

Combining Eq. (53) with Eq. (54), we have,

$$(\eta_{\boldsymbol{x}} - \frac{L\eta_{\boldsymbol{x}}^2}{2})||\nabla_{\boldsymbol{x}}\hat{L}_p(\boldsymbol{x}^t, \boldsymbol{y}^t)||^2 + (\eta_{\boldsymbol{y}} - \frac{L\eta_{\boldsymbol{y}}^2}{2})||\nabla_{\boldsymbol{y}}\hat{L}_p(\boldsymbol{x}^{t+1}, \boldsymbol{y}^t)||^2 \le \hat{L}_p(\boldsymbol{x}^t, \boldsymbol{y}^t) - \hat{L}_p(\boldsymbol{x}^{t+1}, \boldsymbol{y}^{t+1}).$$
(55)

According to the setting of $\eta_{\boldsymbol{x}}$, $\eta_{\boldsymbol{y}}$, we have that $\eta_{\boldsymbol{x}} - \frac{L\eta_{\boldsymbol{x}}^2}{2} > 0$, $\eta_{\boldsymbol{y}} - \frac{L\eta_{\boldsymbol{y}}^2}{2} > 0$. And we set constant $d = \min\{\eta_{\boldsymbol{x}} - \frac{L\eta_{\boldsymbol{x}}^2}{2}, \eta_{\boldsymbol{y}} - \frac{L\eta_{\boldsymbol{y}}^2}{2}\}$, thus we can obtain that,

$$||\nabla_{\boldsymbol{x}}\hat{L}_p(\boldsymbol{x}^t, \boldsymbol{y}^t)||^2 + ||\nabla_{\boldsymbol{y}}\hat{L}_p(\boldsymbol{x}^{t+1}, \boldsymbol{y}^t)||^2 \le \frac{\hat{L}_p(\boldsymbol{x}^t, \boldsymbol{y}^t) - \hat{L}_p(\boldsymbol{x}^{t+1}, \boldsymbol{y}^{t+1})}{d}.$$
(56)

Summing both sides of Eq. (56) for $t = \{T_1, \cdots, T-1\}$, we obtain that,

$$\frac{1}{T - T_1}\sum_{t=T_1}^{T-1}(||\nabla_{\boldsymbol{x}}\hat{L}_p(\boldsymbol{x}^t, \boldsymbol{y}^t)||^2 + ||\nabla_{\boldsymbol{y}}\hat{L}_p(\boldsymbol{x}^{t+1}, \boldsymbol{y}^t)||^2) \le \frac{\hat{L}_p(\boldsymbol{x}^{T_1}, \boldsymbol{y}^{T_1}) - \hat{L}_p^*}{(T - T_1)d},$$
(57)

where $\hat{L}_p^* = \min \hat{L}_p(\boldsymbol{x}, \boldsymbol{y})$. Combining Eq. (57) with Definition A.1, we have that the number of iterations required by Algorithm 2 to return an $\epsilon$-stationary point is bounded by

$$\mathcal{O}(\frac{\hat{L}_p(\boldsymbol{x}^{T_1}, \boldsymbol{y}^{T_1}) - \hat{L}_p^*}{d}\frac{1}{\epsilon} + T_1).$$
(58)

# B    PROOF OF THEOREM 2

In this section, we provide complete proofs for Theorem 2. Firstly, we make some definitions about our problem.

**Definition B.1** *Following (Xu et al., 2020), the stationarity gap at $t^{th}$ iteration is defined as:*

$$\nabla G^t = \begin{bmatrix} \{\nabla_{\boldsymbol{x}_i}L_p(\{\boldsymbol{x}_i^t\},\{\boldsymbol{y}_i^t\},\boldsymbol{v}^t,\boldsymbol{z}^t,\{\lambda_l^t\},\{\boldsymbol{\theta}_i^t\})\} \\ \{\nabla_{\boldsymbol{y}_i}L_p(\{\boldsymbol{x}_i^t\},\{\boldsymbol{y}_i^t\},\boldsymbol{v}^t,\boldsymbol{z}^t,\{\lambda_l^t\},\{\boldsymbol{\theta}_i^t\})\} \\ \nabla_{\boldsymbol{v}}L_p(\{\boldsymbol{x}_i^t\},\{\boldsymbol{y}_i^t\},\boldsymbol{v}^t,\boldsymbol{z}^t,\{\lambda_l^t\},\{\boldsymbol{\theta}_i^t\}) \\ \nabla_{\boldsymbol{z}}L_p(\{\boldsymbol{x}_i^t\},\{\boldsymbol{y}_i^t\},\boldsymbol{v}^t,\boldsymbol{z}^t,\{\lambda_l^t\},\{\boldsymbol{\theta}_i^t\}) \\ \{\nabla_{\lambda_l}L_p(\{\boldsymbol{x}_i^t\},\{\boldsymbol{y}_i^t\},\boldsymbol{v}^t,\boldsymbol{z}^t,\{\lambda_l^t\},\{\boldsymbol{\theta}_i^t\})\} \\ \{\nabla_{\boldsymbol{\theta}_i}L_p(\{\boldsymbol{x}_i^t\},\{\boldsymbol{y}_i^t\},\boldsymbol{v}^t,\boldsymbol{z}^t,\{\lambda_l^t\},\{\boldsymbol{\theta}_i^t\})\} \end{bmatrix}.$$
(59)

*And we also define:*

$$
\begin{aligned}
(\nabla G^t)_{\boldsymbol{x}_i} &= \nabla_{\boldsymbol{x}_i} L_p(\{\boldsymbol{x}_i^t\},\{\boldsymbol{y}_i^t\},\boldsymbol{v}^t,\boldsymbol{z}^t,\{\lambda_l^t\},\{\boldsymbol{\theta}_i^t\}), \\
(\nabla G^t)_{\boldsymbol{y}_i} &= \nabla_{\boldsymbol{y}_i} L_p(\{\boldsymbol{x}_i^t\},\{\boldsymbol{y}_i^t\},\boldsymbol{v}^t,\boldsymbol{z}^t,\{\lambda_l^t\},\{\boldsymbol{\theta}_i^t\}), \\
(\nabla G^t)_{\boldsymbol{v}} &= \nabla_{\boldsymbol{v}} L_p(\{\boldsymbol{x}_i^t\},\{\boldsymbol{y}_i^t\},\boldsymbol{v}^t,\boldsymbol{z}^t,\{\lambda_l^t\},\{\boldsymbol{\theta}_i^t\}), \\
(\nabla G^t)_{\boldsymbol{z}} &= \nabla_{\boldsymbol{z}} L_p(\{\boldsymbol{x}_i^t\},\{\boldsymbol{y}_i^t\},\boldsymbol{v}^t,\boldsymbol{z}^t,\{\lambda_l^t\},\{\boldsymbol{\theta}_i^t\}), \\
(\nabla G^t)_{\lambda_l} &= \nabla_{\lambda_l} L_p(\{\boldsymbol{x}_i^t\},\{\boldsymbol{y}_i^t\},\boldsymbol{v}^t,\boldsymbol{z}^t,\{\lambda_l^t\},\{\boldsymbol{\theta}_i^t\}), \\
(\nabla G^t)_{\boldsymbol{\theta}_i} &= \nabla_{\boldsymbol{\theta}_i} L_p(\{\boldsymbol{x}_i^t\},\{\boldsymbol{y}_i^t\},\boldsymbol{v}^t,\boldsymbol{z}^t,\{\lambda_l^t\},\{\boldsymbol{\theta}_i^t\}).
\end{aligned}
\tag{60}
$$

*It follows that,*

$$
||\nabla G^t||^2 = \sum_{i=1}^{N}(||(\nabla G^t)_{\boldsymbol{x}_i}||^2 + ||(\nabla G^t)_{\boldsymbol{y}_i}||^2 + ||(\nabla G^t)_{\boldsymbol{\theta}_i}||^2) + ||(\nabla G^t)_{\boldsymbol{v}}||^2 + ||(\nabla G^t)_{\boldsymbol{z}}||^2 + \sum_{l=1}^{|\boldsymbol{\mathcal{P}}^t|} ||(\nabla G^t)_{\lambda_l}||^2.
\tag{61}
$$

**Definition B.2** *At $t^{th}$ iteration, the stationarity gap w.r.t $\widetilde{L}_p(\{\boldsymbol{x}_i\},\{\boldsymbol{y}_i\},\boldsymbol{v},\boldsymbol{z},\{\lambda_l\},\{\boldsymbol{\theta}_i\})$ is defined as:*

$$
\nabla \widetilde{G}^t =
\begin{bmatrix}
\{\nabla_{\boldsymbol{x}_i}\widetilde{L}_p(\{\boldsymbol{x}_i^t\},\{\boldsymbol{y}_i^t\},\boldsymbol{v}^t,\boldsymbol{z}^t,\{\lambda_l^t\},\{\boldsymbol{\theta}_i^t\})\} \\
\{\nabla_{\boldsymbol{y}_i}\widetilde{L}_p(\{\boldsymbol{x}_i^t\},\{\boldsymbol{y}_i^t\},\boldsymbol{v}^t,\boldsymbol{z}^t,\{\lambda_l^t\},\{\boldsymbol{\theta}_i^t\})\} \\
\nabla_{\boldsymbol{v}}\widetilde{L}_p(\{\boldsymbol{x}_i^t\},\{\boldsymbol{y}_i^t\},\boldsymbol{v}^t,\boldsymbol{z}^t,\{\lambda_l^t\},\{\boldsymbol{\theta}_i^t\}) \\
\nabla_{\boldsymbol{z}}\widetilde{L}_p(\{\boldsymbol{x}_i^t\},\{\boldsymbol{y}_i^t\},\boldsymbol{v}^t,\boldsymbol{z}^t,\{\lambda_l^t\},\{\boldsymbol{\theta}_i^t\}) \\
\{\nabla_{\lambda_l}\widetilde{L}_p(\{\boldsymbol{x}_i^t\},\{\boldsymbol{y}_i^t\},\boldsymbol{v}^t,\boldsymbol{z}^t,\{\lambda_l^t\},\{\boldsymbol{\theta}_i^t\})\} \\
\{\nabla_{\boldsymbol{\theta}_i}\widetilde{L}_p(\{\boldsymbol{x}_i^t\},\{\boldsymbol{y}_i^t\},\boldsymbol{v}^t,\boldsymbol{z}^t,\{\lambda_l^t\},\{\boldsymbol{\theta}_i^t\})\}
\end{bmatrix}.
\tag{62}
$$

*We further define:*

$$
\begin{aligned}
(\nabla \widetilde{G}^t)_{\boldsymbol{x}_i} &= \nabla_{\boldsymbol{x}_i} \widetilde{L}_p(\{\boldsymbol{x}_i^t\},\{\boldsymbol{y}_i^t\},\boldsymbol{v}^t,\boldsymbol{z}^t,\{\lambda_l^t\},\{\boldsymbol{\theta}_i^t\}), \\
(\nabla \widetilde{G}^t)_{\boldsymbol{y}_i} &= \nabla_{\boldsymbol{y}_i} \widetilde{L}_p(\{\boldsymbol{x}_i^t\},\{\boldsymbol{y}_i^t\},\boldsymbol{v}^t,\boldsymbol{z}^t,\{\lambda_l^t\},\{\boldsymbol{\theta}_i^t\}), \\
(\nabla \widetilde{G}^t)_{\boldsymbol{v}} &= \nabla_{\boldsymbol{v}} \widetilde{L}_p(\{\boldsymbol{x}_i^t\},\{\boldsymbol{y}_i^t\},\boldsymbol{v}^t,\boldsymbol{z}^t,\{\lambda_l^t\},\{\boldsymbol{\theta}_i^t\}), \\
(\nabla \widetilde{G}^t)_{\boldsymbol{z}} &= \nabla_{\boldsymbol{z}} \widetilde{L}_p(\{\boldsymbol{x}_i^t\},\{\boldsymbol{y}_i^t\},\boldsymbol{v}^t,\boldsymbol{z}^t,\{\lambda_l^t\},\{\boldsymbol{\theta}_i^t\}), \\
(\nabla \widetilde{G}^t)_{\lambda_l} &= \nabla_{\lambda_l} \widetilde{L}_p(\{\boldsymbol{x}_i^t\},\{\boldsymbol{y}_i^t\},\boldsymbol{v}^t,\boldsymbol{z}^t,\{\lambda_l^t\},\{\boldsymbol{\theta}_i^t\}), \\
(\nabla \widetilde{G}^t)_{\boldsymbol{\theta}_i} &= \nabla_{\boldsymbol{\theta}_i} \widetilde{L}_p(\{\boldsymbol{x}_i^t\},\{\boldsymbol{y}_i^t\},\boldsymbol{v}^t,\boldsymbol{z}^t,\{\lambda_l^t\},\{\boldsymbol{\theta}_i^t\}).
\end{aligned}
\tag{63}
$$

*It follows that,*

$$
||\nabla \widetilde{G}^t||^2 = \sum_{i=1}^{N}(||(\nabla \widetilde{G}^t)_{\boldsymbol{x}_i}||^2 + ||(\nabla \widetilde{G}^t)_{\boldsymbol{y}_i}||^2 + ||(\nabla \widetilde{G}^t)_{\boldsymbol{\theta}_i}||^2) + ||(\nabla \widetilde{G}^t)_{\boldsymbol{v}}||^2 + ||(\nabla \widetilde{G}^t)_{\boldsymbol{z}}||^2 + \sum_{l=1}^{|\boldsymbol{\mathcal{P}}^t|} ||(\nabla \widetilde{G}^t)_{\lambda_l}||^2.
\tag{64}
$$

**Definition B.3** *In the proposed asynchronous algorithm, for the $i^{th}$ worker in $t^{th}$ iteration, the last iteration where this worker was active is defined as $\hat{t}_i$. And the next iteration this worker will be active is defined as $\bar{t}_i$. For the iteration index set which $i^{th}$ worker is active during $T_1 + T + \tau$ iteration, it is defined as $\mathcal{V}_i(T)$. And the $j^{th}$ element in $\mathcal{V}_i(T)$ is defined as $\hat{v}_i(j)$.*

Then, we provide some useful lemmas used for proving the main convergence results in Theorem 2.

**Lemma 1** *Let sequences $\eta_{\boldsymbol{x}}^t = \eta_{\boldsymbol{y}}^t = \eta_{\boldsymbol{v}}^t = \eta_{\boldsymbol{z}}^t = \frac{2}{L+\eta_\lambda|\mathcal{P}^t|L^2+\eta_{\boldsymbol{\theta}}NL^2+8(\frac{|\mathcal{P}^t|\gamma L^2}{\eta_\lambda(c_1^t)^2}+\frac{N\gamma L^2}{\eta_{\boldsymbol{\theta}}(c_2^t)^2})}$, suppose Assumption 1 and 2 hold, we can obtain that,*

$$
L_p(\{\boldsymbol{x}_i^{t+1}\},\{\boldsymbol{y}_i^{t+1}\},\boldsymbol{v}^{t+1},\boldsymbol{z}^{t+1},\{\lambda_l^t\},\{\boldsymbol{\theta}_i^t\}) - L_p(\{\boldsymbol{x}_i^t\},\{\boldsymbol{y}_i^t\},\boldsymbol{v}^t,\boldsymbol{z}^t,\{\lambda_l^t\},\{\boldsymbol{\theta}_i^t\})
$$
$$
\leq \sum_{i=1}^N (\tfrac{L+L^2+1}{2} - \tfrac{1}{\eta_{\boldsymbol{x}}^t})||\boldsymbol{x}_i^{t+1}-\boldsymbol{x}_i^t||^2 + \sum_{i=1}^N (\tfrac{L+1}{2} - \tfrac{1}{\eta_{\boldsymbol{y}}^t})||\boldsymbol{y}_i^{t+1}-\boldsymbol{y}_i^t||^2 + 3NL^2\tau k_1 \sum_{l=1}^{|\mathcal{P}^t|} ||\lambda_l^{t+1}-\lambda_l^t||^2
$$
$$
+(\tfrac{L+6NL^2\tau k_1}{2} - \tfrac{1}{\eta_{\boldsymbol{v}}^t})||\boldsymbol{v}^{t+1}-\boldsymbol{v}^t||^2 + (\tfrac{L+6NL^2\tau k_1}{2} - \tfrac{1}{\eta_{\boldsymbol{z}}^t})||\boldsymbol{z}^{t+1}-\boldsymbol{z}^t||^2.
$$
(65)

*Proof of Lemma 1:*

Utilizing the Lipschitz properties in Assumption 1, we can obtain that,

$$
L_p(\{\boldsymbol{x}_1^{t+1}, \boldsymbol{x}_2^t,\cdots,\boldsymbol{x}_N^t\},\{\boldsymbol{y}_i^t\},\boldsymbol{v}^t, \boldsymbol{z}^t,\{\lambda_l^t\},\{\boldsymbol{\theta}_i^t\}) - L_p(\{\boldsymbol{x}_i^t\},\{\boldsymbol{y}_i^t\},\boldsymbol{v}^t,\boldsymbol{z}^t,\{\lambda_l^t\},\{\boldsymbol{\theta}_i^t\})
$$
$$
\leq \langle \nabla_{\boldsymbol{x}_1} L_p(\{\boldsymbol{x}_i^t\},\{\boldsymbol{y}_i^t\},\boldsymbol{v}^t, \boldsymbol{z}^t,\{\lambda_l^t\},\{\boldsymbol{\theta}_i^t\}), \boldsymbol{x}_1^{t+1}-\boldsymbol{x}_1^t \rangle + \tfrac{L}{2}||\boldsymbol{x}_1^{t+1}-\boldsymbol{x}_1^t||^2,
$$

$$
L_p(\{\boldsymbol{x}_1^{t+1}, \boldsymbol{x}_2^{t+1},\cdots,\boldsymbol{x}_N^t\},\{\boldsymbol{y}_i^t\},\boldsymbol{v}^t,\boldsymbol{z}^t,\{\lambda_l^t\},\{\boldsymbol{\theta}_i^t\}) - L_p(\{\boldsymbol{x}_1^{t+1}, \boldsymbol{x}_2^t,\cdots,\boldsymbol{x}_N^t\},\{\boldsymbol{y}_i^t\},\boldsymbol{v}^t,\boldsymbol{z}^t,\{\lambda_l^t\},\{\boldsymbol{\theta}_i^t\})
$$
$$
\leq \langle \nabla_{\boldsymbol{x}_2} L_p(\{\boldsymbol{x}_i^t\},\{\boldsymbol{y}_i^t\},\boldsymbol{v}^t,\boldsymbol{z}^t,\{\lambda_l^t\},\{\boldsymbol{\theta}_i^t\}), \boldsymbol{x}_2^{t+1}-\boldsymbol{x}_2^t \rangle + \tfrac{L}{2}||\boldsymbol{x}_2^{t+1}-\boldsymbol{x}_2^t||^2,
$$

$$
\vdots
$$

$$
L_p(\{\boldsymbol{x}_i^{t+1}\},\{\boldsymbol{y}_i^t\},\boldsymbol{v}^t,\boldsymbol{z}^t,\{\lambda_l^t\},\{\boldsymbol{\theta}_i^t\}) - L_p(\{\boldsymbol{x}_1^{t+1},\cdots,\boldsymbol{x}_{N-1}^{t+1},\boldsymbol{x}_N^t\},\{\boldsymbol{y}_i^t\},\boldsymbol{v}^t,\boldsymbol{z}^t,\{\lambda_l^t\},\{\boldsymbol{\theta}_i^t\})
$$
$$
\leq \langle \nabla_{\boldsymbol{x}_N} L_p(\{\boldsymbol{x}_i^t\},\{\boldsymbol{y}_i^t\},\boldsymbol{v}^t,\boldsymbol{z}^t,\{\lambda_l^t\},\{\boldsymbol{\theta}_i^t\}), \boldsymbol{x}_N^{t+1}-\boldsymbol{x}_N^t \rangle + \tfrac{L}{2}||\boldsymbol{x}_N^{t+1}-\boldsymbol{x}_N^t||^2.
$$
(66)

Summing up the above inequalities in Eq. (66), we can obtain that,

$$
L_p(\{\boldsymbol{x}_i^{t+1}\},\{\boldsymbol{y}_i^t\},\boldsymbol{v}^t,\boldsymbol{z}^t,\{\lambda_l^t\},\{\boldsymbol{\theta}_i^t\}) - L_p(\{\boldsymbol{x}_i^t\},\{\boldsymbol{y}_i^t\},\boldsymbol{v}^t,\boldsymbol{z}^t,\{\lambda_l^t\},\{\boldsymbol{\theta}_i^t\})
$$
$$
\leq \sum_{i=1}^N \left( \langle \nabla_{\boldsymbol{x}_i} L_p(\{\boldsymbol{x}_i^t\},\{\boldsymbol{y}_i^t\},\boldsymbol{v}^t,\boldsymbol{z}^t,\{\lambda_l^t\},\{\boldsymbol{\theta}_i^t\}), \boldsymbol{x}_i^{t+1}-\boldsymbol{x}_i^t \rangle + \tfrac{L}{2}||\boldsymbol{x}_i^{t+1}-\boldsymbol{x}_i^t||^2 \right).
$$
(67)

Combining $\nabla_{\boldsymbol{x}_i} L_p(\{\boldsymbol{x}_i^{\hat{t}_i}\}, \{\boldsymbol{y}_i^{\hat{t}_i}\}, \boldsymbol{v}^{\hat{t}_i}, \boldsymbol{z}^{\hat{t}_i}, \{\lambda_l^{\hat{t}_i}\}, \{\boldsymbol{\theta}_i^{\hat{t}_i}\}) = \nabla_{\boldsymbol{x}_i} \widetilde{L}_p(\{\boldsymbol{x}_i^{\hat{t}_i}\}, \{\boldsymbol{y}_i^{\hat{t}_i}\}, \boldsymbol{v}^{\hat{t}_i}, \boldsymbol{z}^{\hat{t}_i}, \{\lambda_l^{\hat{t}_i}\}, \{\boldsymbol{\theta}_i^{\hat{t}_i}\})$ with Eq. (15), we have that,

$$
\left\langle \boldsymbol{x}_i^{t+1}-\boldsymbol{x}_i^t, \nabla_{\boldsymbol{x}_i} L_p(\{\boldsymbol{x}_i^{\hat{t}_i}\}, \{\boldsymbol{y}_i^{\hat{t}_i}\}, \boldsymbol{v}^{\hat{t}_i}, \boldsymbol{z}^{\hat{t}_i}, \{\lambda_l^{\hat{t}_i}\}, \{\boldsymbol{\theta}_i^{\hat{t}_i}\}) \right\rangle = -\frac{1}{\eta_{\boldsymbol{x}}}||\boldsymbol{x}_i^{t+1}-\boldsymbol{x}_i^t||^2 \leq -\frac{1}{\eta_{\boldsymbol{x}}^t}||\boldsymbol{x}_i^{t+1}-\boldsymbol{x}_i^t||^2.
$$
(68)

Next, combining the Cauchy-Schwarz inequality with Assumption 1, 2, we can get,

$$
\left\langle \boldsymbol{x}_i^{t+1}-\boldsymbol{x}_i^t, \nabla_{\boldsymbol{x}_i} L_p(\{\boldsymbol{x}_i^t\},\{\boldsymbol{y}_i^t\},\boldsymbol{v}^t,\boldsymbol{z}^t,\{\lambda_l^t\},\{\boldsymbol{\theta}_i^t\}) - \nabla_{\boldsymbol{x}_i} L_p(\{\boldsymbol{x}_i^{\hat{t}_i}\}, \{\boldsymbol{y}_i^{\hat{t}_i}\}, \boldsymbol{v}^{\hat{t}_i}, \boldsymbol{z}^{\hat{t}_i}, \{\lambda_l^{\hat{t}_i}\}, \{\boldsymbol{\theta}_i^{\hat{t}_i}\}) \right\rangle
$$
$$
\leq \tfrac{1}{2}||\boldsymbol{x}_i^{t+1}-\boldsymbol{x}_i^t||^2 + \tfrac{L^2}{2}(||\boldsymbol{v}^t-\boldsymbol{v}^{\hat{t}_j}||^2 + ||\boldsymbol{z}^t-\boldsymbol{z}^{\hat{t}_j}||^2 + \sum_{l=1}^{|\mathcal{P}^t|} ||\lambda_l^t-\lambda_l^{\hat{t}_j}||^2)
$$
$$
\leq \tfrac{1}{2}||\boldsymbol{x}_i^{t+1}-\boldsymbol{x}_i^t||^2 + \tfrac{3L^2\tau k_1}{2}(||\boldsymbol{v}^{t+1}-\boldsymbol{v}^t||^2 + ||\boldsymbol{z}^{t+1}-\boldsymbol{z}^t||^2 + \sum_{l=1}^{|\mathcal{P}^t|} ||\lambda_l^{t+1}-\lambda_l^t||^2).
$$
(69)

Thus, according to Eq. (67), (68) and (69), we can obtain that,

$$
L_p(\{\boldsymbol{x}_i^{t+1}\},\{\boldsymbol{y}_i^t\},\boldsymbol{v}^t,\boldsymbol{z}^t,\{\lambda_l^t\},\{\boldsymbol{\theta}_i^t\}) - L_p(\{\boldsymbol{x}_i^t\},\{\boldsymbol{y}_i^t\},\boldsymbol{v}^t,\boldsymbol{z}^t,\{\lambda_l^t\},\{\boldsymbol{\theta}_i^t\})
$$
$$
\leq \sum_{i=1}^N (\tfrac{L+1}{2} - \tfrac{1}{\eta_{\boldsymbol{x}}^t})||\boldsymbol{x}_i^{t+1}-\boldsymbol{x}_i^t||^2 + \tfrac{3NL^2\tau k_1}{2}(||\boldsymbol{v}^{t+1}-\boldsymbol{v}^t||^2 + ||\boldsymbol{z}^{t+1}-\boldsymbol{z}^t||^2 + \sum_{l=1}^{|\mathcal{P}^t|} ||\lambda_l^{t+1}-\lambda_l^t||^2).
$$
(70)

Similarly, using the Lipschitz properties in Assumption 1, we have,

$$
\begin{aligned}
&L_p(\{\boldsymbol{x}_i^{t+1}\},\{\boldsymbol{y}_i^{t+1}\},\boldsymbol{v}^t,\boldsymbol{z}^t,\{\lambda_l^t\},\{\boldsymbol{\theta}_i^t\}) - L_p(\{\boldsymbol{x}_i^{t+1}\},\{\boldsymbol{y}_i^t\},\boldsymbol{v}^t,\boldsymbol{z}^t,\{\lambda_l^t\},\{\boldsymbol{\theta}_i^t\}) \\
&\leq \sum_{i=1}^N \left( \langle \nabla_{\boldsymbol{y}_i} L_p(\{\boldsymbol{x}_i^{t+1}\},\{\boldsymbol{y}_i^t\},\boldsymbol{v}^t,\boldsymbol{z}^t,\{\lambda_l^t\},\{\boldsymbol{\theta}_i^t\}), \boldsymbol{y}_i^{t+1}-\boldsymbol{y}_i^t \rangle + \tfrac{L}{2}\|\boldsymbol{y}_i^{t+1}-\boldsymbol{y}_i^t\|^2 \right).
\end{aligned}
\tag{71}
$$

Combining $\nabla_{\boldsymbol{y}_i} L_p(\{\boldsymbol{x}_i^{\hat{t}_i}\},\{\boldsymbol{y}_i^{\hat{t}_i}\},\boldsymbol{v}^{\hat{t}_i},\boldsymbol{z}^{\hat{t}_i},\{\lambda_l^{\hat{t}_i}\},\{\boldsymbol{\theta}_i^{\hat{t}_i}\}) = \nabla_{\boldsymbol{y}_i}\widetilde{L}_p(\{\boldsymbol{x}_i^{\hat{t}_i}\},\{\boldsymbol{y}_i^{\hat{t}_i}\},\boldsymbol{v}^{\hat{t}_i},\boldsymbol{z}^{\hat{t}_i},\{\lambda_l^{\hat{t}_i}\},\{\boldsymbol{\theta}_i^{\hat{t}_i}\})$ with Eq. (16), we can obtain that,

$$
\left\langle \boldsymbol{y}_i^{t+1}-\boldsymbol{y}_i^t, \nabla_{\boldsymbol{y}_i} L_p(\{\boldsymbol{x}_i^{\hat{t}_i}\},\{\boldsymbol{y}_i^{\hat{t}_i}\},\boldsymbol{v}^{\hat{t}_i},\boldsymbol{z}^{\hat{t}_i},\{\lambda_l^{\hat{t}_i}\},\{\boldsymbol{\theta}_i^{\hat{t}_i}\}) \right\rangle = -\frac{1}{\eta_{\boldsymbol{y}}}\|\boldsymbol{y}_i^{t+1}-\boldsymbol{y}_i^t\|^2 \leq -\frac{1}{\eta_{\boldsymbol{y}}^t}\|\boldsymbol{y}_i^{t+1}-\boldsymbol{y}_i^t\|^2.
\tag{72}
$$

Then, combining the Cauchy-Schwarz inequality with Assumption 1, 2, we can get the following inequalities,

$$
\begin{aligned}
&\left\langle \boldsymbol{y}_i^{t+1}-\boldsymbol{y}_i^t, \nabla_{\boldsymbol{y}_i} L_p(\{\boldsymbol{x}_i^{t+1}\},\{\boldsymbol{y}_i^t\},\boldsymbol{v}^t,\boldsymbol{z}^t,\{\lambda_l^t\},\{\boldsymbol{\theta}_i^t\}) - \nabla_{\boldsymbol{y}_i} L_p(\{\boldsymbol{x}_i^{\hat{t}_i}\},\{\boldsymbol{y}_i^{\hat{t}_i}\},\boldsymbol{v}^{\hat{t}_i},\boldsymbol{z}^{\hat{t}_i},\{\lambda_l^{\hat{t}_i}\},\{\boldsymbol{\theta}_i^{\hat{t}_i}\}) \right\rangle \\
&\leq \tfrac{1}{2}\|\boldsymbol{y}_i^{t+1}-\boldsymbol{y}_i^t\|^2 + \tfrac{L^2}{2}(\|\boldsymbol{x}_i^{t+1}-\boldsymbol{x}_i^t\|^2 + \|\boldsymbol{v}^t-\boldsymbol{v}^{\hat{t}_j}\|^2 + \|\boldsymbol{z}^t-\boldsymbol{z}^{\hat{t}_j}\|^2 + \sum_{l=1}^{|\mathcal{P}^t|}\|\lambda_l^t-\lambda_l^{\hat{t}_j}\|^2) \\
&\leq \tfrac{1}{2}\|\boldsymbol{y}_i^{t+1}-\boldsymbol{y}_i^t\|^2 + \tfrac{L^2}{2}\|\boldsymbol{x}_i^{t+1}-\boldsymbol{x}_i^t\|^2 + \tfrac{3L^2\tau k_1}{2}(\|\boldsymbol{v}^{t+1}-\boldsymbol{v}^t\|^2 + \|\boldsymbol{z}^{t+1}-\boldsymbol{z}^t\|^2 + \sum_{l=1}^{|\mathcal{P}^t|}\|\lambda_l^{t+1}-\lambda_l^t\|^2).
\end{aligned}
\tag{73}
$$

Thus, combining Eq. (71), (72) with (73), we have,

$$
\begin{aligned}
&L_p(\{\boldsymbol{x}_i^{t+1}\},\{\boldsymbol{y}_i^{t+1}\},\boldsymbol{v}^t,\boldsymbol{z}^t,\{\lambda_l^t\},\{\boldsymbol{\theta}_i^t\}) - L_p(\{\boldsymbol{x}_i^{t+1}\},\{\boldsymbol{y}_i^t\},\boldsymbol{v}^t,\boldsymbol{z}^t,\{\lambda_l^t\},\{\boldsymbol{\theta}_i^t\}) \\
&\leq \sum_{i=1}^N (\tfrac{L+1}{2}-\tfrac{1}{\eta_{\boldsymbol{y}}^t})\|\boldsymbol{y}_i^{t+1}-\boldsymbol{y}_i^t\|^2 + \sum_{i=1}^N \tfrac{L^2}{2}\|\boldsymbol{x}_i^{t+1}-\boldsymbol{x}_i^t\|^2 \\
&\quad + \tfrac{3NL^2\tau k_1}{2}(\|\boldsymbol{v}^{t+1}-\boldsymbol{v}^t\|^2 + \|\boldsymbol{z}^{t+1}-\boldsymbol{z}^t\|^2 + \sum_{l=1}^{|\mathcal{P}^t|}\|\lambda_l^{t+1}-\lambda_l^t\|^2).
\end{aligned}
\tag{74}
$$

Combining the Lipschitz properties in Assumption 1 with Eq. (17), we have,

$$
\begin{aligned}
&L_p(\{\boldsymbol{x}_i^{t+1}\},\{\boldsymbol{y}_i^{t+1}\},\boldsymbol{v}^{t+1},\boldsymbol{z}^t,\{\lambda_l^t\},\{\boldsymbol{\theta}_i^t\}) - L_p(\{\boldsymbol{x}_i^{t+1}\},\{\boldsymbol{y}_i^{t+1}\},\boldsymbol{v}^t,\boldsymbol{z}^t,\{\lambda_l^t\},\{\boldsymbol{\theta}_i^t\}) \\
&\leq \langle \nabla_{\boldsymbol{v}} L_p(\{\boldsymbol{x}_i^{t+1}\},\{\boldsymbol{y}_i^{t+1}\},\boldsymbol{v}^t,\boldsymbol{z}^t,\{\lambda_l^t\},\{\boldsymbol{\theta}_i^t\}), \boldsymbol{v}^{t+1}-\boldsymbol{v}^t \rangle + \tfrac{L}{2}\|\boldsymbol{v}^{t+1}-\boldsymbol{v}^t\|^2 \\
&\leq (\tfrac{L}{2}-\tfrac{1}{\eta_{\boldsymbol{v}}^t})\|\boldsymbol{v}^{t+1}-\boldsymbol{v}^t\|^2.
\end{aligned}
\tag{75}
$$

Similarly, combining the Lipschitz properties in Assumption 1 with Eq. (18), we have,

$$
\begin{aligned}
&L_p(\{\boldsymbol{x}_i^{t+1}\},\{\boldsymbol{y}_i^{t+1}\},\boldsymbol{v}^{t+1},\boldsymbol{z}^{t+1},\{\lambda_l^t\},\{\boldsymbol{\theta}_i^t\}) - L_p(\{\boldsymbol{x}_i^{t+1}\},\{\boldsymbol{y}_i^{t+1}\},\boldsymbol{v}^{t+1},\boldsymbol{z}^t,\{\lambda_l^t\},\{\boldsymbol{\theta}_i^t\}) \\
&\leq \langle \nabla_{\boldsymbol{z}} L_p(\{\boldsymbol{x}_i^{t+1}\},\{\boldsymbol{y}_i^{t+1}\},\boldsymbol{v}^{t+1},\boldsymbol{z}^t,\{\lambda_l^t\},\{\boldsymbol{\theta}_i^t\}), \boldsymbol{z}^{t+1}-\boldsymbol{z}^t \rangle + \tfrac{L}{2}\|\boldsymbol{z}^{t+1}-\boldsymbol{z}^t\|^2 \\
&\leq (\tfrac{L}{2}-\tfrac{1}{\eta_{\boldsymbol{z}}^t})\|\boldsymbol{z}^{t+1}-\boldsymbol{z}^t\|^2.
\end{aligned}
\tag{76}
$$

By combining Eq. (70), (74), (75), (76), we conclude the proof of Lemma 1.

**Lemma 2** *Suppose Assumption 1 and 2 hold, $\forall t \geq T_1$, we have:*

$$L_p(\{\boldsymbol{x}_i^{t+1}\},\{\boldsymbol{y}_i^{t+1}\},\boldsymbol{v}^{t+1},\boldsymbol{z}^{t+1},\{\lambda_l^{t+1}\},\{\boldsymbol{\theta}_i^{t+1}\}) - L_p(\{\boldsymbol{x}_i^t\},\{\boldsymbol{y}_i^t\},\boldsymbol{v}^t,\boldsymbol{z}^t,\{\lambda_l^t\},\{\boldsymbol{\theta}_i^t\})$$

$$\leq \left(\frac{L+L^2+1}{2} - \frac{1}{\eta_x^t} + \frac{|\boldsymbol{\mathcal{P}}^t|L^2}{2a_1} + \frac{|\boldsymbol{\mathcal{Q}}^{t+1}|L^2}{2a_3}\right)\sum_{i=1}^N \|\boldsymbol{x}_i^{t+1} - \boldsymbol{x}_i^t\|^2$$

$$+ \left(\frac{L+1}{2} - \frac{1}{\eta_y^t} + \frac{|\boldsymbol{\mathcal{P}}^t|L^2}{2a_1} + \frac{|\boldsymbol{\mathcal{Q}}^{t+1}|L^2}{2a_3}\right)\sum_{i=1}^N \|\boldsymbol{y}_i^{t+1} - \boldsymbol{y}_i^t\|^2$$

$$+ \left(\frac{L+6\tau k_1 NL^2}{2} - \frac{1}{\eta_v^t} + \frac{|\boldsymbol{\mathcal{P}}^t|L^2}{2a_1} + \frac{|\boldsymbol{\mathcal{Q}}^{t+1}|L^2}{2a_3}\right)\|\boldsymbol{v}^{t+1} - \boldsymbol{v}^t\|^2$$

$$+ \left(\frac{L+6\tau k_1 NL^2}{2} - \frac{1}{\eta_z^t} + \frac{|\boldsymbol{\mathcal{P}}^t|L^2}{2a_1} + \frac{|\boldsymbol{\mathcal{Q}}^{t+1}|L^2}{2a_3}\right)\|\boldsymbol{z}^{t+1} - \boldsymbol{z}^t\|^2 + \frac{1}{2\eta_\theta}\sum_{i=1}^N \|\boldsymbol{\theta}_i^t - \boldsymbol{\theta}_i^{t-1}\|^2$$

$$+ \left(\frac{a_1+6\tau k_1 NL^2}{2} - \frac{c_1^{t-1}-c_1^t}{2} + \frac{1}{2\eta_\lambda}\right)\sum_{l=1}^{|\boldsymbol{\mathcal{P}}^t|} \|\lambda_l^{t+1} - \lambda_l^t\|^2 + \left(\frac{a_3}{2} - \frac{c_2^{t-1}-c_2^t}{2} + \frac{1}{2\eta_\theta}\right)\sum_{i=1}^N \|\boldsymbol{\theta}_i^{t+1} - \boldsymbol{\theta}_i^t\|^2$$

$$+ \frac{c_1^{t-1}}{2}\sum_{l=1}^{|\boldsymbol{\mathcal{P}}^t|}(\|\lambda_l^{t+1}\|^2 - \|\lambda_l^t\|^2) + \frac{1}{2\eta_\lambda}\sum_{l=1}^{|\boldsymbol{\mathcal{P}}^t|}\|\lambda_l^t - \lambda_l^{t-1}\|^2 + \frac{c_2^{t-1}}{2}\sum_{i=1}^N(\|\boldsymbol{\theta}_i^{t+1}\|^2 - \|\boldsymbol{\theta}_i^t\|^2),$$

$$(77)$$

*where $a_1 > 0$ and $a_3 > 0$ are constants.*

*Proof of Lemma 2:*

According to Eq. (19), in $(t+1)^{\text{th}}$ iteration, $\forall \lambda \in \boldsymbol{\Lambda}$, it follows that:

$$\left\langle \lambda_l^{t+1} - \lambda_l^t - \eta_\lambda \nabla_{\lambda_l}\widetilde{L}_p(\{\boldsymbol{x}_i^{t+1}\},\{\boldsymbol{y}_i^{t+1}\},\boldsymbol{v}^{t+1},\boldsymbol{z}^{t+1},\{\lambda_l^t\},\{\boldsymbol{\theta}_i^t\}), \lambda - \lambda_l^{t+1}\right\rangle = 0. \qquad (78)$$

Let $\lambda = \lambda_l^t$, we can obtain:

$$\left\langle \nabla_{\lambda_l}\widetilde{L}_p(\{\boldsymbol{x}_i^{t+1}\},\{\boldsymbol{y}_i^{t+1}\},\boldsymbol{v}^{t+1},\boldsymbol{z}^{t+1},\{\lambda_l^t\},\{\boldsymbol{\theta}_i^t\}) - \frac{1}{\eta_\lambda}(\lambda_l^{t+1} - \lambda_l^t), \lambda_l^t - \lambda_l^{t+1}\right\rangle = 0. \qquad (79)$$

Likewise, in $t^{\text{th}}$ iteration, we can obtain:

$$\left\langle \nabla_{\lambda_l}\widetilde{L}_p(\{\boldsymbol{x}_i^t\},\{\boldsymbol{y}_i^t\},\boldsymbol{v}^t,\boldsymbol{z}^t,\{\lambda_l^{t-1}\},\{\boldsymbol{\theta}_i^{t-1}\}) - \frac{1}{\eta_\lambda}(\lambda_l^t - \lambda_l^{t-1}), \lambda_l^{t+1} - \lambda_l^t\right\rangle = 0. \qquad (80)$$

Since $\widetilde{L}_p(\{\boldsymbol{x}_i\},\{\boldsymbol{y}_i\},\boldsymbol{v},\boldsymbol{z},\{\lambda_l\},\{\boldsymbol{\theta}_i\})$ is concave with respect to $\lambda_l$ and follows from Eq. (79) and Eq. (80), $\forall t \geq T_1$, we have,

$$\widetilde{L}_p(\{\boldsymbol{x}_i^{t+1}\},\{\boldsymbol{y}_i^{t+1}\},\boldsymbol{v}^{t+1},\boldsymbol{z}^{t+1},\{\lambda_l^{t+1}\},\{\boldsymbol{\theta}_i^t\}) - \widetilde{L}_p(\{\boldsymbol{x}_i^{t+1}\},\{\boldsymbol{y}_i^{t+1}\},\boldsymbol{v}^{t+1},\boldsymbol{z}^{t+1},\{\lambda_l^t\},\{\boldsymbol{\theta}_i^t\})$$

$$\leq \sum_{l=1}^{|\boldsymbol{\mathcal{P}}^t|}\left\langle \nabla_{\lambda_l}\widetilde{L}_p(\{\boldsymbol{x}_i^{t+1}\},\{\boldsymbol{y}_i^{t+1}\},\boldsymbol{v}^{t+1},\boldsymbol{z}^{t+1},\{\lambda_l^t\},\{\boldsymbol{\theta}_i^t\}), \lambda_l^{t+1} - \lambda_l^t\right\rangle$$

$$\leq \sum_{l=1}^{|\boldsymbol{\mathcal{P}}^t|}\left(\left\langle \nabla_{\lambda_l}\widetilde{L}_p(\{\boldsymbol{x}_i^{t+1}\},\{\boldsymbol{y}_i^{t+1}\},\boldsymbol{v}^{t+1},\boldsymbol{z}^{t+1},\{\lambda_l^t\},\{\boldsymbol{\theta}_i^t\}) - \nabla_{\lambda_l}\widetilde{L}_p(\{\boldsymbol{x}_i^t\},\{\boldsymbol{y}_i^t\},\boldsymbol{v}^t,\boldsymbol{z}^t,\{\lambda_l^{t-1}\},\{\boldsymbol{\theta}_i^{t-1}\}), \lambda_l^{t+1} - \lambda_l^t\right\rangle\right.$$

$$\left. + \frac{1}{\eta_\lambda}\left\langle \lambda_l^t - \lambda_l^{t-1}, \lambda_l^{t+1} - \lambda_l^t\right\rangle\right).$$

$$(81)$$

Denoting $\boldsymbol{v}_{1,l}^{t+1} = \lambda_l^{t+1} - \lambda_l^t - (\lambda_l^t - \lambda_l^{t-1})$, we can get the following equality,

$$\sum_{l=1}^{|\boldsymbol{\mathcal{P}}^t|}\left\langle \nabla_{\lambda_l}\widetilde{L}_p(\{\boldsymbol{x}_i^{t+1}\},\{\boldsymbol{y}_i^{t+1}\},\boldsymbol{v}^{t+1},\boldsymbol{z}^{t+1},\{\lambda_l^t\},\{\boldsymbol{\theta}_i^t\}) - \nabla_{\lambda_l}\widetilde{L}_p(\{\boldsymbol{x}_i^t\},\{\boldsymbol{y}_i^t\},\boldsymbol{v}^t,\boldsymbol{z}^t,\{\lambda_l^{t-1}\},\{\boldsymbol{\theta}_i^{t-1}\}), \lambda_l^{t+1} - \lambda_l^t\right\rangle$$

$$= \sum_{l=1}^{|\boldsymbol{\mathcal{P}}^t|}\left\langle \nabla_{\lambda_l}\widetilde{L}_p(\{\boldsymbol{x}_i^{t+1}\},\{\boldsymbol{y}_i^{t+1}\},\boldsymbol{v}^{t+1},\boldsymbol{z}^{t+1},\{\lambda_l^t\},\{\boldsymbol{\theta}_i^t\}) - \nabla_{\lambda_l}\widetilde{L}_p(\{\boldsymbol{x}_i^t\},\{\boldsymbol{y}_i^t\},\boldsymbol{v}^t,\boldsymbol{z}^t,\{\lambda_l^t\},\{\boldsymbol{\theta}_i^t\})), \lambda_l^{t+1} - \lambda_l^t\right\rangle (1a)$$

$$+ \sum_{l=1}^{|\boldsymbol{\mathcal{P}}^t|}\left\langle \nabla_{\lambda_l}\widetilde{L}_p(\{\boldsymbol{x}_i^t\},\{\boldsymbol{y}_i^t\},\boldsymbol{v}^t,\boldsymbol{z}^t,\{\lambda_l^t\},\{\boldsymbol{\theta}_i^t\}) - \nabla_{\lambda_l}\widetilde{L}_p(\{\boldsymbol{x}_i^t\},\{\boldsymbol{y}_i^t\},\boldsymbol{v}^t,\boldsymbol{z}^t,\{\lambda_l^{t-1}\},\{\boldsymbol{\theta}_i^{t-1}\}), \boldsymbol{v}_{1,l}^{t+1}\right\rangle (1b)$$

$$+ \sum_{l=1}^{|\boldsymbol{\mathcal{P}}^t|}\left\langle \nabla_{\lambda_l}\widetilde{L}_p(\{\boldsymbol{x}_i^t\},\{\boldsymbol{y}_i^t\},\boldsymbol{v}^t,\boldsymbol{z}^t,\{\lambda_l^t\},\{\boldsymbol{\theta}_i^t\}) - \nabla_{\lambda_l}\widetilde{L}_p(\{\boldsymbol{x}_i^t\},\{\boldsymbol{y}_i^t\},\boldsymbol{v}^t,\boldsymbol{z}^t,\{\lambda_l^{t-1}\},\{\boldsymbol{\theta}_i^{t-1}\}), \lambda_l^t - \lambda_l^{t-1}\right\rangle (1c).$$

$$(82)$$

First, we put attention on the $(1a)$ in Eq. (82), $(1a)$ can be expressed as follows,

$$
\left\langle \nabla_{\lambda_l} \widetilde{L}_p(\{\boldsymbol{x}_i^{t+1}\},\{\boldsymbol{y}_i^{t+1}\},\boldsymbol{v}^{t+1},\boldsymbol{z}^{t+1},\{\lambda_l^t\},\{\boldsymbol{\theta}_i^t\}) - \nabla_{\lambda_l} \widetilde{L}_p(\{\boldsymbol{x}_i^t\},\{\boldsymbol{y}_i^t\},\boldsymbol{v}^t,\boldsymbol{z}^t,\{\lambda_l^t\},\{\boldsymbol{\theta}_i^t\}), \lambda_l^{t+1} - \lambda_l^t \right\rangle
$$
$$
= \left\langle \nabla_{\lambda_l} L_p(\{\boldsymbol{x}_i^{t+1}\},\{\boldsymbol{y}_i^{t+1}\},\boldsymbol{v}^{t+1},\boldsymbol{z}^{t+1},\{\lambda_l^t\},\{\boldsymbol{\theta}_i^t\}) - \nabla_{\lambda_l} L_p(\{\boldsymbol{x}_i^t\},\{\boldsymbol{y}_i^t\},\boldsymbol{v}^t,\boldsymbol{z}^t,\{\lambda_l^t\},\{\boldsymbol{\theta}_i^t\}), \lambda_l^{t+1} - \lambda_l^t \right\rangle
$$
$$
+ \frac{c_1^{t-1} - c_1^t}{2}(||\lambda_l^{t+1}||^2 - ||\lambda_l^t||^2) - \frac{c_1^{t-1} - c_1^t}{2}||\lambda_l^{t+1} - \lambda_l^t||^2.
\tag{83}
$$

Combining Cauchy-Schwarz inequality with Assumption 1, we can obtain,

$$
\left\langle \nabla_{\lambda_l} L_p(\{\boldsymbol{x}_i^{t+1}\},\{\boldsymbol{y}_i^{t+1}\},\boldsymbol{v}^{t+1},\boldsymbol{z}^{t+1},\{\lambda_l^t\},\{\boldsymbol{\theta}_i^t\}) - \nabla_{\lambda_l} L_p(\{\boldsymbol{x}_i^t\},\{\boldsymbol{y}_i^t\},\boldsymbol{v}^t,\boldsymbol{z}^t,\{\lambda_l^t\},\{\boldsymbol{\theta}_i^t\}), \lambda_l^{t+1} - \lambda_l^t \right\rangle
$$
$$
\leq \frac{L^2}{2a_1}(\sum_{i=1}^{N}(||\boldsymbol{x}_i^{t+1} - \boldsymbol{x}_i^t||^2 + ||\boldsymbol{y}_i^{t+1} - \boldsymbol{y}_i^t||^2) + ||\boldsymbol{v}^{t+1} - \boldsymbol{v}^t||^2 + ||\boldsymbol{z}^{t+1} - \boldsymbol{z}^t||^2) + \frac{a_1}{2}||\lambda_l^{t+1} - \lambda_l^t||^2,
\tag{84}
$$

where $a_1 > 0$ is a constant. Combining Eq. (83) with Eq. (84), we can obtain that,

$$
\sum_{l=1}^{|\mathcal{P}^t|} \left\langle \nabla_{\lambda_l} \widetilde{L}_p(\{\boldsymbol{x}_i^{t+1}\},\{\boldsymbol{y}_i^{t+1}\},\boldsymbol{v}^{t+1},\boldsymbol{z}^{t+1},\{\lambda_l^t\},\{\boldsymbol{\theta}_i^t\}) - \nabla_{\lambda_l} \widetilde{L}_p(\{\boldsymbol{x}_i^t\},\{\boldsymbol{y}_i^t\},\boldsymbol{v}^t,\boldsymbol{z}^t,\{\lambda_l^t\},\{\boldsymbol{\theta}_i^t\}), \lambda_l^{t+1} - \lambda_l^t \right\rangle
$$
$$
\leq \sum_{l=1}^{|\mathcal{P}^t|}(\frac{L^2}{2a_1}(\sum_{i=1}^{N}(||\boldsymbol{x}_i^{t+1} - \boldsymbol{x}_i^t||^2 + ||\boldsymbol{y}_i^{t+1} - \boldsymbol{y}_i^t||^2) + ||\boldsymbol{v}^{t+1} - \boldsymbol{v}^t||^2 + ||\boldsymbol{z}^{t+1} - \boldsymbol{z}^t||^2) + \frac{a_1}{2}||\lambda_l^{t+1} - \lambda_l^t||^2
$$
$$
+ \frac{c_1^{t-1} - c_1^t}{2}(||\lambda_l^{t+1}||^2 - ||\lambda_l^t||^2) - \frac{c_1^{t-1} - c_1^t}{2}||\lambda_l^{t+1} - \lambda_l^t||^2).
\tag{85}
$$

Then, we focus on the $(1b)$ in Eq. (82). According to Cauchy-Schwarz inequality, $(1b)$ can be expressed as follows,

$$
\sum_{l=1}^{|\mathcal{P}^t|} \left\langle \nabla_{\lambda_l} \widetilde{L}_p(\{\boldsymbol{x}_i^t\},\{\boldsymbol{y}_i^t\},\boldsymbol{v}^t,\boldsymbol{z}^t,\{\lambda_l^t\},\{\boldsymbol{\theta}_i^t\}) - \nabla_{\lambda_l} \widetilde{L}_p(\{\boldsymbol{x}_i^t\},\{\boldsymbol{y}_i^t\},\boldsymbol{v}^t,\boldsymbol{z}^t,\{\lambda_l^{t-1}\},\{\boldsymbol{\theta}_i^{t-1}\}), \boldsymbol{v}_{1,l}^{t+1} \right\rangle
$$
$$
\leq \sum_{l=1}^{|\mathcal{P}^t|}(\frac{a_2}{2}||\nabla_{\lambda_l} \widetilde{L}_p(\{\boldsymbol{x}_i^t\},\{\boldsymbol{y}_i^t\},\boldsymbol{v}^t,\boldsymbol{z}^t,\{\lambda_l^t\},\{\boldsymbol{\theta}_i^t\}) - \nabla_{\lambda_l} \widetilde{L}_p(\{\boldsymbol{x}_i^t\},\{\boldsymbol{y}_i^t\},\boldsymbol{v}^t,\boldsymbol{z}^t,\{\lambda_l^{t-1}\},\{\boldsymbol{\theta}_i^{t-1}\})||^2
$$
$$
+ \frac{1}{2a_2}||\boldsymbol{v}_{1,l}^{t+1}||^2),
\tag{86}
$$

where $a_2 > 0$ is a constant. Next, we focus on the $(1c)$ in Eq. (82). Defining $L_1' = L + c_1^0$, according to Assumption 1 and the trigonometric inequality, $\forall \lambda_l$, we have,

$$
||\nabla_{\lambda_l} \widetilde{L}_p(\{\boldsymbol{x}_i^t\},\{\boldsymbol{y}_i^t\},\boldsymbol{v}^t,\boldsymbol{z}^t,\{\lambda_l^t\},\{\boldsymbol{\theta}_i^t\}) - \nabla_{\lambda_l} \widetilde{L}_p(\{\boldsymbol{x}_i^t\},\{\boldsymbol{y}_i^t\},\boldsymbol{v}^t,\boldsymbol{z}^t,\{\lambda_l^{t-1}\},\{\boldsymbol{\theta}_i^{t-1}\})||
$$
$$
= ||\nabla_{\lambda_l} L_p(\{\boldsymbol{x}_i^t\},\{\boldsymbol{y}_i^t\},\boldsymbol{v}^t,\boldsymbol{z}^t,\{\lambda_l^t\},\{\boldsymbol{\theta}_i^t\}) - \nabla_{\lambda_l} L_p(\{\boldsymbol{x}_i^t\},\{\boldsymbol{y}_i^t\},\boldsymbol{v}^t,\boldsymbol{z}^t,\{\lambda_l^{t-1}\},\{\boldsymbol{\theta}_i^t\}) - c_1^{t-1}(\lambda_l^t - \lambda_l^{t-1})||
$$
$$
\leq (L + c_1^{t-1})||\lambda_l^t - \lambda_l^{t-1}||
$$
$$
\leq L_1'||\lambda_l^t - \lambda_l^{t-1}||.
\tag{87}
$$

Following from Eq. (87) and the strong concavity of $\widetilde{L}_p(\{\boldsymbol{x}_i\},\{\boldsymbol{y}_i\},\boldsymbol{v},\boldsymbol{z},\{\lambda_l\},\{\boldsymbol{\theta}_i\})$ *w.r.t* $\lambda_l$ (Nesterov, 2003; Xu et al., 2020), we can obtain that,

$$
\sum_{l=1}^{|\mathcal{P}^t|} \left\langle \nabla_{\lambda_l} \widetilde{L}_p(\{\boldsymbol{x}_i^t\},\{\boldsymbol{y}_i^t\},\boldsymbol{v}^t,\boldsymbol{z}^t,\{\lambda_l^t\},\{\boldsymbol{\theta}_i^t\}) - \nabla_{\lambda_l} \widetilde{L}_p(\{\boldsymbol{x}_i^t\},\{\boldsymbol{y}_i^t\},\boldsymbol{v}^t,\boldsymbol{z}^t,\{\lambda_l^{t-1}\},\{\boldsymbol{\theta}_i^{t-1}\}) \right\rangle
$$
$$
\leq \sum_{l=1}^{|\mathcal{P}^t|}(-\frac{1}{L_1' + c_1^{t-1}}||\nabla_{\lambda_l} \widetilde{L}_p(\{\boldsymbol{x}_i^t\},\{\boldsymbol{y}_i^t\},\boldsymbol{v}^t,\boldsymbol{z}^t,\{\lambda_l^t\},\{\boldsymbol{\theta}_i^t\}) - \nabla_{\lambda_l} \widetilde{L}_p(\{\boldsymbol{x}_i^t\},\{\boldsymbol{y}_i^t\},\boldsymbol{v}^t,\boldsymbol{z}^t,\{\lambda_l^{t-1}\},\{\boldsymbol{\theta}_i^{t-1}\})||^2
$$
$$
- \frac{c_1^{t-1} L_1'}{L_1' + c_1^{t-1}}||\lambda_l^t - \lambda_l^{t-1}||^2).
\tag{88}
$$

In addition, the following inequality can be obtained,

$$
\frac{1}{\eta_\lambda} \left\langle \lambda_l^t - \lambda_l^{t-1}, \lambda_l^{t+1} - \lambda_l^t \right\rangle \leq \frac{1}{2\eta_\lambda}||\lambda_l^{t+1} - \lambda_l^t||^2 - \frac{1}{2\eta_\lambda}||\boldsymbol{v}_{1,l}^{t+1}||^2 + \frac{1}{2\eta_\lambda}||\lambda_l^t - \lambda_l^{t-1}||^2.
\tag{89}
$$

Combining Eq. (81), (82), (85), (86), (88), (89), $\frac{\eta_\lambda}{2} \leq \frac{1}{L_1' + c_1^0}$, and setting $a_2 = \eta_\lambda$, we have:

$$
L_p(\{x_i^{t+1}\}, \{y_i^{t+1}\}, v^{t+1}, z^{t+1}, \{\lambda_l^{t+1}\}, \{\theta_i^t\}) - L_p(\{x_i^{t+1}\}, \{y_i^{t+1}\}, v^{t+1}, z^{t+1}, \{\lambda_l^t\}, \{\theta_i^t\})
$$

$$
\leq \frac{|\mathcal{P}^t| L^2}{2a_1} \Big( \sum_{i=1}^N (\|x_i^{t+1} - x_i^t\|^2 + \|y_i^{t+1} - y_i^t\|^2) + \|v^{t+1} - v^t\|^2 + \|z^{t+1} - z^t\|^2 \Big)
$$

$$
+ \Big( \frac{a_1}{2} - \frac{c_1^{t-1} - c_1^t}{2} + \frac{1}{2\eta_\lambda} \Big) \sum_{l=1}^{|\mathcal{P}^t|} \|\lambda_l^{t+1} - \lambda_l^t\|^2 + \frac{c_1^{t-1}}{2} \sum_{l=1}^{|\mathcal{P}^t|} (\|\lambda_l^{t+1}\|^2 - \|\lambda_l^t\|^2) + \frac{1}{2\eta_\lambda} \sum_{l=1}^{|\mathcal{P}^t|} \|\lambda_l^t - \lambda_l^{t-1}\|^2.
$$

$$(90)$$

According to Eq. (20), in $(t+1)^{\text{th}}$ iteration, $\forall \theta \in \Theta$, it follows that,

$$
\Big\langle \theta_i^{t+1} - \theta_i^t - \eta_\theta \nabla_{\theta_i} \widetilde{L}_p(\{x_i^{t+1}\}, \{y_i^{t+1}\}, v^{t+1}, z^{t+1}, \{\lambda_l^{t+1}\}, \{\theta_i^t\}), \theta - \theta_i^{t+1} \Big\rangle = 0. \quad (91)
$$

Choosing $\theta = \theta_i^t$, we can obtain,

$$
\Big\langle \nabla_{\theta_i} \widetilde{L}_p(\{x_i^{t+1}\}, \{y_i^{t+1}\}, v^{t+1}, z^{t+1}, \{\lambda_l^{t+1}\}, \{\theta_i^t\}) - \frac{1}{\eta_\theta}(\theta_i^{t+1} - \theta_i^t), \theta_i^t - \theta_i^{t+1} \Big\rangle = 0. \quad (92)
$$

Likewise, in $t^{\text{th}}$ iteration, we have,

$$
\Big\langle \nabla_{\theta_i} \widetilde{L}_p(\{x_i^t\}, \{y_i^t\}, v^t, z^t, \{\lambda_l^t\}, \{\theta_i^{t-1}\}) - \frac{1}{\eta_\theta}(\theta_i^t - \theta_i^{t-1}), \theta_i^{t+1} - \theta_i^t \Big\rangle = 0. \quad (93)
$$

Since $\widetilde{L}_p(\{x_i\}, \{y_i\}, v, z, \{\lambda_l\}, \{\theta_i\})$ is concave with respect to $\theta_i$ and follows from Eq. (93):

$$
\widetilde{L}_p(\{x_i^{t+1}\}, \{y_i^{t+1}\}, v^{t+1}, z^{t+1}, \{\lambda_l^{t+1}\}, \{\theta_i^{t+1}\}) - \widetilde{L}_p(\{x_i^{t+1}\}, \{y_i^{t+1}\}, v^{t+1}, z^{t+1}, \{\lambda_l^{t+1}\}, \{\theta_i^t\})
$$

$$
\leq \sum_{i=1}^N \Big\langle \nabla_{\theta_i} \widetilde{L}_p(\{x_i^{t+1}\}, \{y_i^{t+1}\}, v^{t+1}, z^{t+1}, \{\lambda_l^{t+1}\}, \{\theta_i^t\}), \theta_i^{t+1} - \theta_i^t \Big\rangle
$$

$$
\leq \sum_{i=1}^N \Big( \Big\langle \nabla_{\theta_i} \widetilde{L}_p(\{x_i^{t+1}\}, \{y_i^{t+1}\}, v^{t+1}, z^{t+1}, \{\lambda_l^{t+1}\}, \{\theta_i^t\}) - \nabla_{\theta_i} \widetilde{L}_p(\{x_i^t\}, \{y_i^t\}, v^t, z^t, \{\lambda_l^t\}, \{\theta_i^{t-1}\}), \theta_i^{t+1} - \theta_i^t \Big\rangle
$$

$$
+ \frac{1}{\eta_\theta} \Big\langle \theta_i^t - \theta_i^{t-1}, \theta_i^{t+1} - \theta_i^t \Big\rangle \Big).
$$

$$(94)$$

Denoting $v_{2,l}^{t+1} = \theta_i^{t+1} - \theta_i^t - (\theta_i^t - \theta_i^{t-1})$, we have that,

$$
\sum_{i=1}^N \Big\langle \nabla_{\theta_i} \widetilde{L}_p(\{x_i^{t+1}\}, \{y_i^{t+1}\}, v^{t+1}, z^{t+1}, \{\lambda_l^{t+1}\}, \{\theta_i^t\}) - \nabla_{\theta_i} \widetilde{L}_p(\{x_i^t\}, \{y_i^t\}, v^t, z^t, \{\lambda_l^t\}, \{\theta_i^{t-1}\}), \theta_i^{t+1} - \theta_i^t \Big\rangle
$$

$$
= \sum_{i=1}^N \Big\langle \nabla_{\theta_i} \widetilde{L}_p(\{x_i^{t+1}\}, \{y_i^{t+1}\}, v^{t+1}, z^{t+1}, \{\lambda_l^{t+1}\}, \{\theta_i^t\}) - \nabla_{\theta_i} \widetilde{L}_p(\{x_i^t\}, \{y_i^t\}, v^t, z^t, \{\lambda_l^t\}, \{\theta_i^t\}), \theta_i^{t+1} - \theta_i^t \Big\rangle (2a)
$$

$$
+ \sum_{i=1}^N \Big\langle \nabla_{\theta_i} \widetilde{L}_p(\{x_i^t\}, \{y_i^t\}, v^t, z^t, \{\lambda_l^t\}, \{\theta_i^t\}) - \nabla_{\theta_i} \widetilde{L}_p(\{x_i^t\}, \{y_i^t\}, v^t, z^t, \{\lambda_l^t\}, \{\theta_i^{t-1}\}), v_{2,l}^{t+1} \Big\rangle (2b)
$$

$$
+ \sum_{i=1}^N \Big\langle \nabla_{\theta_i} \widetilde{L}_p(\{x_i^t\}, \{y_i^t\}, v^t, z^t, \{\lambda_l^t\}, \{\theta_i^t\}) - \nabla_{\theta_i} \widetilde{L}_p(\{x_i^t\}, \{y_i^t\}, v^t, z^t, \{\lambda_l^t\}, \{\theta_i^{t-1}\}), \theta_i^t - \theta_i^{t-1} \Big\rangle (2c).
$$

$$(95)$$

We firstly focus on the $(2a)$ in Eq. (95), we can write the $(2a)$ as,

$$
\Big\langle \nabla_{\theta_i} \widetilde{L}_p(\{x_i^{t+1}\}, \{y_i^{t+1}\}, v^{t+1}, z^{t+1}, \{\lambda_l^{t+1}\}, \{\theta_i^t\}) - \nabla_{\theta_i} \widetilde{L}_p(\{x_i^t\}, \{y_i^t\}, v^t, z^t, \{\lambda_l^t\}, \{\theta_i^t\}), \theta_i^{t+1} - \theta_i^t \Big\rangle
$$

$$
= \Big\langle \nabla_{\theta_i} L_p(\{x_i^{t+1}\}, \{y_i^{t+1}\}, v^{t+1}, z^{t+1}, \{\lambda_l^{t+1}\}, \{\theta_i^t\}) - \nabla_{\theta_i} L_p(\{x_i^t\}, \{y_i^t\}, v^t, z^t, \{\lambda_l^t\}, \{\theta_i^t\}), \theta_i^{t+1} - \theta_i^t \Big\rangle
$$

$$
+ \frac{c_2^{t-1} - c_2^t}{2} (\|\theta_i^{t+1}\|^2 - \|\theta_i^t\|^2) - \frac{c_2^{t-1} - c_2^t}{2} \|\theta_i^{t+1} - \theta_i^t\|^2.
$$

$$(96)$$

And combining the Cauchy-Schwarz inequality with Assumption 1, we can obtain,

$$\left\langle \nabla_{\boldsymbol{\theta}_i} L_p(\{\boldsymbol{x}_i^{t+1}\}, \{\boldsymbol{y}_i^{t+1}\}, \boldsymbol{v}^{t+1}, \boldsymbol{z}^{t+1}, \{\lambda_l^{t+1}\}, \{\boldsymbol{\theta}_i^t\}) - \nabla_{\boldsymbol{\theta}_i} L_p(\{\boldsymbol{x}_i^t\}, \{\boldsymbol{y}_i^t\}, \boldsymbol{v}^t, \boldsymbol{z}^t, \{\lambda_l^t\}, \{\boldsymbol{\theta}_i^t\}), \boldsymbol{\theta}_i^{t+1} - \boldsymbol{\theta}_i^t \right\rangle$$

$$= \left\langle \nabla_{\boldsymbol{\theta}_i} L_p(\{\boldsymbol{x}_i^{t+1}\}, \{\boldsymbol{y}_i^{t+1}\}, \boldsymbol{v}^{t+1}, \boldsymbol{z}^{t+1}, \{\lambda_l^t\}, \{\boldsymbol{\theta}_i^t\}) - \nabla_{\boldsymbol{\theta}_i} L_p(\{\boldsymbol{x}_i^t\}, \{\boldsymbol{y}_i^t\}, \boldsymbol{v}^t, \boldsymbol{z}^t, \{\lambda_l^t\}, \{\boldsymbol{\theta}_i^t\}), \boldsymbol{\theta}_i^{t+1} - \boldsymbol{\theta}_i^t \right\rangle$$

$$\leq \frac{L^2}{2a_3} \left( \sum_{i=1}^N (||\boldsymbol{x}_i^{t+1} - \boldsymbol{x}_i^t||^2 + ||\boldsymbol{y}_i^{t+1} - \boldsymbol{y}_i^t||^2) + ||\boldsymbol{v}^{t+1} - \boldsymbol{v}^t||^2 + ||\boldsymbol{z}^{t+1} - \boldsymbol{z}^t||^2 \right) + \frac{a_3}{2} ||\boldsymbol{\theta}_i^{t+1} - \boldsymbol{\theta}_i^t||^2, \tag{97}$$

where $a_3 > 0$ is a constant. Thus, we can get the upper bound of $(2a)$ by combining Eq. (96) with Eq. (97), that is,

$$\sum_{i=1}^N \left\langle \nabla_{\boldsymbol{\theta}_i} \widetilde{L}_p(\{\boldsymbol{x}_i^{t+1}\}, \{\boldsymbol{y}_i^{t+1}\}, \boldsymbol{v}^{t+1}, \boldsymbol{z}^{t+1}, \{\lambda_l^{t+1}\}, \{\boldsymbol{\theta}_i^t\}) - \nabla_{\boldsymbol{\theta}_i} \widetilde{L}_p(\{\boldsymbol{x}_i^t\}, \{\boldsymbol{y}_i^t\}, \boldsymbol{v}^t, \boldsymbol{z}^t, \{\lambda_l^t\}, \{\boldsymbol{\theta}_i^t\}), \boldsymbol{\theta}_i^{t+1} - \boldsymbol{\theta}_i^t \right\rangle$$

$$\leq \sum_{i \in \boldsymbol{\mathcal{Q}}^{t+1}} \left( \frac{L^2}{2a_3} \left( \sum_{i=1}^N (||\boldsymbol{x}_i^{t+1} - \boldsymbol{x}_i^t||^2 + ||\boldsymbol{y}_i^{t+1} - \boldsymbol{y}_i^t||^2) + ||\boldsymbol{v}^{t+1} - \boldsymbol{v}^t||^2 + ||\boldsymbol{z}^{t+1} - \boldsymbol{z}^t||^2 \right) + \frac{a_3}{2} ||\boldsymbol{\theta}_i^{t+1} - \boldsymbol{\theta}_i^t||^2 \right.$$

$$\left. + \frac{c_2^{t-1} - c_2^t}{2} (||\boldsymbol{\theta}_i^{t+1}||^2 - ||\boldsymbol{\theta}_i^t||^2) - \frac{c_2^{t-1} - c_2^t}{2} ||\boldsymbol{\theta}_i^{t+1} - \boldsymbol{\theta}_i^t||^2 \right). \tag{98}$$

Next we focus on the $(2b)$ in Eq. (95). According to Cauchy-Schwarz inequality we can write $(2b)$ as,

$$\sum_{i=1}^N \left\langle \nabla_{\boldsymbol{\theta}_i} \widetilde{L}_p(\{\boldsymbol{x}_i^t\}, \{\boldsymbol{y}_i^t\}, \boldsymbol{v}^t, \boldsymbol{z}^t, \{\lambda_l^t\}, \{\boldsymbol{\theta}_i^t\}) - \nabla_{\boldsymbol{\theta}_i} \widetilde{L}_p(\{\boldsymbol{x}_i^t\}, \{\boldsymbol{y}_i^t\}, \boldsymbol{v}^t, \boldsymbol{z}^t, \{\lambda_l^t\}, \{\boldsymbol{\theta}_i^{t-1}\}), \boldsymbol{v}_{2,l}^{t+1} \right\rangle$$

$$\leq \sum_{i=1}^N \left( \frac{a_4}{2} ||\nabla_{\boldsymbol{\theta}_i} \widetilde{L}_p(\{\boldsymbol{x}_i^t\}, \{\boldsymbol{y}_i^t\}, \boldsymbol{v}^t, \boldsymbol{z}^t, \{\lambda_l^t\}, \{\boldsymbol{\theta}_i^t\}) - \nabla_{\boldsymbol{\theta}_i} \widetilde{L}_p(\{\boldsymbol{x}_i^t\}, \{\boldsymbol{y}_i^t\}, \boldsymbol{v}^t, \boldsymbol{z}^t, \{\lambda_l^t\}, \{\boldsymbol{\theta}_i^{t-1}\})||^2 \right.$$

$$\left. + \frac{1}{2a_4} ||\boldsymbol{v}_{2,l}^{t+1}||^2 \right), \tag{99}$$

where $a_4 > 0$ is a constant. Then, we focus on the $(2c)$ in Eq. (95). Defining $L_2' = L + c_2^0$, according to Assumption 1 and the trigonometric inequality, we have,

$$||\nabla_{\boldsymbol{\theta}_i} \widetilde{L}_p(\{\boldsymbol{x}_i^t\}, \{\boldsymbol{y}_i^t\}, \boldsymbol{v}^t, \boldsymbol{z}^t, \{\lambda_l^t\}, \{\boldsymbol{\theta}_i^t\}) - \nabla_{\boldsymbol{\theta}_i} \widetilde{L}_p(\{\boldsymbol{x}_i^t\}, \{\boldsymbol{y}_i^t\}, \boldsymbol{v}^t, \boldsymbol{z}^t, \{\lambda_l^t\}, \{\boldsymbol{\theta}_i^{t-1}\})||$$

$$\leq L_2' ||\boldsymbol{\theta}_i^t - \boldsymbol{\theta}_i^{t-1}||. \tag{100}$$

Following Eq. (100) and the strong concavity of $\widetilde{L}_p(\{\boldsymbol{x}_i\}, \{\boldsymbol{y}_i\}, \boldsymbol{v}, \boldsymbol{z}, \{\lambda_l\}, \{\boldsymbol{\theta}_i\})$ *w.r.t* $\boldsymbol{\theta}_i$, the upper bound of $(2c)$ can be obtained, that is,

$$\sum_{i=1}^N \left\langle \nabla_{\boldsymbol{\theta}_i} \widetilde{L}_p(\{\boldsymbol{x}_i^t\}, \{\boldsymbol{y}_i^t\}, \boldsymbol{v}^t, \boldsymbol{z}^t, \{\lambda_l^t\}, \{\boldsymbol{\theta}_i^t\}) - \nabla_{\boldsymbol{\theta}_i} \widetilde{L}_p(\{\boldsymbol{x}_i^t\}, \{\boldsymbol{y}_i^t\}, \boldsymbol{v}^t, \boldsymbol{z}^t, \{\lambda_l^t\}, \{\boldsymbol{\theta}_i^{t-1}\}), \boldsymbol{\theta}_i^t - \boldsymbol{\theta}_i^{t-1} \right\rangle$$

$$\leq \sum_{i=1}^N \left( -\frac{1}{L_2' + c_2^{t-1}} ||\nabla_{\boldsymbol{\theta}_i} \widetilde{L}_p(\{\boldsymbol{x}_i^t\}, \{\boldsymbol{y}_i^t\}, \boldsymbol{v}^t, \boldsymbol{z}^t, \{\lambda_l^t\}, \{\boldsymbol{\theta}_i^t\}) - \nabla_{\boldsymbol{\theta}_i} \widetilde{L}_p(\{\boldsymbol{x}_i^t\}, \{\boldsymbol{y}_i^t\}, \boldsymbol{v}^t, \boldsymbol{z}^t, \{\lambda_l^t\}, \{\boldsymbol{\theta}_i^{t-1}\})||^2 \right.$$

$$\left. - \frac{c_2^{t-1} L_2'}{L_2' + c_2^{t-1}} ||\boldsymbol{\theta}_i^t - \boldsymbol{\theta}_i^{t-1}||^2 \right). \tag{101}$$

In addition, the following inequality can also be obtained,

$$\sum_{i=1}^N \frac{1}{\eta_{\boldsymbol{\theta}}} \left\langle \boldsymbol{\theta}_i^t - \boldsymbol{\theta}_i^{t-1}, \boldsymbol{\theta}_i^{t+1} - \boldsymbol{\theta}_i^t \right\rangle \leq \sum_{i=1}^N \left( \frac{1}{2\eta_{\boldsymbol{\theta}}} ||\boldsymbol{\theta}_i^{t+1} - \boldsymbol{\theta}_i^t||^2 - \frac{1}{2\eta_{\boldsymbol{\theta}}} ||\boldsymbol{v}_{2,l}^{t+1}||^2 + \frac{1}{2\eta_{\boldsymbol{\theta}}} ||\boldsymbol{\theta}_i^t - \boldsymbol{\theta}_i^{t-1}||^2 \right). \tag{102}$$

Combining Eq. (94), (95), (98), (99), (101), (102), $\frac{\eta_{\boldsymbol{\theta}}}{2} \leq \frac{1}{L_2' + c_2^0}$, and setting $a_4 = \eta_{\boldsymbol{\theta}}$, we have,

$$L_p(\{\boldsymbol{x}_i^{t+1}\}, \{\boldsymbol{y}_i^{t+1}\}, \boldsymbol{v}^{t+1}, \boldsymbol{z}^{t+1}, \{\lambda_l^{t+1}\}, \{\boldsymbol{\theta}_i^{t+1}\}) - L_p(\{\boldsymbol{x}_i^{t+1}\}, \{\boldsymbol{y}_i^{t+1}\}, \boldsymbol{v}^{t+1}, \boldsymbol{z}^{t+1}, \{\lambda_l^{t+1}\}, \{\boldsymbol{\theta}_i^t\})$$

$$\leq \frac{|\boldsymbol{\mathcal{Q}}^{t+1}| L^2}{2a_3} \left( \sum_{i=1}^N (||\boldsymbol{x}_i^{t+1} - \boldsymbol{x}_i^t||^2 + ||\boldsymbol{y}_i^{t+1} - \boldsymbol{y}_i^t||^2) + ||\boldsymbol{v}^{t+1} - \boldsymbol{v}^t||^2 + ||\boldsymbol{z}^{t+1} - \boldsymbol{z}^t||^2 \right)$$

$$+ \left( \frac{a_3}{2} - \frac{c_2^{t-1} - c_2^t}{2} + \frac{1}{2\eta_{\boldsymbol{\theta}}} \right) \sum_{i=1}^N ||\boldsymbol{\theta}_i^{t+1} - \boldsymbol{\theta}_i^t||^2 + \frac{c_2^{t-1}}{2} \sum_{i=1}^N (||\boldsymbol{\theta}_i^{t+1}||^2 - ||\boldsymbol{\theta}_i^t||^2) + \frac{1}{2\eta_{\boldsymbol{\theta}}} \sum_{i=1}^N ||\boldsymbol{\theta}_i^t - \boldsymbol{\theta}_i^{t-1}||^2. \tag{103}$$

By combining Lemma 1 with Eq. (90) and Eq. (103), we conclude the proof of Lemma 2.

**Lemma 3** *Firstly, we denote $S_1^{t+1}$, $S_2^{t+1}$ and $F^{t+1}$ as,*

$$S_1^{t+1} = \frac{4}{\eta_\lambda^2 c_1^{t+1}} \sum_{l=1}^{|\mathcal{P}^t|} ||\lambda_l^{t+1} - \lambda_l^t||^2 - \frac{4}{\eta_\lambda}\left(\frac{c_1^{t-1}}{c_1^t} - 1\right) \sum_{l=1}^{|\mathcal{P}^t|} ||\lambda_l^{t+1}||^2, \tag{104}$$

$$S_2^{t+1} = \frac{4}{\eta_\theta^2 c_2^{t+1}} \sum_{i=1}^{N} ||\theta_i^{t+1} - \theta_i^t||^2 - \frac{4}{\eta_\theta}\left(\frac{c_2^{t-1}}{c_2^t} - 1\right) \sum_{i=1}^{N} ||\theta_i^{t+1}||^2, \tag{105}$$

$$F^{t+1} = L_p(\{x_i^{t+1}\}, \{y_i^{t+1}\}, z^{t+1}, h^{t+1}, \{\lambda_l^{t+1}\}, \{\theta_i^{t+1}\}) + S_1^{t+1} + S_2^{t+1}$$
$$- \frac{7}{2\eta_\lambda} \sum_{l=1}^{|\mathcal{P}^t|} ||\lambda_l^{t+1} - \lambda_l^t||^2 - \frac{c_1^t}{2} \sum_{l=1}^{|\mathcal{P}^t|} ||\lambda_l^{t+1}||^2 - \frac{7}{2\eta_\theta} \sum_{i=1}^{N} ||\theta_i^{t+1} - \theta_i^t||^2 - \frac{c_2^t}{2} \sum_{i=1}^{N} ||\theta_i^{t+1}||^2. \tag{106}$$

*Defining $a_5 = \max\{1, 1 + L^2, 6\tau k_1 N L^2\}$, $\forall t \geq T_1$, we have,*

$$F^{t+1} - F^t$$
$$\leq \left(\frac{L+a_5}{2} - \frac{1}{\eta_x^t} + \frac{\eta_\lambda|\mathcal{P}^t|L^2}{2} + \frac{\eta_\theta|\mathcal{Q}^{t+1}|L^2}{2} + \frac{8|\mathcal{P}^t|L^2}{\eta_\lambda(c_1^t)^2} + \frac{8NL^2}{\eta_\theta(c_2^t)^2}\right) \sum_{i=1}^{N} ||x_i^{t+1} - x_i^t||^2$$
$$+ \left(\frac{L+a_5}{2} - \frac{1}{\eta_y^t} + \frac{\eta_\lambda|\mathcal{P}^t|L^2}{2} + \frac{\eta_\theta|\mathcal{Q}^{t+1}|L^2}{2} + \frac{8|\mathcal{P}^t|L^2}{\eta_\lambda(c_1^t)^2} + \frac{8NL^2}{\eta_\theta(c_2^t)^2}\right) \sum_{i=1}^{N} ||y_i^{t+1} - y_i^t||^2$$
$$+ \left(\frac{L+a_5}{2} - \frac{1}{\eta_v^t} + \frac{\eta_\lambda|\mathcal{P}^t|L^2}{2} + \frac{\eta_\theta|\mathcal{Q}^{t+1}|L^2}{2} + \frac{8|\mathcal{P}^t|L^2}{\eta_\lambda(c_1^t)^2} + \frac{8NL^2}{\eta_\theta(c_2^t)^2}\right) ||v^{t+1} - v^t||^2$$
$$+ \left(\frac{L+a_5}{2} - \frac{1}{\eta_z^t} + \frac{\eta_\lambda|\mathcal{P}^t|L^2}{2} + \frac{\eta_\theta|\mathcal{Q}^{t+1}|L^2}{2} + \frac{8|\mathcal{P}^t|L^2}{\eta_\lambda(c_1^t)^2} + \frac{8NL^2}{\eta_\theta(c_2^t)^2}\right) ||z^{t+1} - z^t||^2$$
$$- \left(\frac{1}{10\eta_\lambda} - \frac{6\tau k_1 NL^2}{2}\right) \sum_{l=1}^{|\mathcal{P}^t|} ||\lambda_l^{t+1} - \lambda_l^t||^2 - \frac{1}{10\eta_\theta} \sum_{i=1}^{N} ||\theta_i^{t+1} - \theta_i^t||^2 + \frac{c_1^{t-1} - c_1^t}{2} \sum_{l=1}^{|\mathcal{P}^t|} ||\lambda_l^{t+1}||^2$$
$$+ \frac{c_2^{t-1} - c_2^t}{2} \sum_{i=1}^{N} ||\theta_i^{t+1}||^2 + \frac{4}{\eta_\lambda}\left(\frac{c_1^{t-2}}{c_1^{t-1}} - \frac{c_1^{t-1}}{c_1^t}\right) \sum_{l=1}^{|\mathcal{P}^t|} ||\lambda_l^t||^2 + \frac{4}{\eta_\theta}\left(\frac{c_2^{t-2}}{c_2^{t-1}} - \frac{c_2^{t-1}}{c_2^t}\right) \sum_{i=1}^{N} ||\theta_i^t||^2. \tag{107}$$

***Proof of Lemma 3:***

Let $a_1 = \frac{1}{\eta_\lambda}$, $a_3 = \frac{1}{\eta_\theta}$ and substitute them into the Lemma 2, $\forall t \geq T_1$, we have,

$$L_p(\{x_i^{t+1}\}, \{y_i^{t+1}\}, v^{t+1}, z^{t+1}, \{\lambda_l^{t+1}\}, \{\theta_i^{t+1}\}) - L_p(\{x_i^t\}, \{y_i^t\}, v^t, z^t, \{\lambda_l^t\}, \{\theta_i^t\})$$
$$\leq \left(\frac{L+L^2+1}{2} - \frac{1}{\eta_x^t} + \frac{\eta_\lambda|\mathcal{P}^t|L^2 + \eta_\theta|\mathcal{Q}^{t+1}|L^2}{2}\right) \sum_{i=1}^{N} ||x_i^{t+1} - x_i^t||^2$$
$$+ \left(\frac{L+1}{2} - \frac{1}{\eta_y^t} + \frac{\eta_\lambda|\mathcal{P}^t|L^2 + \eta_\theta|\mathcal{Q}^{t+1}|L^2}{2}\right) \sum_{i=1}^{N} ||y_i^{t+1} - y_i^t||^2$$
$$+ \left(\frac{L+6\tau k_1 NL^2}{2} - \frac{1}{\eta_v^t} + \frac{\eta_\lambda|\mathcal{P}^t|L^2 + \eta_\theta|\mathcal{Q}^{t+1}|L^2}{2}\right) ||v^{t+1} - v^t||^2 + \frac{1}{2\eta_\lambda} \sum_{l=1}^{|\mathcal{P}^t|} ||\lambda_l^t - \lambda_l^{t-1}||^2$$
$$+ \left(\frac{L+6\tau k_1 NL^2}{2} - \frac{1}{\eta_z^t} + \frac{\eta_\lambda|\mathcal{P}^t|L^2 + \eta_\theta|\mathcal{Q}^{t+1}|L^2}{2}\right) ||z^{t+1} - z^t||^2 + \frac{1}{2\eta_\theta} \sum_{i=1}^{N} ||\theta_i^t - \theta_i^{t-1}||^2$$
$$+ \left(\frac{6\tau k_1 NL^2}{2} - \frac{c_1^{t-1} - c_1^t}{2} + \frac{1}{\eta_\lambda}\right) \sum_{l=1}^{|\mathcal{P}^t|} ||\lambda_l^{t+1} - \lambda_l^t||^2 + \left(\frac{1}{\eta_\theta} - \frac{c_2^{t-1} - c_2^t}{2}\right) \sum_{i=1}^{N} ||\theta_i^{t+1} - \theta_i^t||^2$$
$$+ \frac{c_1^{t-1}}{2} \sum_{l=1}^{|\mathcal{P}^t|} (||\lambda_l^{t+1}||^2 - ||\lambda_l^t||^2) + \frac{c_2^{t-1}}{2} \sum_{i=1}^{N} (||\theta_i^{t+1}||^2 - ||\theta_i^t||^2). \tag{108}$$

According to Eq. (19), in $(t+1)^{\text{th}}$ iteration, it follows that:

$$\left\langle \lambda_l^{t+1} - \lambda_l^t - \eta_\lambda \nabla_{\lambda_l} \widetilde{L}_p(\{x_i^{t+1}\}, \{y_i^{t+1}\}, v^{t+1}, z^{t+1}, \{\lambda_l^t\}, \{\theta_i^t\}), \lambda_l^t - \lambda_l^{t+1} \right\rangle = 0. \tag{109}$$

Similar to Eq. (109), in $t^{\text{th}}$ iteration, we have,

$$\left\langle \lambda_l^t - \lambda_l^{t-1} - \eta_\lambda \nabla_{\lambda_l} \widetilde{L}_p(\{x_i^t\}, \{y_i^t\}, v^t, z^t, \{\lambda_l^{t-1}\}, \{\theta_i^{t-1}\}), \lambda_l^{t+1} - \lambda_l^t \right\rangle = 0. \tag{110}$$

Thus, $\forall t \geq T_1$, by combining Eq. (109) with Eq. (110), we can obtain that,

$$
\begin{aligned}
&\tfrac{1}{\eta_\lambda} \left\langle \boldsymbol{v}_{1,l}^{t+1}, \lambda_l^{t+1} - \lambda_l^t \right\rangle \\
&= \left\langle \nabla_{\lambda_l} \widetilde{L}_p(\{\boldsymbol{x}_i^{t+1}\},\{\boldsymbol{y}_i^{t+1}\},\boldsymbol{v}^{t+1},\boldsymbol{z}^{t+1},\{\lambda_l^t\},\{\boldsymbol{\theta}_i^t\}) - \nabla_{\lambda_l} \widetilde{L}_p(\{\boldsymbol{x}_i^t\},\{\boldsymbol{y}_i^t\},\boldsymbol{v}^t,\boldsymbol{z}^t,\{\lambda_l^{t-1}\},\{\boldsymbol{\theta}_i^{t-1}\}), \lambda_l^{t+1} - \lambda_l^t \right\rangle \\
&= \left\langle \nabla_{\lambda_l} \widetilde{L}_p(\{\boldsymbol{x}_i^{t+1}\},\{\boldsymbol{y}_i^{t+1}\},\boldsymbol{v}^{t+1},\boldsymbol{z}^{t+1},\{\lambda_l^t\},\{\boldsymbol{\theta}_i^t\}) - \nabla_{\lambda_l} \widetilde{L}_p(\{\boldsymbol{x}_i^t\},\{\boldsymbol{y}_i^t\},\boldsymbol{v}^t,\boldsymbol{z}^t,\{\lambda_l^t\},\{\boldsymbol{\theta}_i^t\}), \lambda_l^{t+1} - \lambda_l^t \right\rangle \\
&\quad + \left\langle \nabla_{\lambda_l} \widetilde{L}_p(\{\boldsymbol{x}_i^t\},\{\boldsymbol{y}_i^t\},\boldsymbol{v}^t,\boldsymbol{z}^t,\{\lambda_l^t\},\{\boldsymbol{\theta}_i^t\}) - \nabla_{\lambda_l} \widetilde{L}_p(\{\boldsymbol{x}_i^t\},\{\boldsymbol{y}_i^t\},\boldsymbol{v}^t,\boldsymbol{z}^t,\{\lambda_l^{t-1}\},\{\boldsymbol{\theta}_i^{t-1}\}), \boldsymbol{v}_{1,l}^{t+1} \right\rangle \\
&\quad + \left\langle \nabla_{\lambda_l} \widetilde{L}_p(\{\boldsymbol{x}_i^t\},\{\boldsymbol{y}_i^t\},\boldsymbol{v}^t,\boldsymbol{z}^t,\{\lambda_l^t\},\{\boldsymbol{\theta}_i^t\}) - \nabla_{\lambda_l} \widetilde{L}_p(\{\boldsymbol{x}_i^t\},\{\boldsymbol{y}_i^t\},\boldsymbol{v}^t,\boldsymbol{z}^t,\{\lambda_l^{t-1}\},\{\boldsymbol{\theta}_i^{t-1}\}), \lambda_l^t - \lambda_l^{t-1} \right\rangle .
\end{aligned}
\tag{111}
$$

Since we have that,

$$
\tfrac{1}{\eta_\lambda} \left\langle \boldsymbol{v}_{1,l}^{t+1}, \lambda_l^{t+1} - \lambda_l^t \right\rangle = \tfrac{1}{2\eta_\lambda} \|\lambda_l^{t+1} - \lambda_l^t\|^2 + \tfrac{1}{2\eta_\lambda} \|\boldsymbol{v}_{1,l}^{t+1}\|^2 - \tfrac{1}{2\eta_\lambda} \|\lambda_l^t - \lambda_l^{t-1}\|^2,
\tag{112}
$$

it follows from Eq. (111) and Eq. (112) that,

$$
\begin{aligned}
&\tfrac{1}{2\eta_\lambda} \|\lambda_l^{t+1} - \lambda_l^t\|^2 + \tfrac{1}{2\eta_\lambda} \|\boldsymbol{v}_{1,l}^{t+1}\|^2 - \tfrac{1}{2\eta_\lambda} \|\lambda_l^t - \lambda_l^{t-1}\|^2 \\
&= \tfrac{L^2}{2b_1^t} \left( \sum_{i=1}^N (\|\boldsymbol{x}_i^{t+1} - \boldsymbol{x}_i^t\|^2 + \|\boldsymbol{y}_i^{t+1} - \boldsymbol{y}_i^t\|^2) + \|\boldsymbol{v}^{t+1} - \boldsymbol{v}^t\|^2 + \|\boldsymbol{z}^{t+1} - \boldsymbol{z}^t\|^2 \right) + \tfrac{b_1^t}{2} \|\lambda_l^{t+1} - \lambda_l^t\|^2 \\
&\quad + \tfrac{c_1^{t-1} - c_1^t}{2} (\|\lambda_l^{t+1}\|^2 - \|\lambda_l^t\|^2) - \tfrac{c_1^{t-1} - c_1^t}{2} \|\lambda_l^{t+1} - \lambda_l^t\|^2 \\
&\quad + \tfrac{\eta_\lambda}{2} \|\nabla_{\lambda_l} \widetilde{L}_p(\{\boldsymbol{x}_i^t\},\{\boldsymbol{y}_i^t\},\boldsymbol{v}^t,\boldsymbol{z}^t,\{\lambda_l^t\},\{\boldsymbol{\theta}_i^t\}) - \nabla_{\lambda_l} \widetilde{L}_p(\{\boldsymbol{x}_i^t\},\{\boldsymbol{y}_i^t\},\boldsymbol{v}^t,\boldsymbol{z}^t,\{\lambda_l^{t-1}\},\{\boldsymbol{\theta}_i^{t-1}\})\|^2 + \tfrac{1}{2\eta_\lambda} \|\boldsymbol{v}_{1,l}^{t+1}\|^2 \\
&\quad - \tfrac{1}{L_1' + c_1^{t-1}} \|\nabla_{\lambda_l} \widetilde{L}_p(\{\boldsymbol{x}_i^t\},\{\boldsymbol{y}_i^t\},\boldsymbol{v}^t,\boldsymbol{z}^t,\{\lambda_l^t\},\{\boldsymbol{\theta}_i^t\}) - \nabla_{\lambda_l} \widetilde{L}_p(\{\boldsymbol{x}_i^t\},\{\boldsymbol{y}_i^t\},\boldsymbol{v}^t,\boldsymbol{z}^t,\{\lambda_l^{t-1}\},\{\boldsymbol{\theta}_i^{t-1}\})\|^2 \\
&\quad - \tfrac{c_1^{t-1} L_1'}{L_1' + c_1^{t-1}} \|\lambda_l^t - \lambda_l^{t-1}\|^2,
\end{aligned}
\tag{113}
$$

where $b_1^t > 0$. According to the setting that $c_1^0 \leq L_1'$, we have $-\tfrac{c_1^{t-1} L_1'}{L_1' + c_1^{t-1}} \leq -\tfrac{c_1^{t-1} L_1'}{2L_1'} = -\tfrac{c_1^{t-1}}{2} \leq -\tfrac{c_1^t}{2}$. Multiplying both sides of Eq. (113) by $\tfrac{8}{\eta_\lambda c_1^t}$, we have,

$$
\begin{aligned}
&\tfrac{4}{\eta_\lambda^2 c_1^t} \|\lambda_l^{t+1} - \lambda_l^t\|^2 - \tfrac{4}{\eta_\lambda} \left( \tfrac{c_1^{t-1} - c_1^t}{c_1^t} \right) \|\lambda_l^{t+1}\|^2 \\
&\leq \tfrac{4}{\eta_\lambda^2 c_1^t} \|\lambda_l^t - \lambda_l^{t-1}\|^2 - \tfrac{4}{\eta_\lambda} \left( \tfrac{c_1^{t-1} - c_1^t}{c_1^t} \right) \|\lambda_l^t\|^2 + \tfrac{4 b_1^t}{\eta_\lambda c_1^t} \|\lambda_l^{t+1} - \lambda_l^t\|^2 - \tfrac{4}{\eta_\lambda} \|\lambda_l^t - \lambda_l^{t-1}\|^2 \\
&\quad + \tfrac{4 L^2}{\eta_\lambda c_1^t b_1^t} \left( \sum_{i=1}^N (\|\boldsymbol{x}_i^{t+1} - \boldsymbol{x}_i^t\|^2 + \|\boldsymbol{y}_i^{t+1} - \boldsymbol{y}_i^t\|^2) + \|\boldsymbol{v}^{t+1} - \boldsymbol{v}^t\|^2 + \|\boldsymbol{z}^{t+1} - \boldsymbol{z}^t\|^2 \right).
\end{aligned}
\tag{114}
$$

Setting $b_1^t = \tfrac{c_1^t}{2}$ in Eq. (114) and using the definition of $S_1^t$, $\forall t \geq T_1$, we have,

$$
\begin{aligned}
&S_1^{t+1} - S_1^t \\
&\leq \sum_{l=1}^{|\boldsymbol{\mathcal{P}}^t|} \tfrac{4}{\eta_\lambda} \left( \tfrac{c_1^{t-2}}{c_1^{t-1}} - \tfrac{c_1^{t-1}}{c_1^t} \right) \|\lambda_l^t\|^2 + \sum_{l=1}^{|\boldsymbol{\mathcal{P}}^t|} \left( \tfrac{2}{\eta_\lambda} + \tfrac{4}{\eta_\lambda^2} \left( \tfrac{1}{c_1^{t+1}} - \tfrac{1}{c_1^t} \right) \right) \|\lambda_l^{t+1} - \lambda_l^t\|^2 \\
&\quad - \sum_{l=1}^{|\boldsymbol{\mathcal{P}}^t|} \tfrac{4}{\eta_\lambda} \|\lambda_l^t - \lambda_l^{t-1}\|^2 + \tfrac{8 |\boldsymbol{\mathcal{P}}^t| L^2}{\eta_\lambda (c_1^t)^2} \left( \sum_{i=1}^N (\|\boldsymbol{x}_i^{t+1} - \boldsymbol{x}_i^t\|^2 + \|\boldsymbol{y}_i^{t+1} - \boldsymbol{y}_i^t\|^2) + \|\boldsymbol{v}^{t+1} - \boldsymbol{v}^t\|^2 + \|\boldsymbol{z}^{t+1} - \boldsymbol{z}^t\|^2 \right).
\end{aligned}
\tag{115}
$$

Similarly, according to Eq. (20), it follows that,

$$
\begin{aligned}
&\tfrac{1}{\eta_\theta}\left\langle \boldsymbol{v}_{2,l}^{t+1}, \boldsymbol{\theta}_i^{t+1}-\boldsymbol{\theta}_i^t\right\rangle\\
&=\left\langle\nabla_{\boldsymbol{\theta}_i}\widetilde{L}_p(\{\boldsymbol{x}_i^{t+1}\},\{\boldsymbol{y}_i^{t+1}\},\boldsymbol{v}^{t+1},\boldsymbol{z}^{t+1},\{\lambda_l^{t+1}\},\{\boldsymbol{\theta}_i^t\})-\nabla_{\boldsymbol{\theta}_i}\widetilde{L}_p(\{\boldsymbol{x}_i^t\},\{\boldsymbol{y}_i^t\},\boldsymbol{v}^t,\boldsymbol{z}^t,\{\lambda_l^t\},\{\boldsymbol{\theta}_i^{t-1}\}), \boldsymbol{\theta}_i^{t+1}-\boldsymbol{\theta}_i^t\right\rangle\\
&=\left\langle\nabla_{\boldsymbol{\theta}_i}\widetilde{L}_p(\{\boldsymbol{x}_i^{t+1}\},\{\boldsymbol{y}_i^{t+1}\},\boldsymbol{v}^{t+1},\boldsymbol{z}^{t+1},\{\lambda_l^{t+1}\},\{\boldsymbol{\theta}_i^t\})-\nabla_{\boldsymbol{\theta}_i}\widetilde{L}_p(\{\boldsymbol{x}_i^t\},\{\boldsymbol{y}_i^t\},\boldsymbol{v}^t,\boldsymbol{z}^t,\{\lambda_l^t\},\{\boldsymbol{\theta}_i^t\}), \boldsymbol{\theta}_i^{t+1}-\boldsymbol{\theta}_i^t\right\rangle\\
&+\left\langle\nabla_{\boldsymbol{\theta}_i}\widetilde{L}_p(\{\boldsymbol{x}_i^t\},\{\boldsymbol{y}_i^t\},\boldsymbol{v}^t,\boldsymbol{z}^t,\{\lambda_l^t\},\{\boldsymbol{\theta}_i^t\})-\nabla_{\boldsymbol{\theta}_i}\widetilde{L}_p(\{\boldsymbol{x}_i^t\},\{\boldsymbol{y}_i^t\},\boldsymbol{v}^t,\boldsymbol{z}^t,\{\lambda_l^t\},\{\boldsymbol{\theta}_i^{t-1}\}), \boldsymbol{v}_{2,l}^{t+1}\right\rangle\\
&+\left\langle\nabla_{\boldsymbol{\theta}_i}\widetilde{L}_p(\{\boldsymbol{x}_i^t\},\{\boldsymbol{y}_i^t\},\boldsymbol{v}^t,\boldsymbol{z}^t,\{\lambda_l^t\},\{\boldsymbol{\theta}_i^t\})-\nabla_{\boldsymbol{\theta}_i}\widetilde{L}_p(\{\boldsymbol{x}_i^t\},\{\boldsymbol{y}_i^t\},\boldsymbol{v}^t,\boldsymbol{z}^t,\{\lambda_l^t\},\{\boldsymbol{\theta}_i^{t-1}\}), \boldsymbol{\theta}_i^t-\boldsymbol{\theta}_i^{t-1}\right\rangle.
\end{aligned}
\tag{116}
$$

In addition, since

$$
\tfrac{1}{\eta_\theta}\left\langle\boldsymbol{v}_{2,l}^{t+1},\boldsymbol{\theta}_i^{t+1}-\boldsymbol{\theta}_i^t\right\rangle=\tfrac{1}{2\eta_\theta}||\boldsymbol{\theta}_i^{t+1}-\boldsymbol{\theta}_i^t||^2+\tfrac{1}{2\eta_\theta}||\boldsymbol{v}_{2,l}^{t+1}||^2-\tfrac{1}{2\eta_\theta}||\boldsymbol{\theta}_i^t-\boldsymbol{\theta}_i^{t-1}||^2,
\tag{117}
$$

it follows that,

$$
\begin{aligned}
&\tfrac{1}{2\eta_\theta}||\boldsymbol{\theta}_i^{t+1}-\boldsymbol{\theta}_i^t||^2+\tfrac{1}{2\eta_\theta}||\boldsymbol{v}_{2,l}^{t+1}||^2-\tfrac{1}{2\eta_\theta}||\boldsymbol{\theta}_i^t-\boldsymbol{\theta}_i^{t-1}||^2\\
&=\tfrac{L^2}{2b_2^t}(\sum_{i=1}^N(||\boldsymbol{x}_i^{t+1}-\boldsymbol{x}_i^t||^2+||\boldsymbol{y}_i^{t+1}-\boldsymbol{y}_i^t||^2)+||\boldsymbol{v}^{t+1}-\boldsymbol{v}^t||^2+||\boldsymbol{z}^{t+1}-\boldsymbol{z}^t||^2)+\tfrac{b_2^t}{2}||\boldsymbol{\theta}_i^{t+1}-\boldsymbol{\theta}_i^t||^2\\
&+\tfrac{c_2^{t-1}-c_2^t}{2}(||\boldsymbol{\theta}_i^{t+1}||^2-||\boldsymbol{\theta}_i^t||^2)-\tfrac{c_2^{t-1}-c_2^t}{2}||\boldsymbol{\theta}_i^{t+1}-\boldsymbol{\theta}_i^t||^2-\tfrac{c_2^{t-1}L_2'}{L_2'+c_2^{t-1}}||\boldsymbol{\theta}_i^t-\boldsymbol{\theta}_i^{t-1}||^2+\tfrac{1}{2\eta_\theta}||\boldsymbol{v}_{2,l}^{t+1}||^2\\
&+\tfrac{\eta_\theta}{2}||\nabla_{\boldsymbol{\theta}_i}\widetilde{L}_p(\{\boldsymbol{x}_i^t\},\{\boldsymbol{y}_i^t\},\boldsymbol{v}^t,\boldsymbol{z}^t,\{\lambda_l^t\},\{\boldsymbol{\theta}_i^t\})-\nabla_{\boldsymbol{\theta}_i}\widetilde{L}_p(\{\boldsymbol{x}_i^t\},\{\boldsymbol{y}_i^t\},\boldsymbol{v}^t,\boldsymbol{z}^t,\{\lambda_l^t\},\{\boldsymbol{\theta}_i^{t-1}\})||^2\\
&-\tfrac{1}{L_2'+c_2^{t-1}}||\nabla_{\boldsymbol{\theta}_i}\widetilde{L}_p(\{\boldsymbol{x}_i^t\},\{\boldsymbol{y}_i^t\},\boldsymbol{v}^t,\boldsymbol{z}^t,\{\lambda_l^t\},\{\boldsymbol{\theta}_i^t\})-\nabla_{\boldsymbol{\theta}_i}\widetilde{L}_p(\{\boldsymbol{x}_i^t\},\{\boldsymbol{y}_i^t\},\boldsymbol{v}^t,\boldsymbol{z}^t,\{\lambda_l^t\},\{\boldsymbol{\theta}_i^{t-1}\})||^2.
\end{aligned}
\tag{118}
$$

According to the setting $c_2^0\le L_2'$, we have $-\tfrac{c_2^{t-1}L_2'}{L_2'+c_2^{t-1}}\le-\tfrac{c_2^{t-1}L_2'}{2L_2'}=-\tfrac{c_2^{t-1}}{2}\le-\tfrac{c_2^t}{2}$. Multiplying both sides of Eq. (118) by $\tfrac{8}{\eta_\theta c_2^t}$, we have,

$$
\begin{aligned}
&\tfrac{4}{\eta_\theta^2 c_2^t}||\boldsymbol{\theta}_i^{t+1}-\boldsymbol{\theta}_i^t||^2-\tfrac{4}{\eta_\theta}(\tfrac{c_2^{t-1}-c_2^t}{c_2^t})||\boldsymbol{\theta}_i^{t+1}||^2\\
&\le\tfrac{4}{\eta_\theta^2 c_2^t}||\boldsymbol{\theta}_i^t-\boldsymbol{\theta}_i^{t-1}||^2-\tfrac{4}{\eta_\theta}(\tfrac{c_2^{t-1}-c_2^t}{c_2^t})||\boldsymbol{\theta}_i^t||^2+\tfrac{4b_2^t}{\eta_\theta c_2^t}||\boldsymbol{\theta}_i^{t+1}-\boldsymbol{\theta}_i^t||^2-\tfrac{4}{\eta_\theta}||\boldsymbol{\theta}_i^t-\boldsymbol{\theta}_i^{t-1}||^2\\
&+\tfrac{4L^2}{\eta_\theta c_2^t b_2^t}(\sum_{i=1}^N(||\boldsymbol{x}_i^{t+1}-\boldsymbol{x}_i^t||^2+||\boldsymbol{y}_i^{t+1}-\boldsymbol{y}_i^t||^2)+||\boldsymbol{v}^{t+1}-\boldsymbol{v}^t||^2+||\boldsymbol{z}^{t+1}-\boldsymbol{z}^t||^2).
\end{aligned}
\tag{119}
$$

Setting $b_2^t=\tfrac{c_2^t}{2}$ in Eq. (119) and utilizing the definition of $S_2^t$, we have that,

$$
\begin{aligned}
&S_2^{t+1}-S_2^t\\
&\le\sum_{i=1}^N\tfrac{4}{\eta_\theta}(\tfrac{c_2^{t-2}}{c_2^{t-1}}-\tfrac{c_2^{t-1}}{c_2^t})||\boldsymbol{\theta}_i^t||^2+\sum_{i=1}^N(\tfrac{2}{\eta_\theta}+\tfrac{4}{\eta_\theta^2}(\tfrac{1}{c_2^{t+1}}-\tfrac{1}{c_2^t}))||\boldsymbol{\theta}_i^{t+1}-\boldsymbol{\theta}_i^t||^2\\
&-\sum_{i=1}^N\tfrac{4}{\eta_\theta}||\boldsymbol{\theta}_i^t-\boldsymbol{\theta}_i^{t-1}||^2+\tfrac{8NL^2}{\eta_\theta(c_2^t)^2}(\sum_{i=1}^N(||\boldsymbol{x}_i^{t+1}-\boldsymbol{x}_i^t||^2+||\boldsymbol{y}_i^{t+1}-\boldsymbol{y}_i^t||^2)+||\boldsymbol{v}^{t+1}-\boldsymbol{v}^t||^2+||\boldsymbol{z}^{t+1}-\boldsymbol{z}^t||^2).
\end{aligned}
\tag{120}
$$

Based on the setting of $c_1^t$ and $c_2^t$, we can obtain that $\tfrac{\eta_\lambda}{10}\ge\tfrac{1}{c_1^{t+1}}-\tfrac{1}{c_1^t}, \tfrac{\eta_\theta}{10}\ge\tfrac{1}{c_2^{t+1}}-\tfrac{1}{c_2^t}, \forall t\ge T_1$. Defining $a_5=\max\{1,1+L^2,6\tau k_1 NL^2\}$. Combining the definition of $F^{t+1}$ with Eq. (115) and

Eq. (120), $\forall t \geq T_1$, we can obtain that,

$$F^{t+1} - F^t$$

$$\leq \left( \frac{L+a_5}{2} - \frac{1}{\eta_{\boldsymbol{x}}^t} + \frac{\eta_\lambda |\boldsymbol{\mathcal{P}}^t| L^2}{2} + \frac{\eta_{\boldsymbol{\theta}} |\boldsymbol{\mathcal{Q}}^{t+1}| L^2}{2} + \frac{8|\boldsymbol{\mathcal{P}}^t| L^2}{\eta_\lambda (c_1^t)^2} + \frac{8NL^2}{\eta_{\boldsymbol{\theta}} (c_2^t)^2} \right) \sum_{i=1}^{N} ||\boldsymbol{x}_i^{t+1} - \boldsymbol{x}_i^t||^2$$

$$+ \left( \frac{L+a_5}{2} - \frac{1}{\eta_{\boldsymbol{y}}^t} + \frac{\eta_\lambda |\boldsymbol{\mathcal{P}}^t| L^2}{2} + \frac{\eta_{\boldsymbol{\theta}} |\boldsymbol{\mathcal{Q}}^{t+1}| L^2}{2} + \frac{8|\boldsymbol{\mathcal{P}}^t| L^2}{\eta_\lambda (c_1^t)^2} + \frac{8NL^2}{\eta_{\boldsymbol{\theta}} (c_2^t)^2} \right) \sum_{i=1}^{N} ||\boldsymbol{y}_i^{t+1} - \boldsymbol{y}_i^t||^2$$

$$+ \left( \frac{L+a_5}{2} - \frac{1}{\eta_{\boldsymbol{v}}^t} + \frac{\eta_\lambda |\boldsymbol{\mathcal{P}}^t| L^2}{2} + \frac{\eta_{\boldsymbol{\theta}} |\boldsymbol{\mathcal{Q}}^{t+1}| L^2}{2} + \frac{8|\boldsymbol{\mathcal{P}}^t| L^2}{\eta_\lambda (c_1^t)^2} + \frac{8NL^2}{\eta_{\boldsymbol{\theta}} (c_2^t)^2} \right) ||\boldsymbol{v}^{t+1} - \boldsymbol{v}^t||^2$$

$$+ \left( \frac{L+a_5}{2} - \frac{1}{\eta_{\boldsymbol{z}}^t} + \frac{\eta_\lambda |\boldsymbol{\mathcal{P}}^t| L^2}{2} + \frac{\eta_{\boldsymbol{\theta}} |\boldsymbol{\mathcal{Q}}^{t+1}| L^2}{2} + \frac{8|\boldsymbol{\mathcal{P}}^t| L^2}{\eta_\lambda (c_1^t)^2} + \frac{8NL^2}{\eta_{\boldsymbol{\theta}} (c_2^t)^2} \right) ||\boldsymbol{z}^{t+1} - \boldsymbol{z}^t||^2 \qquad (121)$$

$$- \left( \frac{1}{10\eta_\lambda} - \frac{6\tau k_1 N L^2}{2} \right) \sum_{l=1}^{|\boldsymbol{\mathcal{P}}^t|} ||\lambda_l^{t+1} - \lambda_l^t||^2 - \frac{1}{10\eta_{\boldsymbol{\theta}}} \sum_{i=1}^{N} ||\boldsymbol{\theta}_i^{t+1} - \boldsymbol{\theta}_i^t||^2 + \frac{c_1^{t-1} - c_1^t}{2} \sum_{l=1}^{|\boldsymbol{\mathcal{P}}^t|} ||\lambda_l^{t+1}||^2$$

$$+ \frac{c_2^{t-1} - c_2^t}{2} \sum_{i=1}^{N} ||\boldsymbol{\theta}_i^{t+1}||^2 + \frac{4}{\eta_\lambda} \left( \frac{c_1^{t-2}}{c_1^{t-1}} - \frac{c_1^{t-1}}{c_1^t} \right) \sum_{l=1}^{|\boldsymbol{\mathcal{P}}^t|} ||\lambda_l^t||^2 + \frac{4}{\eta_{\boldsymbol{\theta}}} \left( \frac{c_2^{t-2}}{c_2^{t-1}} - \frac{c_2^{t-1}}{c_2^t} \right) \sum_{i=1}^{N} ||\boldsymbol{\theta}_i^t||^2.$$

which concludes the proof of Lemma 3.

***Proof of Theorem 1:***

First, we set that,

$$a_6^t = \frac{4|\boldsymbol{\mathcal{P}}^t|(\gamma - 2)L^2}{\eta_\lambda (c_1^t)^2} + \frac{4N(\gamma - 2)L^2}{\eta_{\boldsymbol{\theta}} (c_2^t)^2} + \frac{\eta_{\boldsymbol{\theta}} (N - |\boldsymbol{\mathcal{Q}}^{t+1}|)L^2}{2} - \frac{a_5}{2}, \qquad (122)$$

where constant $\gamma$ satisfies that $\gamma > 2$ and $\frac{4(\gamma-2)L^2}{\eta_\lambda (c_1^0)^2} + \frac{4N(\gamma-2)L^2}{\eta_{\boldsymbol{\theta}} (c_2^0)^2} > \frac{a_5}{2}$, thus we have that $a_6^t > 0, \forall t$.
According to the setting of $\eta_{\boldsymbol{x}}^t, \eta_{\boldsymbol{y}}^t, \eta_{\boldsymbol{v}}^t, \eta_{\boldsymbol{z}}^t$ and $c_1^t, c_2^t$, we have,

$$\frac{L+a_5}{2} - \frac{1}{\eta_{\boldsymbol{x}}^t} + \frac{\eta_\lambda |\boldsymbol{\mathcal{P}}^t| L^2}{2} + \frac{\eta_{\boldsymbol{\theta}} |\boldsymbol{\mathcal{Q}}^{t+1}| L^2}{2} + \frac{8|\boldsymbol{\mathcal{P}}^t| L^2}{\eta_\lambda (c_1^t)^2} + \frac{8NL^2}{\eta_{\boldsymbol{\theta}} (c_2^t)^2} = -a_6^t, \qquad (123)$$

$$\frac{L+a_5}{2} - \frac{1}{\eta_{\boldsymbol{y}}^t} + \frac{\eta_\lambda |\boldsymbol{\mathcal{P}}^t| L^2}{2} + \frac{\eta_{\boldsymbol{\theta}} |\boldsymbol{\mathcal{Q}}^{t+1}| L^2}{2} + \frac{8|\boldsymbol{\mathcal{P}}^t| L^2}{\eta_\lambda (c_1^t)^2} + \frac{8NL^2}{\eta_{\boldsymbol{\theta}} (c_2^t)^2} = -a_6^t, \qquad (124)$$

$$\frac{L+a_5}{2} - \frac{1}{\eta_{\boldsymbol{v}}^t} + \frac{\eta_\lambda |\boldsymbol{\mathcal{P}}^t| L^2}{2} + \frac{\eta_{\boldsymbol{\theta}} |\boldsymbol{\mathcal{Q}}^{t+1}| L^2}{2} + \frac{8|\boldsymbol{\mathcal{P}}^t| L^2}{\eta_\lambda (c_1^t)^2} + \frac{8NL^2}{\eta_{\boldsymbol{\theta}} (c_2^t)^2} = -a_6^t, \qquad (125)$$

$$\frac{L+a_5}{2} - \frac{1}{\eta_{\boldsymbol{z}}^t} + \frac{\eta_\lambda |\boldsymbol{\mathcal{P}}^t| L^2}{2} + \frac{\eta_{\boldsymbol{\theta}} |\boldsymbol{\mathcal{Q}}^{t+1}| L^2}{2} + \frac{8|\boldsymbol{\mathcal{P}}^t| L^2}{\eta_\lambda (c_1^t)^2} + \frac{8NL^2}{\eta_{\boldsymbol{\theta}} (c_2^t)^2} = -a_6^t. \qquad (126)$$

Combining Eq. (123), (124), (125), (126) with Lemma 3, $\forall t \geq T_1$, we can obtain that,

$$a_6^t \sum_{i=1}^{N} \left( ||\boldsymbol{x}_i^{t+1} - \boldsymbol{x}_i^t||^2 + ||\boldsymbol{y}_i^{t+1} - \boldsymbol{y}_i^t||^2 \right) + a_6^t ||\boldsymbol{v}^{t+1} - \boldsymbol{v}^t||^2 + a_6^t ||\boldsymbol{z}^{t+1} - \boldsymbol{z}^t||^2$$

$$+ \left( \frac{1}{10\eta_\lambda} - \frac{6\tau k_1 N L^2}{2} \right) \sum_{l=1}^{|\boldsymbol{\mathcal{P}}^t|} ||\lambda_l^{t+1} - \lambda_l^t||^2 + \frac{1}{10\eta_{\boldsymbol{\theta}}} \sum_{i=1}^{N} ||\boldsymbol{\theta}_i^{t+1} - \boldsymbol{\theta}_i^t||^2$$

$$\leq F^t - F^{t+1} + \frac{c_1^{t-1} - c_1^t}{2} \sum_{l=1}^{|\boldsymbol{\mathcal{P}}^t|} ||\lambda_l^{t+1}||^2 + \frac{c_2^{t-1} - c_2^t}{2} \sum_{i=1}^{N} ||\boldsymbol{\theta}_i^{t+1}||^2 \qquad (127)$$

$$+ \frac{4}{\eta_\lambda} \left( \frac{c_1^{t-2}}{c_1^{t-1}} - \frac{c_1^{t-1}}{c_1^t} \right) \sum_{l=1}^{|\boldsymbol{\mathcal{P}}^t|} ||\lambda_l^t||^2 + \frac{4}{\eta_{\boldsymbol{\theta}}} \left( \frac{c_2^{t-2}}{c_2^{t-1}} - \frac{c_2^{t-1}}{c_2^t} \right) \sum_{i=1}^{N} ||\boldsymbol{\theta}_i^t||^2.$$

Utilizing the definition of $(\nabla \widetilde{G}^t)_{\boldsymbol{x}_i}$ and combining it with trigonometric inequality, Cauchy-Schwarz inequality and Assumption 1 and 2, we can obtain that,

$$||(\nabla \widetilde{G}^t)_{\boldsymbol{x}_i}||^2 \leq \frac{2}{\eta_{\boldsymbol{x}}^2} ||\boldsymbol{x}_i^{\overline{t_i}} - \boldsymbol{x}_i^t||^2 + 6L^2 \tau k_1 \left( ||\boldsymbol{v}^{t+1} - \boldsymbol{v}^t||^2 + ||\boldsymbol{z}^{t+1} - \boldsymbol{z}^t||^2 + \sum_{l=1}^{|\boldsymbol{\mathcal{P}}^t|} ||\lambda_l^{t+1} - \lambda_l^t||^2 \right).$$
$$(128)$$

Utilizing the definition of $(\nabla \widetilde{G}^t)_{\boldsymbol{y}_i}$ and combining it with trigonometric inequality and Cauchy-Schwarz inequality, it follows that,

$$
\begin{aligned}
&||(\nabla \widetilde{G}^t)_{\boldsymbol{y}_i}||^2 \\
&\leq \tfrac{2}{\eta_{\boldsymbol{y}}^2}||\boldsymbol{y}_i^{\overline{t_i}}-\boldsymbol{y}_i^t||^2+6L^2\tau k_1(||\boldsymbol{v}^{t+1}-\boldsymbol{v}^t||^2+||\boldsymbol{z}^{t+1}-\boldsymbol{z}^t||^2+\sum_{l=1}^{|\boldsymbol{\mathcal{P}}^t|}||\lambda_l^{t+1}-\lambda_l^t||^2).
\end{aligned}
\tag{129}
$$

Utilizing the definition of $(\nabla \widetilde{G}^t)_{\boldsymbol{v}}$ and combining it with trigonometric inequality and Cauchy-Schwarz inequality, we have that,

$$
||(\nabla \widetilde{G}^t)_{\boldsymbol{v}}||^2 \leq 2L^2\sum_{i=1}^N(||\boldsymbol{x}_i^{t+1}-\boldsymbol{x}_i^t||^2+||\boldsymbol{y}_i^{t+1}-\boldsymbol{y}_i^t||^2)+\tfrac{2}{\eta_{\boldsymbol{v}}^2}||\boldsymbol{v}^{t+1}-\boldsymbol{v}^t||^2.
\tag{130}
$$

Using the definition of $(\nabla \widetilde{G}^t)_{\boldsymbol{z}}$ and combining it with trigonometric inequality and Cauchy-Schwarz inequality, it follows that,

$$
||(\nabla \widetilde{G}^t)_{\boldsymbol{z}}||^2 \leq 2L^2\left(\sum_{i=1}^N(||\boldsymbol{x}_i^{t+1}-\boldsymbol{x}_i^t||^2+||\boldsymbol{y}_i^{t+1}-\boldsymbol{y}_i^t||^2)+||\boldsymbol{v}^{t+1}-\boldsymbol{v}^t||^2\right)+\tfrac{2}{\eta_{\boldsymbol{z}}^2}||\boldsymbol{z}^{t+1}-\boldsymbol{z}^t||^2.
\tag{131}
$$

Using the definition of $(\nabla \widetilde{G}^t)_{\lambda_l}$ and combining it with trigonometric inequality and Cauchy-Schwarz inequality, we can obtain the following inequality,

$$
\begin{aligned}
||(\nabla \widetilde{G}^t)_{\lambda_l}||^2 \leq{}& \tfrac{3}{\eta_\lambda^2}||\lambda_l^{t+1}-\lambda_l^t||^2 + 3((c_1^{t-1})^2-(c_1^t)^2)||\lambda_l^t||^2 \\
&+3L^2\left(\sum_{i=1}^N(||\boldsymbol{x}_i^{t+1}-\boldsymbol{x}_i^t||^2+||\boldsymbol{y}_i^{t+1}-\boldsymbol{y}_i^t||^2)+||\boldsymbol{v}^{t+1}-\boldsymbol{v}^t||^2+||\boldsymbol{z}^{t+1}-\boldsymbol{z}^t||^2\right).
\end{aligned}
\tag{132}
$$

Combining the definition of $(\nabla \widetilde{G}^t)_{\boldsymbol{\theta}_i}$ with Cauchy-Schwarz inequality and Assumption 2, we have,

$$
\begin{aligned}
&||(\nabla \widetilde{G}^t)_{\boldsymbol{\theta}_i}||^2 \\
&\leq \tfrac{3}{\eta_{\boldsymbol{\theta}}^2}||\boldsymbol{\theta}_i^{\overline{t_i}}-\boldsymbol{\theta}_i^t||^2+3L^2\left(\sum_{i=1}^N(||\boldsymbol{x}_i^{\overline{t_i}}-\boldsymbol{x}_i^t||^2+||\boldsymbol{y}_i^{\overline{t_i}}-\boldsymbol{y}_i^t||^2)+||\boldsymbol{v}^{\overline{t_i}}-\boldsymbol{v}^t||^2\right)+3(c_2^{\hat{t}_i-1}-c_2^{\overline{t_i}-1})^2||\boldsymbol{\theta}_i^t||^2 \\
&\leq \tfrac{3}{\eta_{\boldsymbol{\theta}}^2}||\boldsymbol{\theta}_i^{\overline{t_i}}-\boldsymbol{\theta}_i^t||^2+3L^2\sum_{i=1}^N(||\boldsymbol{x}_i^{\overline{t_i}}-\boldsymbol{x}_i^t||^2+||\boldsymbol{y}_i^{\overline{t_i}}-\boldsymbol{y}_i^t||^2) \\
&\quad+3L^2\tau k_1(||\boldsymbol{v}^{t+1}-\boldsymbol{v}^t||^2+||\boldsymbol{z}^{t+1}-\boldsymbol{z}^t||^2+\sum_{l=1}^{|\boldsymbol{\mathcal{P}}^t|}||\lambda_l^{t+1}-\lambda_l^t||^2) + 3((c_2^{\hat{t}_i-1})^2-(c_2^{\overline{t_i}-1})^2)||\boldsymbol{\theta}_i^t||^2.
\end{aligned}
\tag{133}
$$

In sight of the Definition B.2 as well as Eq. (128), (129), (130), (131), (132) and Eq. (133), we can obtain that,

$$
\begin{aligned}
&||\nabla \widetilde{G}^t||^2 \\
&= \sum_{i=1}^N(||(\nabla \widetilde{G}^t)_{\boldsymbol{x}_i}||^2+||(\nabla \widetilde{G}^t)_{\boldsymbol{y}_i}||^2+||(\nabla \widetilde{G}^t)_{\boldsymbol{\theta}_i}||^2)+||(\nabla \widetilde{G}^t)_{\boldsymbol{v}}||^2+||(\nabla \widetilde{G}^t)_{\boldsymbol{z}}||^2+\sum_{l=1}^{|\boldsymbol{\mathcal{P}}^t|}||(\nabla \widetilde{G}^t)_{\lambda_l}||^2 \\
&\leq (\tfrac{2}{\eta_{\boldsymbol{x}}^2}+3NL^2)\sum_{i=1}^N||\boldsymbol{x}_i^{\overline{t_i}}-\boldsymbol{x}_i^t||^2+(\tfrac{2}{\eta_{\boldsymbol{y}}^2}+3NL^2)\sum_{i=1}^N||\boldsymbol{y}_i^{\overline{t_i}}-\boldsymbol{y}_i^t||^2 \\
&\quad+(4+3|\boldsymbol{\mathcal{P}}^t|L^2)\sum_{i=1}^N||\boldsymbol{x}_i^{t+1}-\boldsymbol{x}_i^t||^2 + (4+3|\boldsymbol{\mathcal{P}}^t|L^2)\sum_{i=1}^N||\boldsymbol{y}_i^{t+1}-\boldsymbol{y}_i^t||^2 \\
&\quad+(\tfrac{2}{\eta_{\boldsymbol{v}}^2}+(2+15\tau k_1 N+3|\boldsymbol{\mathcal{P}}^t|)L^2)||\boldsymbol{v}^{t+1}-\boldsymbol{v}^t||^2+(\tfrac{2}{\eta_{\boldsymbol{z}}^2}+(15\tau k_1 N+3|\boldsymbol{\mathcal{P}}^t|)L^2)||\boldsymbol{z}^{t+1}-\boldsymbol{z}^t||^2 \\
&\quad+\sum_{l=1}^{|\boldsymbol{\mathcal{P}}^t|}(\tfrac{3}{\eta_\lambda^2}+15\tau k_1 NL^2)||\lambda_l^{t+1}-\lambda_l^t||^2+\sum_{l=1}^{|\boldsymbol{\mathcal{P}}^t|}3((c_1^{t-1})^2-(c_1^t)^2)||\lambda_l^t||^2 \\
&\quad+\sum_{i=1}^N\tfrac{3}{\eta_{\boldsymbol{\theta}}^2}||\boldsymbol{\theta}_i^{\overline{t_i}}-\boldsymbol{\theta}_i^t||^2+\sum_{i=1}^N 3((c_2^{\hat{t}_i-1})^2-(c_2^{\overline{t_i}-1})^2)||\boldsymbol{\theta}_i^t||^2.
\end{aligned}
\tag{134}
$$

Let constant $\underline{a_6}$ denote the lower bound of $a_6^t$ ($\underline{a_6} > 0$), and we set constants $d_1, d_2, d_3, d_4$ that,

$$d_1 = \frac{2k_\tau\tau + (4+3M+3k_\tau\tau N)L^2\eta_{\boldsymbol{x}}^2}{\eta_{\boldsymbol{x}}^2(\underline{a_6})^2} \geq \frac{2k_\tau\tau + (4+3|\boldsymbol{\mathcal{P}}^t|+3k_\tau\tau N)L^2\eta_{\boldsymbol{x}}^2}{\eta_{\boldsymbol{x}}^2(a_6^t)^2}, \tag{135}$$

$$d_2 = \frac{2k_\tau\tau + (4+3M+3k_\tau\tau N)L^2\eta_{\boldsymbol{y}}^2}{\eta_{\boldsymbol{y}}^2(\underline{a_6})^2} \geq \frac{2k_\tau\tau + (4+3|\boldsymbol{\mathcal{P}}^t|+3k_\tau\tau N)L^2\eta_{\boldsymbol{y}}^2}{\eta_{\boldsymbol{y}}^2(a_6^t)^2}, \tag{136}$$

$$d_3 = \frac{2+(2+15\tau k_1 N+3M)L^2\eta_{\boldsymbol{v}}^2}{\eta_{\boldsymbol{v}}^2(\underline{a_6})^2} \geq \frac{2+(2+15\tau k_1 N+3|\boldsymbol{\mathcal{P}}^t|)L^2\eta_{\boldsymbol{v}}^2}{\eta_{\boldsymbol{v}}^2(a_6^t)^2}, \tag{137}$$

$$d_4 = \frac{2+(15\tau k_1 N+3M)L^2\eta_{\boldsymbol{z}}^2}{\eta_{\boldsymbol{z}}^2(\underline{a_6})^2} \geq \frac{2+(15\tau k_1 N+3|\boldsymbol{\mathcal{P}}^t|)L^2\eta_{\boldsymbol{z}}^2}{\eta_{\boldsymbol{z}}^2(a_6^t)^2}, \tag{138}$$

where $k_\tau$ is a positive constant. Thus, combining Eq. (134) with Eq. (135), Eq. (136), (137), (138), we can obtain,

$$\begin{aligned}
||\nabla\widetilde{G}^t||^2 \leq &\sum_{i=1}^{N} d_1(a_6^t)^2||\boldsymbol{x}_i^{t+1}-\boldsymbol{x}_i^t||^2 + \sum_{i=1}^{N} d_2(a_6^t)^2||\boldsymbol{y}_i^{t+1}-\boldsymbol{y}_i^t||^2 \\
&+d_3(a_6^t)^2||\boldsymbol{v}^{t+1}-\boldsymbol{v}^t||^2+d_4(a_6^t)^2||\boldsymbol{z}^{t+1}-\boldsymbol{z}^t||^2+\sum_{i=1}^{N}\frac{3}{\eta_{\boldsymbol{\theta}}^2}||\boldsymbol{\theta}_i^{\overline{t_i}}-\boldsymbol{\theta}_i^t||^2 \\
&+\sum_{l=1}^{|\boldsymbol{\mathcal{P}}^t|}(\frac{3}{\eta_\lambda^2}+15\tau k_1 NL^2)||\lambda_l^{t+1}-\lambda_l^t||^2+\sum_{l=1}^{|\boldsymbol{\mathcal{P}}^t|}3((c_1^{t-1})^2-(c_1^t)^2)||\lambda_l^t||^2 \\
&+\sum_{i=1}^{N}3((c_2^{\hat{t}_i-1})^2-(c_2^{\overline{t_i}-1})^2)||\boldsymbol{\theta}_i^t||^2+(\frac{2}{\eta_{\boldsymbol{x}}^2}+3NL^2)\sum_{i=1}^{N}||\boldsymbol{x}_i^{\overline{t_i}}-\boldsymbol{x}_i^t||^2 \\
&-(\frac{2k_\tau\tau}{\eta_{\boldsymbol{x}}^2}+3k_\tau\tau NL^2)\sum_{i=1}^{N}||\boldsymbol{x}_i^{t+1}-\boldsymbol{x}_i^t||^2+(\frac{2}{\eta_{\boldsymbol{y}}^2}+3NL^2)\sum_{i=1}^{N}||\boldsymbol{y}_i^{\overline{t_i}}-\boldsymbol{y}_i^t||^2 \\
&-(\frac{2k_\tau\tau}{\eta_{\boldsymbol{y}}^2}+3k_\tau\tau NL^2)\sum_{i=1}^{N}||\boldsymbol{y}_i^{t+1}-\boldsymbol{y}_i^t||^2.
\end{aligned} \tag{139}$$

Let $d_5^t$ denote a nonnegative sequence, i.e., $d_5^t = \frac{1}{\max\{d_1 a_6^t, d_2 a_6^t, d_3 a_6^t, d_4 a_6^t, \frac{\frac{30}{\eta_\lambda}+150\eta_\lambda\tau k_1 NL^2}{1-30\eta_\lambda\tau k_1 NL^2}, \frac{30\tau}{\eta_{\boldsymbol{\theta}}}\}}$. We denote the upper and lower bound of $d_5^t$ as $\overline{d_5}$ and $\underline{d_5}$, respectively. And we set the constant $k_\tau$ satisfies $k_\tau \geq \max\{\frac{\overline{d_5}(\frac{2}{\eta_{\boldsymbol{y}}^2}+3NL^2)}{\underline{d_5}(\frac{2}{\eta_{\boldsymbol{y}}^2}+3NL^2)}, \frac{\overline{d_5}(\frac{2}{\eta_{\boldsymbol{x}}^2}+3NL^2)}{\underline{d_5}(\frac{2}{\eta_{\boldsymbol{x}}^2}+3NL^2)}\}$, where $\overline{\eta_{\boldsymbol{x}}}$ and $\overline{\eta_{\boldsymbol{y}}}$ are the upper bounds of $\eta_{\boldsymbol{x}}^t$ and $\eta_{\boldsymbol{y}}^t$, respectively. We can obtain the following inequality by combining Eq. (139) with the definition of $d_5^t$:

$$\begin{aligned}
d_5^t||\nabla\widetilde{G}^t||^2 \leq &a_6^t\sum_{i=1}^{N}(||\boldsymbol{x}_i^{t+1}-\boldsymbol{x}_i^t||^2+||\boldsymbol{y}_i^{t+1}-\boldsymbol{y}_i^t||^2)+a_6^t||\boldsymbol{v}^{t+1}-\boldsymbol{v}^t||^2+a_6^t||\boldsymbol{z}^{t+1}-\boldsymbol{z}^t||^2 \\
&+(\frac{1}{10\eta_\lambda}-\frac{6\tau k_1 NL^2}{2})\sum_{l=1}^{|\boldsymbol{\mathcal{P}}^t|}||\lambda_l^{t+1}-\lambda_l^t||^2+\frac{1}{10\tau\eta_{\boldsymbol{\theta}}}\sum_{i=1}^{N}||\boldsymbol{\theta}_i^{\overline{t_i}}-\boldsymbol{\theta}_i^t||^2 \\
&+\sum_{l=1}^{|\boldsymbol{\mathcal{P}}^t|}3d_5^t((c_1^{t-1})^2-(c_1^t)^2)||\lambda_l^t||^2+\sum_{i=1}^{N}3d_5^t((c_2^{\hat{t}_i-1})^2-(c_2^{\overline{t_i}-1})^2)||\boldsymbol{\theta}_i^t||^2 \\
&+d_5^t(\frac{2}{\eta_{\boldsymbol{x}}^2}+3NL^2)\sum_{i=1}^{N}||\boldsymbol{x}_i^{\overline{t_i}}-\boldsymbol{x}_i^t||^2-d_5^t(\frac{2k_\tau\tau}{\eta_{\boldsymbol{x}}^2}+3k_\tau\tau NL^2)\sum_{i=1}^{N}||\boldsymbol{x}_i^{t+1}-\boldsymbol{x}_i^t||^2 \\
&+d_5^t(\frac{2}{\eta_{\boldsymbol{y}}^2}+3NL^2)\sum_{i=1}^{N}||\boldsymbol{y}_i^{\overline{t_i}}-\boldsymbol{y}_i^t||^2-d_5^t(\frac{2k_\tau\tau}{\eta_{\boldsymbol{y}}^2}+3k_\tau\tau NL^2)\sum_{i=1}^{N}||\boldsymbol{y}_i^{t+1}-\boldsymbol{y}_i^t||^2.
\end{aligned} \tag{140}$$

Combining the definition of $d_5^t$ with Eq. (127) and according to the setting $||\lambda_l^t||^2 \leq \alpha_3, ||\boldsymbol{\theta}_i^t||^2 \leq \alpha_4$ and $\overline{d_5} \geq d_5^t \geq \underline{d_5}, \forall t \geq T_1$, thus, we have,

$$
\begin{aligned}
&d_5^t ||\nabla \widetilde{G}^t||^2 \\
&\leq F^t - F^{t+1} + \frac{c_1^{t-1} - c_1^t}{2} M\alpha_3 + \frac{c_2^{t-1} - c_2^t}{2} N\alpha_4 + \frac{4}{\eta_\lambda}(\frac{c_1^{t-2}}{c_1^{t-1}} - \frac{c_1^{t-1}}{c_1^t}) M\alpha_3 \\
&\quad + \frac{4}{\eta_{\boldsymbol{\theta}}}(\frac{c_2^{t-2}}{c_2^{t-1}} - \frac{c_2^{t-1}}{c_2^t}) N\alpha_4 + 3\overline{d_5}((c_1^{t-1})^2 - (c_1^t)^2) M\alpha_3 + 3\overline{d_5}\sum_{i=1}^N ((c_2^{\hat{t}_i-1})^2 - (c_2^{\overline{t}_i-1})^2)\alpha_4 \\
&\quad + \frac{1}{10\tau\eta_{\boldsymbol{\theta}}}\sum_{i=1}^N ||\boldsymbol{\theta}_i^{\overline{t}_i} - \boldsymbol{\theta}_i^t||^2 - \frac{1}{10\eta_{\boldsymbol{\theta}}}\sum_{i=1}^N ||\boldsymbol{\theta}_i^{t+1} - \boldsymbol{\theta}_i^t||^2 \\
&\quad + \overline{d_5}(\frac{2}{\eta_{\boldsymbol{x}}^2} + 3NL^2)\sum_{i=1}^N ||\boldsymbol{x}_i^{\overline{t}_i} - \boldsymbol{x}_i^t||^2 - \underline{d_5}(\frac{2k_\tau\tau}{\eta_{\boldsymbol{x}}^2} + 3k_\tau\tau NL^2)\sum_{i=1}^N ||\boldsymbol{x}_i^{t+1} - \boldsymbol{x}_i^t||^2 \\
&\quad + \overline{d_5}(\frac{2}{\eta_{\boldsymbol{y}}^2} + 3NL^2)\sum_{i=1}^N ||\boldsymbol{y}_i^{\overline{t}_i} - \boldsymbol{y}_i^t||^2 - \underline{d_5}(\frac{2k_\tau\tau}{\eta_{\boldsymbol{y}}^2} + 3k_\tau\tau NL^2)\sum_{i=1}^N ||\boldsymbol{y}_i^{t+1} - \boldsymbol{y}_i^t||^2.
\end{aligned}
\tag{141}
$$

Denoting $\widetilde{T}(\epsilon)$ as $\widetilde{T}(\epsilon) = \min\{t \mid ||\nabla \widetilde{G}^{T_1+t}||^2 \leq \frac{\epsilon}{4}, t \geq 2\}$. Summing up Eq. (141) from $t = T_1+2$ to $t = T_1 + \widetilde{T}(\epsilon)$, we have,

$$
\begin{aligned}
&\sum_{t=T_1+2}^{T_1+\widetilde{T}(\epsilon)} d_5^t ||\nabla \widetilde{G}^t||^2 \\
&\leq F^{T_1+2} - \underline{L} + \frac{4}{\eta_\lambda}(\frac{c_1^0}{c_1^1} + \frac{c_1^1}{c_1^2}) M\alpha_3 + \frac{c_1^1}{2} M\alpha_3 + \frac{7}{2\eta_\lambda} M\sigma_3^2 + 3\overline{d_5}(c_1^1)^2 M\alpha_3 \\
&\quad + \frac{4}{\eta_{\boldsymbol{\theta}}}(\frac{c_2^0}{c_2^1} + \frac{c_2^1}{c_2^2}) N\alpha_4 + \frac{c_2^1}{2} N\alpha_4 + \frac{7}{2\eta_{\boldsymbol{\theta}}} N\sigma_4^2 + \sum_{i=1}^N \sum_{t=T_1+2}^{T_1+\widetilde{T}(\epsilon)} 3\overline{d_5}((c_2^{\hat{t}_i-1})^2 - (c_2^{\overline{t}_i-1})^2)\alpha_4 \\
&\quad + \frac{c_1^{T_1+2}}{2} M\sigma_3^2 + \frac{c_2^{T_1+2}}{2} N\sigma_4^2 + \frac{1}{10\tau\eta_{\boldsymbol{\theta}}}\sum_{t=T_1+2}^{T_1+\widetilde{T}(\epsilon)}\sum_{i=1}^N ||\boldsymbol{\theta}_i^{\overline{t}_i} - \boldsymbol{\theta}_i^t||^2 - \frac{1}{10\eta_{\boldsymbol{\theta}}}\sum_{t=T_1+2}^{T_1+\widetilde{T}(\epsilon)}\sum_{i=1}^N ||\boldsymbol{\theta}_i^{t+1} - \boldsymbol{\theta}_i^t||^2 \\
&\quad + \overline{d_5}(\frac{2}{\eta_{\boldsymbol{x}}^2} + 3NL^2)\sum_{t=T_1+2}^{T_1+\widetilde{T}(\epsilon)}\sum_{i=1}^N ||\boldsymbol{x}_i^{\overline{t}_i} - \boldsymbol{x}_i^t||^2 - \underline{d_5}(\frac{2k_\tau\tau}{\eta_{\boldsymbol{x}}^2} + 3k_\tau\tau NL^2)\sum_{t=T_1+2}^{T_1+\widetilde{T}(\epsilon)}\sum_{i=1}^N ||\boldsymbol{x}_i^{t+1} - \boldsymbol{x}_i^t||^2 \\
&\quad + \overline{d_5}(\frac{2}{\eta_{\boldsymbol{y}}^2} + 3NL^2)\sum_{t=T_1+2}^{T_1+\widetilde{T}(\epsilon)}\sum_{i=1}^N ||\boldsymbol{y}_i^{\overline{t}_i} - \boldsymbol{y}_i^t||^2 - \underline{d_5}(\frac{2k_\tau\tau}{\eta_{\boldsymbol{y}}^2} + 3k_\tau\tau NL^2)\sum_{t=T_1+2}^{T_1+\widetilde{T}(\epsilon)}\sum_{i=1}^N ||\boldsymbol{y}_i^{t+1} - \boldsymbol{y}_i^t||^2,
\end{aligned}
\tag{142}
$$

where $\sigma_3 = \max\{||\lambda_1 - \lambda_2||\}, \sigma_4 = \max\{||\boldsymbol{\theta}_1 - \boldsymbol{\theta}_2||\}$ and $\underline{L} = \min L_p(\{\boldsymbol{x}_i^t\}, \{\boldsymbol{y}_i^t\}, \boldsymbol{v}^t, \boldsymbol{z}^t, \{\lambda_l^t\}, \{\boldsymbol{\theta}_i^t\})$, which satisfy that, $\forall t \geq T_1 + 2$,

$$
F^t \geq \underline{L} - \frac{4}{\eta_\lambda}\frac{c_1^1}{c_1^2} M\alpha_3 - \frac{4}{\eta_{\boldsymbol{\theta}}}\frac{c_2^1}{c_2^2} N\alpha_4 - \frac{7}{2\eta_\lambda} M\sigma_3^2 - \frac{7}{2\eta_{\boldsymbol{\theta}}} N\sigma_4^2 - \frac{c_1^{T_1+2}}{2} M\sigma_3^2 - \frac{c_2^{T_1+2}}{2} N\sigma_4^2. \tag{143}
$$

For each worker $i$, we have that $\overline{t}_i - \hat{t}_i \leq \tau$, thus,

$$
\begin{aligned}
&\sum_{t=T_1+2}^{T_1+\widetilde{T}(\epsilon)} 3\overline{d_5}((c_2^{\hat{t}_i-1})^2 - (c_2^{\overline{t}_i-1})^2)\alpha_4 \\
&\leq \tau \sum_{\substack{\hat{v}_i(j) \in \mathcal{V}_i(\widetilde{T}(\epsilon)), \\ T_1+2 \leq \hat{v}_i(j) \leq T_1+\widetilde{T}(\epsilon)}} 3\overline{d_5}((c_2^{\hat{v}_i(j)-1})^2 - (c_2^{\hat{v}_i(j+1)-1})^2)\alpha_4 \\
&\leq 3\tau\overline{d_5}(c_2^1)^2\alpha_4.
\end{aligned}
\tag{144}
$$

Since the idle workers do not update their variables in each master iteration, for any $t$ that satisfies $\hat{v}_i(j-1) \leq t < \hat{v}_i(j)$, we have $\boldsymbol{\theta}_i^t = \boldsymbol{\theta}_i^{\hat{v}_i(j)-1}$. And for $t \notin \mathcal{V}_i(T)$, we have $||\boldsymbol{\theta}_i^t - \boldsymbol{\theta}_i^{t-1}||^2 = 0$.

Combining with $\hat{v}_i(j) - \hat{v}_i(j-1) \leq \tau$, we can obtain that,

$$
\begin{aligned}
\sum_{t=T_1+2}^{T_1+\widetilde{T}(\epsilon)} \sum_{i=1}^{N} ||\boldsymbol{\theta}_i^{\overline{t_j}} - \boldsymbol{\theta}_i^t||^2 \leq\; & \tau \sum_{\substack{\hat{v}_i(j)\in\mathcal{V}_i(\widetilde{T}(\epsilon)),\\ T_1+3\leq\hat{v}_i(j)}} \sum_{i=1}^{N} ||\boldsymbol{\theta}_i^{\hat{v}_i(j)} - \boldsymbol{\theta}_i^{\hat{v}_i(j)-1}||^2 \\
=\; & \tau \sum_{t=T_1+2}^{T_1+\widetilde{T}(\epsilon)} \sum_{i=1}^{N} ||\boldsymbol{\theta}_i^{t+1} - \boldsymbol{\theta}_i^t||^2 + \tau \sum_{t=\widetilde{T}(\epsilon)+1}^{\widetilde{T}(\epsilon)+\tau-1} \sum_{i=1}^{N} ||\boldsymbol{\theta}_i^{t+1} - \boldsymbol{\theta}_i^t||^2 \\
\leq\; & \tau \sum_{t=T_1+2}^{T_1+\widetilde{T}(\epsilon)} \sum_{i=1}^{N} ||\boldsymbol{\theta}_i^{t+1} - \boldsymbol{\theta}_i^t||^2 + 4\tau(\tau-1)N\alpha_4.
\end{aligned}
\tag{145}
$$

Similarly, for any $t$ that satisfies $\hat{v}_i(j-1) \leq t < \hat{v}_i(j)$, we have $\boldsymbol{x}_i^t = \boldsymbol{x}_i^{\hat{v}_i(j)-1}$, $\boldsymbol{y}_i^t = \boldsymbol{y}_i^{\hat{v}_i(j)-1}$. And for $t \notin \mathcal{V}_i(T)$, we have $||\boldsymbol{x}_i^t - \boldsymbol{x}_i^{t-1}||^2 = 0$, $||\boldsymbol{y}_i^t - \boldsymbol{y}_i^{t-1}||^2 = 0$. Combining with $\hat{v}_i(j) - \hat{v}_i(j-1) \leq \tau$, we can get that,

$$
\begin{aligned}
\sum_{t=T_1+2}^{T_1+\widetilde{T}(\epsilon)} \sum_{i=1}^{N} ||\boldsymbol{x}_i^{\overline{t_j}} - \boldsymbol{x}_i^t||^2 \leq\; & \tau \sum_{\substack{\hat{v}_i(j)\in\mathcal{V}_i(\widetilde{T}(\epsilon)),\\ T_1+3\leq\hat{v}_i(j)}} \sum_{i=1}^{N} ||\boldsymbol{x}_i^{\hat{v}_i(j)} - \boldsymbol{x}_i^{\hat{v}_i(j)-1}||^2 \\
=\; & \tau \sum_{t=T_1+2}^{T_1+\widetilde{T}(\epsilon)} \sum_{i=1}^{N} ||\boldsymbol{x}_i^{t+1} - \boldsymbol{x}_i^t||^2 + \tau \sum_{t=\widetilde{T}(\epsilon)+1}^{\widetilde{T}(\epsilon)+\tau-1} \sum_{i=1}^{N} ||\boldsymbol{x}_i^{t+1} - \boldsymbol{x}_i^t||^2 \\
\leq\; & \tau \sum_{t=T_1+2}^{T_1+\widetilde{T}(\epsilon)} \sum_{i=1}^{N} ||\boldsymbol{x}_i^{t+1} - \boldsymbol{x}_i^t||^2 + 4\tau(\tau-1)N\alpha_1.
\end{aligned}
\tag{146}
$$

$$
\begin{aligned}
\sum_{t=T_1+2}^{T_1+\widetilde{T}(\epsilon)} \sum_{i=1}^{N} ||\boldsymbol{y}_i^{\overline{t_j}} - \boldsymbol{y}_i^t||^2 \leq\; & \tau \sum_{\substack{\hat{v}_i(j)\in\mathcal{V}_i(\widetilde{T}(\epsilon)),\\ 3\leq\hat{v}_i(j)}} \sum_{i=1}^{N} ||\boldsymbol{y}_i^{\hat{v}_i(j)} - \boldsymbol{y}_i^{\hat{v}_i(j)-1}||^2 \\
=\; & \tau \sum_{t=T_1+2}^{T_1+\widetilde{T}(\epsilon)} \sum_{i=1}^{N} ||\boldsymbol{y}_i^{t+1} - \boldsymbol{y}_i^t||^2 + \tau \sum_{t=\widetilde{T}(\epsilon)+1}^{\widetilde{T}(\epsilon)+\tau-1} \sum_{i=1}^{N} ||\boldsymbol{y}_i^{t+1} - \boldsymbol{y}_i^t||^2 \\
\leq\; & \tau \sum_{t=T_1+2}^{T_1+\widetilde{T}(\epsilon)} \sum_{i=1}^{N} ||\boldsymbol{y}_i^{t+1} - \boldsymbol{y}_i^t||^2 + 4\tau(\tau-1)N\alpha_2.
\end{aligned}
\tag{147}
$$

It follows from Eq. (142), (144), (145), (146) that,

$$
\begin{aligned}
& \sum_{t=T_1+2}^{T_1+\widetilde{T}(\epsilon)} d_5^t ||\nabla\widetilde{G}^t||^2 \\
& \leq F^{T_1+2} - \underline{L} + \frac{4}{\eta_\lambda}\left(\frac{c_1^0}{c_1^1} + \frac{c_1^1}{c_1^2}\right)M\alpha_3 + \frac{c_1^1}{2}M\alpha_3 + \frac{7}{2\eta_\lambda}M\sigma_3^2 + 3\overline{d_5}(c_1^1)^2 M\alpha_3 \\
& + \frac{4}{\eta_\theta}\left(\frac{c_1^0}{c_1^1} + \frac{c_1^1}{c_1^2}\right)N\alpha_4 + \frac{c_2^1}{2}N\alpha_4 + \frac{7}{2\eta_\theta}N\sigma_4^2 + 3\tau\overline{d_5}(c_2^1)^2 N\alpha_4 + \frac{c_1^{T_1+2}}{2}M\sigma_3^2 + \frac{c_2^{T_1+2}}{2}N\sigma_4^2 \\
& + \left(\frac{2N\alpha_4}{5\eta_\theta} + 4\overline{d_5}\left(\frac{2}{\eta_{\boldsymbol{x}}^2} + 3NL^2\right)N\alpha_1\tau + 4\overline{d_5}\left(\frac{2}{\eta_{\boldsymbol{y}}^2} + 3NL^2\right)N\alpha_2\tau\right)(\tau-1) \\
& = \overline{d} + k_d\tau(\tau-1),
\end{aligned}
\tag{148}
$$

where $\overline{d}$ and $k_d$ are constants. Constant $d_6$ is given by,

$$
\begin{aligned}
d_6 &= \max\left\{d_1, d_2, d_3, d_4, \frac{\frac{30}{\eta_\lambda} + 150\eta_\lambda\tau k_1 NL^2}{(1-30\eta_\lambda\tau k_1 NL^2)\underline{a_6}}, \frac{30\tau}{\eta_\theta \underline{a_6}}\right\} \\
&\geq \max\left\{d_1, d_2, d_3, d_4, \frac{\frac{30}{\eta_\lambda} + 150\eta_\lambda\tau k_1 NL^2}{(1-30\eta_\lambda\tau k_1 NL^2)a_6^t}, \frac{30\tau}{\eta_\theta a_6^t}\right\} \\
&= \frac{1}{d_5^t a_6^t}.
\end{aligned}
\tag{149}
$$

Thus, we can obtain that,

$$\sum_{t=T_1+2}^{T_1+\widetilde{T}(\epsilon)} \frac{1}{d_6 a_6^t} ||\nabla \widetilde{G}^{T_1+\widetilde{T}(\epsilon)}||^2 \le \sum_{t=T_1+2}^{T_1+\widetilde{T}(\epsilon)} \frac{1}{d_6 a_6^t} ||\nabla \widetilde{G}^t||^2 \le \sum_{t=T_1+2}^{T_1+\widetilde{T}(\epsilon)} d_5^t ||\nabla \widetilde{G}^t||^2 \le \overline{d} + k_d \tau(\tau-1).$$
(150)

And it follows from Eq. (150) that,

$$||\nabla \widetilde{G}^{T_1+\widetilde{T}(\epsilon)}||^2 \le \frac{(\overline{d} + k_d \tau(\tau-1))d_6}{\sum_{t=T_1+2}^{T_1+\widetilde{T}(\epsilon)} \frac{1}{a_6^t}}.$$
(151)

According to the setting of $c_1^t$, $c_2^t$, we have,

$$\frac{1}{a_6^t} \ge \frac{1}{4(\gamma-2)L^2(M\eta_\lambda + N\eta_{\boldsymbol{\theta}})(t+1)^{\frac{1}{2}} + \frac{\eta_{\boldsymbol{\theta}}(N-S)L^2}{2}}.$$
(152)

Summing up $\frac{1}{a_6^t}$ from $t = T_1 + 2$ to $t = T_1 + \widetilde{T}(\epsilon)$, it follows that,

$$\begin{aligned}
\sum_{t=T_1+2}^{T_1+\widetilde{T}(\epsilon)} \frac{1}{a_6^t} &\ge \sum_{t=T_1+2}^{T_1+\widetilde{T}(\epsilon)} \frac{1}{4(\gamma-2)L^2(M\eta_\lambda+N\eta_{\boldsymbol{\theta}})(t+1)^{\frac{1}{2}}+\frac{\eta_{\boldsymbol{\theta}}(N-S)L^2}{2}} \\
&\ge \sum_{t=T_1+2}^{T_1+\widetilde{T}(\epsilon)} \frac{1}{4(\gamma-2)L^2(M\eta_\lambda+N\eta_{\boldsymbol{\theta}})(t+1)^{\frac{1}{2}}+\frac{\eta_{\boldsymbol{\theta}}(N-S)L^2}{2}(t+1)^{\frac{1}{2}}} \\
&\ge \frac{(T_1+\widetilde{T}(\epsilon))^{\frac{1}{2}}-(T_1+2)^{\frac{1}{2}}}{4(\gamma-2)L^2(M\eta_\lambda+N\eta_{\boldsymbol{\theta}})+\frac{\eta_{\boldsymbol{\theta}}(N-S)L^2}{2}}.
\end{aligned}$$
(153)

The second inequality in Eq. (153) is due to that $\forall t \ge T_1 + 2$, we have,

$$4(\gamma-2)L^2(M\eta_\lambda+N\eta_{\boldsymbol{\theta}})(t+1)^{\frac{1}{2}}+\frac{\eta_{\boldsymbol{\theta}}(N-S)L^2}{2} \le (4(\gamma-2)L^2(M\eta_\lambda+N\eta_{\boldsymbol{\theta}})+\frac{\eta_{\boldsymbol{\theta}}(N-S)L^2}{2})(t+1)^{\frac{1}{2}}.$$
(154)

The last inequality in Eq. (153) follows from the fact that $\sum_{t=T_1+2}^{T_1+\widetilde{T}(\epsilon)} \frac{1}{(t+1)^{\frac{1}{2}}} \ge (T_1+\widetilde{T}(\epsilon))^{\frac{1}{2}} - (T_1+2)^{\frac{1}{2}}$.

Thus, plugging Eq. (153) into Eq. (151), we can obtain:

$$||\nabla \widetilde{G}^{T_1+\widetilde{T}(\epsilon)}||^2 \le \frac{(\overline{d}+k_d\tau(\tau-1))d_6}{\sum_{t=T_1+2}^{T_1+\widetilde{T}(\epsilon)} \frac{1}{a_6^t}} \le \frac{(4(\gamma-2)L^2(M\eta_\lambda+N\eta_{\boldsymbol{\theta}})+\frac{\eta_{\boldsymbol{\theta}}(N-S)L^2}{2})(\overline{d}+k_d\tau(\tau-1))d_6}{(T_1+\widetilde{T}(\epsilon))^{\frac{1}{2}}-(T_1+2)^{\frac{1}{2}}}.$$
(155)

Let constant $d_7 = 4(\gamma-2)L^2(M\eta_\lambda+N\eta_{\boldsymbol{\theta}})$, and according to the definition of $\widetilde{T}(\epsilon)$, we have:

$$T_1 + \widetilde{T}(\epsilon) \ge \left(\frac{4(d_7 + \frac{\eta_{\boldsymbol{\theta}}(N-S)L^2}{2})(\overline{d}+k_d\tau(\tau-1))d_6}{\epsilon} + (T_1+2)^{\frac{1}{2}}\right)^2.$$
(156)

Combining the definition of $\nabla G^t$ and $\nabla \widetilde{G}^t$ with trigonometric inequality, we then get:

$$||\nabla G^t|| - ||\nabla \widetilde{G}^t|| \le ||\nabla G^t - \nabla \widetilde{G}^t|| \le \sqrt{\sum_{l=1}^{|\boldsymbol{\mathcal{P}}^t|} ||c_1^{t-1}\lambda_l^t||^2 + \sum_{i=1}^{N} ||c_2^{t-1}\boldsymbol{\theta}_i^t||^2}.$$
(157)

Table 2: Step-sizes of all variables in the experiments.

| Datasets | $\eta_{\boldsymbol{x}}$ | $\eta_{\boldsymbol{y}}$ | $\eta_{\boldsymbol{v}}$ | $\eta_{\boldsymbol{z}}$ | $\eta_{\lambda}$ | $\eta_{\boldsymbol{\theta}}$ |
|---|---|---|---|---|---|---|
| MNIST | 0.001 | 0.02 | 0.001 | 0.02 | 0.1 | 0.001 |
| Fashion MNIST | 0.001 | 0.02 | 0.001 | 0.02 | 0.1 | 0.001 |
| CIFAR-10 | 0.001 | 0.02 | 0.001 | 0.02 | 0.1 | 0.001 |
| Covertype | 0.01 | 0.02 | 0.01 | 0.02 | 0.1 | 0.01 |
| IJCNN1 | 0.01 | 0.005 | 0.01 | 0.005 | 0.1 | 0.01 |
| Australian | 0.001 | 0.02 | 0.001 | 0.02 | 5 | 0.001 |

If $t \geq (\frac{4M\alpha_3}{\eta_\lambda{}^2} + \frac{4N\alpha_4}{\eta_{\boldsymbol{\theta}}{}^2})^2 \frac{1}{\epsilon^2}$, then we have $\sqrt{\sum_{l=1}^{|\boldsymbol{\mathcal{P}}^t|} ||c_1^{t-1}\lambda_l^t||^2 + \sum_{i=1}^{N} ||c_2^{t-1}\boldsymbol{\theta}_i^t||^2} \leq \frac{\sqrt{\epsilon}}{2}$. Combining it with Eq. (156), we can conclude that there exists a

$$T(\epsilon) \sim \mathcal{O}(\max\{(\frac{4M\alpha_3}{\eta_\lambda{}^2} + \frac{4N\alpha_4}{\eta_{\boldsymbol{\theta}}{}^2})^2 \frac{1}{\epsilon^2}, (\frac{4(d_7 + \frac{\eta_{\boldsymbol{\theta}}(N-S)L^2}{2})(\bar{d} + k_d\tau(\tau-1))d_6}{\epsilon} + (T_1+2)^{\frac{1}{2}})^2\}),$$
(158)

such that $||\nabla G^t||^2 \leq \epsilon$, which concludes our proof.

## C  PROOF OF THEOREM 1

Assuming that there are cutting planes added every $k$ iteration, *i.e.*,

$$\boldsymbol{\mathcal{P}}^0 \supseteq \boldsymbol{\mathcal{P}}^k \supseteq \cdots \supseteq \boldsymbol{\mathcal{P}}^{nk}.$$
(159)

Let $\mathcal{R}^k$ denote the feasible region of problem in Eq. (12) in $k^{\text{th}}$ iteration, and let $\mathcal{R}'$ denote the feasible region of problem in Eq. (10), we have that,

$$\mathcal{R}^0 \supseteq \mathcal{R}^k \supseteq \cdots \supseteq \mathcal{R}^{nk} \supseteq \mathcal{R}'.$$
(160)

Let $F(\{\boldsymbol{x}_i^{k*}\}, \{\boldsymbol{y}_i^{k*}\}, \boldsymbol{v}^{k*}, \boldsymbol{z}^{k*})$ denote the optimal objective value of the problem in Eq. (12) in $k^{\text{th}}$ iteration and let $F^*$ denote the optimal objective value of the problem in Eq. (10). According to Eq. (160), we have that,

$$F(\{\boldsymbol{x}_i^{0*}\}, \{\boldsymbol{y}_i^{0*}\}, \boldsymbol{v}^{0*}, \boldsymbol{z}^{0*}) \leq F(\{\boldsymbol{x}_i^{k*}\}, \{\boldsymbol{y}_i^{k*}\}, \boldsymbol{v}^{k*}, \boldsymbol{z}^{k*}) \leq \cdots \leq F(\{\boldsymbol{x}_i^{nk*}\}, \{\boldsymbol{y}_i^{nk*}\}, \boldsymbol{v}^{nk*}, \boldsymbol{z}^{nk*}).$$
(161)

And we can obtain that,

$$\frac{F^*}{F(\{\boldsymbol{x}_i^{0*}\}, \{\boldsymbol{y}_i^{0*}\}, \boldsymbol{v}^{0*}, \boldsymbol{z}^{0*})} \geq \frac{F^*}{F(\{\boldsymbol{x}_i^{k*}\}, \{\boldsymbol{y}_i^{k*}\}, \boldsymbol{v}^{k*}, \boldsymbol{z}^{k*})} \geq \cdots \geq \frac{F^*}{F(\{\boldsymbol{x}_i^{nk*}\}, \{\boldsymbol{y}_i^{nk*}\}, \boldsymbol{v}^{nk*}, \boldsymbol{z}^{nk*})} \geq \beta.$$
(162)

It is seen from Eq. (162) that the sequence $\{\frac{F^*}{F(\{\boldsymbol{x}_i^{k*}\}, \{\boldsymbol{y}_i^{k*}\}, \boldsymbol{v}^{k*}, \boldsymbol{z}^{k*})}\}$ is monotonically non-increasing. When $nk \to \infty$, the optimal objective value of the problem in Eq. (12) monotonically converges to $\beta$ ($\beta \geq 1$).

# D  DETAILS OF EXPERIMENTS

## D.1  ADDITIONAL RESULTS

In this section, additional experiment results on CIFAR-10 (Krizhevsky et al., 2009) and Australian (Quinlan, 1987) datasets are reported in Figure 11 and Figure 12. It is seen from Figure 11 and Figure 12 that the proposed ADBO also achieves faster convergence rate.

## D.2  DETAILS OF EXPERIMENTS

In this section, we provide more details of the experimental setup in this work. In data hyper-cleaning task, experiments are carried out on MNIST, Fashion MNIST and CIFAR-10 datasets. Following (Ji et al., 2021), we utilize the same model in data-hypercleaning task for MNIST, Fashion MNIST and CIFAR-10 datasets, and SGD optimizer is utilized. And the step-sizes are summarized in Table 2. In MNIST and Fashion MNIST datasets, we set $N = 18$, $S = 9$, $\tau = 15$. And in CIFAR-10 dataset, we set $N = 18$, $S = 9$, $\tau = 5$. We set that the (communication + computation) delays of each worker obey log-normal distribution $\mathrm{LN}(3.5, 1)$.

In regularization coefficient optimization task, experiments are carried out on Covertype, IJCNN1 and Australian datasets. Following (Chen et al., 2022a), we utilize the same logistic regression model, and SGD optimizer is used. And the step-sizes are summarized in Table 2. In Covertype dataset, we set $N = 18$, $S = 9$, $\tau = 15$; in IJCNN1 dataset, we set $N = 24$, $S = 12$, $\tau = 15$; and in Australian dataset, we set $N = 4$, $S = 2$, $\tau = 5$. In the experiments that consider straggler problems, three stragglers are set in the distributed system, and the mean of (communication + computation) delay of stragglers is four times the delay of normal workers.

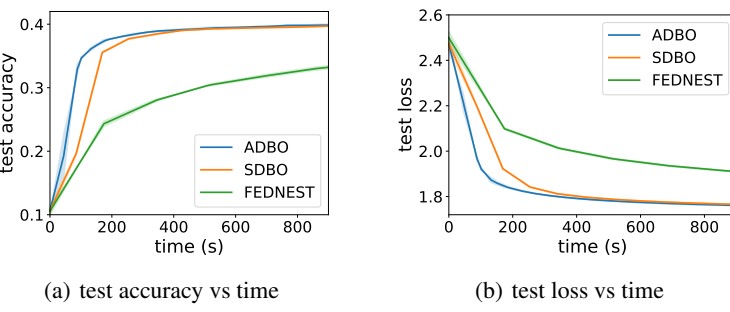

(a) test accuracy vs time        (b) test loss vs time

Figure 11: (a) Test accuracy vs time and (b) Test loss vs time on CIFAR-10 dataset on distributed data hyper-cleaning task.

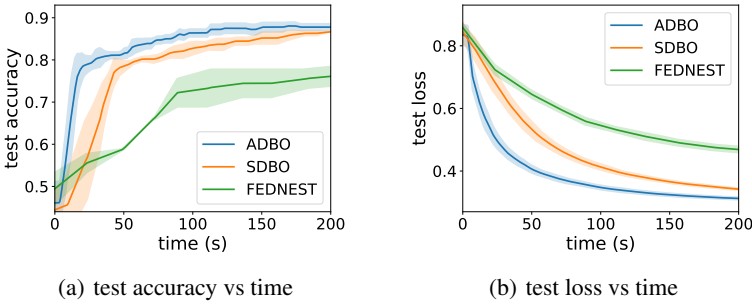

(a) test accuracy vs time        (b) test loss vs time

Figure 12: (a) Test accuracy vs time and (b) Test loss vs time on Australian dataset on distributed regularization coefficient optimization task.

Codes are available in `https://github.com/ICLR23Submission6251/adbo`.

# E    PARAMETER SERVER ARCHITECTURE

In this section, we give the illustration of the parameter server architecture, which is shown in Figure 13. In parameter server architecture, the communication is centralized around a set of master nodes (or servers) that constitute the hubs of a star network, and worker nodes (or clients) pull the shared parameters from and send their updates to the master nodes.

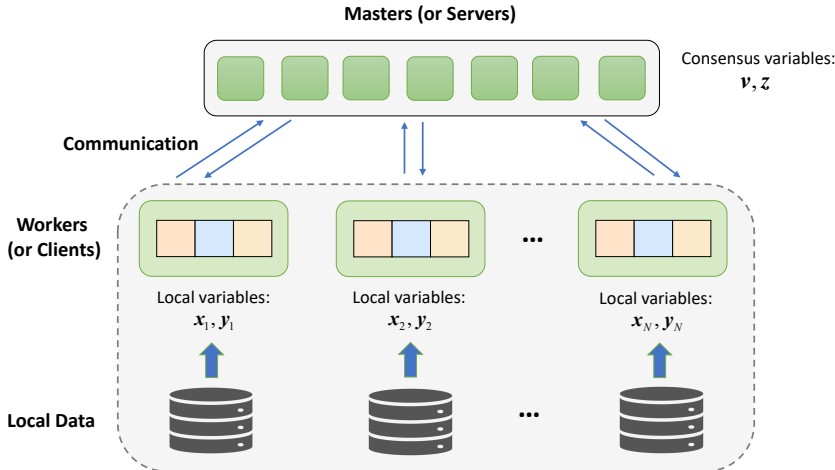

Figure 13: The illustration of parameter server architecture.

