# OpenReview forum: "Asynchronous Distributed Bilevel Optimization"
_ICLR.cc/2023/Conference — ICLR 2023 poster_

### Official Review · Reviewer_4Y6m · 2022-10-25

**Confidence:** 3
**Clarity, Quality, Novelty And Reproducibility:** Look good to me
**Correctness:** 3
**Technical Novelty And Significance:** 3
**Empirical Novelty And Significance:** 2
**Recommendation:** 5

**Strength And Weaknesses:**

Strengths:
1. The proposed method is clearly explained and theoretically analyzed.
2. Experiments have some promising results.

Weaknesses
1. The reason for asynchrony should be better explained, possibly with some real-world use cases. Actually, it has been observed that asynchrony may not be the best choice in distributed training, because it uses out-of-date (i.e., stale) gradients to update models. This staleness often results in degraded performance. Therefore, several state-of-the-art methods still use synchronous methods to achieve strong empirical performance, with some simple tricks such as large batch size, warmup, layerwise adaptive learning rates, and gradient compression (see reference [1-5] from below). I think this tradeoff between synchrony and asynchrony should be properly discussed.
2. Following the first point, the authors is encouraged to consider how to resolve the staleness issue due to asynchrony. This could further enhance this paper.
3. The experiments are conducted on two applications of bi-level optimization: hyper-cleaning and regularization coefficients optimization. I think bi-level optimization has received great attention these days mainly because of some emerging applications such as meta-learning, neural architecture search and etc. The authors are encouraged to consider these applications (e.g., meta-learning, neural architecture search) in the experiments.

Reference:
[1] Goyal, Priya, et al. "Accurate, large minibatch sgd: Training imagenet in 1 hour." arXiv preprint arXiv:1706.02677 (2017).
[2] You, Yang, et al. "Large batch optimization for deep learning: Training bert in 76 minutes." arXiv preprint arXiv:1904.00962 (2019).
[3] Huo, Zhouyuan, Bin Gu, and Heng Huang. "Large batch optimization for deep learning using new complete layer-wise adaptive rate scaling." Proceedings of the AAAI Conference on Artificial Intelligence. Vol. 35. No. 9. 2021.
[4] Liu, Rui, and Barzan Mozafari. "Communication-efficient Distributed Learning for Large Batch Optimization." International Conference on Machine Learning. PMLR, 2022.
[5] Wang, Tong, et al. "Large Batch Optimization for Object Detection: Training COCO in 12 minutes." European Conference on Computer Vision. Springer, Cham, 2020.

**Summary Of The Paper:**

This paper proposed an asynchronous method for distributed bilevel optimization. The proposed method can tackle both nonconvex upper-level and lower-level objective functions. Convergence analysis has been provided.

**Summary Of The Review:**

The paper proposed an interesting method. But I would appreciate it if the authors could address my above concerns.

---

> ### Author Response · Authors · 2022-11-12
> **Author Response (Part 2)**
>
> $(\textbf{Q2})$ Following the first point, the authors are encouraged to consider how to resolve the staleness issue due to asynchrony. This could further enhance this paper.
>
> $\textbf{Reply:}$
>
> Thanks for your insightful comments. According to [11], the staleness issues refer to the slower convergence rate or nonconverging executions issues caused by stale updates in asynchronous methods. And there are some works [12][13] observing that only high-stale information could lead to the staleness issues. There are many ways to alleviate the staleness issue in asynchronous algorithm, e.g., setting a bounded delay condition [14], delay compensation [15], dampening the step size of stale updates [16] and so on. Following [14], the proposed method enforces a bounded delay condition to ensure the sufficient freshness of all the updates in the algorithm, which alleviates the staleness issue. Specifically, every worker has to communicate with the master at least once every $\tau$ iterations, where $\tau$ is a user-defined parameter and can be controlled flexibly. Per your suggestion, we have clarified it in page 5, as ``Following [14], to alleviate the staleness issue in ADBO,  we set that master updates its variables once it receives updates from $S$ active workers every iteration and every worker has to communicate with the master at least once every $\tau$ iterations.''.
>
> Reference
>
> [11] Bäckström, Karl, Marina Papatriantafilou, and Philippas Tsigas. ``ASAP. SGD: Instance-based Adaptiveness to Staleness in Asynchronous SGD." International Conference on Machine Learning. PMLR, 2022.
>
> [12] Zhou, Zihao, et al. ``Towards Efficient and Stable K-Asynchronous Federated Learning with Unbounded Stale Gradients on Non-IID Data." IEEE Transactions on Parallel and Distributed Systems (2022).
>
> [13] Mitliagkas, Ioannis, et al. ``Asynchrony begets momentum, with an application to deep learning." 2016 54th Annual Allerton Conference on Communication, Control, and Computing (Allerton). IEEE, 2016.
>
> [14] Zhang, Ruiliang, and James Kwok. ``Asynchronous distributed ADMM for consensus optimization." International conference on machine learning. PMLR, 2014.
>
> [15] Zheng, Shuxin, et al. ``Asynchronous stochastic gradient descent with delay compensation." International Conference on Machine Learning. PMLR, 2017.
>
> [16] Sra, Suvrit, et al.``Adadelay: Delay adaptive distributed stochastic optimization." Artificial Intelligence and Statistics. PMLR, 2016.
>
> $(\textbf{Q3})$ The experiments are conducted on two applications of bi-level optimization: hyper-cleaning and regularization coefficients optimization. I think bi-level optimization has received great attention these days mainly because of some emerging applications such as meta-learning, neural architecture search and etc. The authors are encouraged to consider these applications (e.g., meta-learning, neural architecture search) in the experiments.
>
> $\textbf{Reply:}$
>
> Thank you for your constructive suggestion. Per your suggestion, we have added more experiments in additional application, i.e., meta-learning. It is seen from the experiment results that the proposed method achieves faster convergence rate than the baseline methods. Details of the added experiment results can be found in Figure 9 and Figure 10, Appendix A, page 17.
>
> $(\textbf{Q4})$ The paper proposed an interesting method. But I would appreciate it if the authors could address my above concerns.
>
> $\textbf{Reply:}$
>
> We truly appreciate your constructive and insightful comments.  Per your suggestion, we have added more discussions about: 1) the trade-off between synchrony and asynchrony; 2) the reason for asynchrony with real-world cases. Secondly, we have explained how the proposed method alleviate the staleness issue and have discussed it in the future work. Finally, more experiment results on additional application, i.e., meta-learning, have been added.

---

> ### Author Response · Authors · 2022-11-12
> **Author Response (Part 1)**
>
> Thank you for your insightful comments. We have provided a point by point reply to your questions as follows.
>
> $(\textbf{Q1})$ The reason for asynchrony should be better explained, possibly with some real-world use cases. Actually, it has been observed that asynchrony may not be the best choice in distributed training, because it uses out-of-date (i.e., stale) gradients to update models. This staleness often results in degraded performance. Therefore, several state-of-the-art methods still use synchronous methods to achieve strong empirical performance, with some simple tricks such as large batch size, warmup, layerwise adaptive learning rates, and gradient compression (see reference [1-5] from below). I think this tradeoff between synchrony and asynchrony should be properly discussed.
>
> $\textbf{Reply:}$
>
> We appreciate your suggestion. We agree with you, the synchronous algorithm and asynchronous algorithm have different application scenarios. When the delay of each worker is not much different, the synchronous algorithm may exhibit better performance than the asynchronous algorithm. While there are stragglers in the distributed system, the asynchronous algorithm is more preferred. Per your suggestion, to properly discuss the tradeoff between synchrony and asynchrony, we have added it in Related Work, as ``In summary, the synchronous and asynchronous algorithm have different application scenarios. When the delay of each worker is not much different, the synchronous algorithm suits better. While there are stragglers in the distributed system, the asynchronous algorithm is more preferred.''.
>
> Moreover, we  have added the suggested reference to better discuss the synchrony in Related Work, as ``There are several advanced techniques been proposed to make the synchronous algorithm more efficient, such as large batch size, warmup and so on [1][2][3][4][5]. ''
>
> The reason for asynchrony has been better explained with real-world cases in the Related Work, as ``The asynchronous distributed algorithm is strongly preferred for large scale distributed systems in practice since it does not suffer from the straggler problem [6]. Asynchronous distributed methods have been employed for many real-world applications, such as Google's DistBelief system [7], the training of  10 million YouTube videos [8], federated learning for edge computing [9][10].''.
>
> Finally, we would like to clarify that while the proposed algorithm is asynchronous when $S<N$, it becomes a synchronous algorithm by simply setting $S=N$, i.e., the master node updates the parameters till it receives the messages from all workers.  Therefore, the proposed algorithm also offers the flexibility of being implemented in either asynchronous or synchronous manner.
>
> Reference
>
> [1] Goyal, Priya, et al. ``Accurate, large minibatch sgd: Training imagenet in 1 hour." arXiv preprint arXiv:1706.02677 (2017).
>
> [2] You, Yang, et al. ``Large batch optimization for deep learning: Training bert in 76 minutes." arXiv preprint arXiv:1904.00962 (2019).
>
> [3] Huo, Zhouyuan, Bin Gu, and Heng Huang. ``Large batch optimization for deep learning using new complete layer-wise adaptive rate scaling." Proceedings of the AAAI Conference on Artificial Intelligence. Vol. 35. No. 9. 2021.
>
> [4] Liu, Rui, and Barzan Mozafari. ``Communication-efficient Distributed Learning for Large Batch Optimization." International Conference on Machine Learning. PMLR, 2022.
>
> [5] Wang, Tong, et al. ``Large Batch Optimization for Object Detection: Training COCO in 12 minutes." European Conference on Computer Vision. Springer, Cham, 2020.
>
> [6] Jiang, Jiyan, et al. ``Asynchronous decentralized online learning."Advances in Neural Information Processing Systems 34 (2021): 20185-20196.
>
> [7] Dean, Jeffrey, et al. ``Large scale distributed deep networks." Advances in neural information processing systems 25 (2012).
>
> [8] Le, Quoc V. ``Building high-level features using large scale unsupervised learning." 2013 IEEE international conference on acoustics, speech and signal processing. IEEE, 2013.
>
> [9] Liu, Yinghui, et al. ``Blockchain-enabled asynchronous federated learning in edge computing."
>
> [10] Lu, Yunlong, et al. ``Differentially private asynchronous federated learning for mobile edge computing in urban informatics."

---

> ### Author Response · Authors · 2022-11-15
> **We are happy to address any further concerns**
>
> We sincerely thank you for raising the concerns in the initial reviews. We have tried our best efforts to clarify those concerns in the responses. Given the limited time for discussion, we would really appreciate it if you could let us know in case there is any additional concern.

---

### Official Review · Reviewer_9ja2 · 2022-10-26

**Confidence:** 4
**Correctness:** 4
**Technical Novelty And Significance:** 3
**Empirical Novelty And Significance:** 2
**Recommendation:** 5

**Clarity, Quality, Novelty And Reproducibility:**

The proposed approach is novel to the best of my knowledge and makes a useful contribution. As mentioned earlier, the paper is a bit dense and I found some of the math a bit hard to follow. Appendix A on the centralized version of the same algorithm is much cleaner and still retains most of the key ideas of the paper - a possible rewrite would be to focus on the centralized version first and then explain the differences in the distributed setting.

The paper includes experimental results in two settings - (i) data hyper-cleaning [classification with noisy labels] and (ii) regularization coefficient optimization on two specific datasets each. The paper would benefit from a discussion of why those specific datasets were chosen (for instance, do the results differ on CIFAR-10/100 instead of MNIST?). Also the paper does not provide enough details for easy reproducibility - making the code available would go a long way.


**Strength And Weaknesses:**

Strengths:
+ The paper addresses a timely problem and provides a novel algorithm.
+ It’s interesting that the centralized version of the same algorithm also yields the optimal convergence rate in the centralized setting.

Weaknesses:
- The paper is a bit dense and I feel that the presentation could be improved significantly. Currently section 3 serves as a large unbroken block of equations, and readability would be improved if it were broken down further. For instance, divide the main algorithm into phases and describe the steps to estimate \phi(v) [Equations 5-9] and the steps to describe and generate the cutting planes separately. Even the paragraph on page 2 regarding the reformulation as a consensus problem feels out of place there - I would much rather see it at the beginning of section 3.


**Summary Of The Paper:**

Bilevel optimization problems, in which one optimization problem is nested within another, occur in multiple machine learning applications such as hyperparameter optimization. The paper considers a generic distributed bilevel optimization problem and proposes a novel asynchronous distributed algorithm. The proposed algorithm is guaranteed to converge to an \eps-stationary point in O(1\eps^2) iterations in a distributed setting.

**Summary Of The Review:**

Overall, the paper tackles an interesting problem and the proposed algorithm is novel and provides the first general framework for solving distributed bilevel optimization problems in an asynchronous setting. The empirical analysis, while limited, also shows promising results.

---

> ### Author Response · Authors · 2022-11-12
> **Author Response**
>
> Thanks for your comments. Below please find the point by point reply to your questions.
>
> $(\textbf{Q1})$ The paper is a bit dense and I feel that the presentation could be improved significantly. Currently section 3 serves as a large unbroken block of equations, and readability would be improved if it were broken down further. For instance, divide the main algorithm into phases and describe the steps to estimate $\phi ({\boldsymbol{v}})$ [Equations 5-9] and the steps to describe and generate the cutting planes separately. Even the paragraph on page 2 regarding the reformulation as a consensus problem feels out of place there - I would much rather see it at the beginning of section 3.
>
> $\textbf{Reply:}$
>
> We appreciate your constructive suggestion. Per your suggestion, we have carefully modified the presentation of the full paper, especially Section 3. And Section 3 has been broken down into four parts, that is, 3.1 estimate of solution to lower-level optimization problem, 3.2 polyhedral approximation, 3.3 asynchronous algorithm, 3.4 updating cutting planes. And we have removed the paragraph on page 2 regarding the reformulation as a consensus problem to the beginning of Section 3.
>
> Moreover, we have added an overview of the proposed method at the beginning of Section 3, as ``To better clarify how ADBO works, we sketch the procedure of ADBO. Firstly, ADBO computes the estimate of the solution to lower-level optimization problem. Then, inspired by cutting plane method, a set of cutting planes is utilized to approximate the feasible region of the upper-level bilevel optimization problem. Finally, the asynchronous algorithm for solving the resulting problem and how to update cutting planes are proposed. The remaining contents are divided into four parts, i.e., estimate of solution to lower-level optimization problem, polyhedral approximation, asynchronous algorithm, updating cutting planes.''
>
> $(\textbf{Q2})$ The paper includes experimental results in two settings - (i) data hyper-cleaning [classification with noisy labels] and (ii) regularization coefficient optimization on two specific datasets each. The paper would benefit from a discussion of why those specific datasets were chosen (for instance, do the results differ on CIFAR-10/100 instead of MNIST?). Also the paper does not provide enough details for easy reproducibility - making the code available would go a long way.
>
> $\textbf{Reply:}$
>
> Thank you for your insightful suggestion. The datasets utilized in experiments are based on the previous bilevel optimization works [1][2][3], and we have clarified it in the experiments as ''Following [1][2], we compare the performance of the proposed ADBO and distributed bilevel optimization FEDNEST on the distributed data hyper-cleaning task  on MNIST and Fashion MNIST datasets.'' and ''Following [3], we compare the proposed ADBO with baseline algorithms (FEDNEST and SDBO) on the regularization coefficient optimization task with Covertype and IJCNN1 datasets.''.
>
> Moreover, we have added additional experiment results on CIFAR-10  and Australian datasets, and the proposed ADBO also achieves faster convergence rate. Details of the added experiment results can be found in Figure 11 and Figure 12, Appendix D.1, page 35. And more details about the experiments setting are given in Appendix D.2, page 36. The code is released in https://github.com/ICLR23Submission6251/adbo.
>
> Reference
>
> [1] Provably Faster Algorithms for Bilevel Optimization.
>
> [2] Bilevel Optimization: Convergence Analysis and Enhanced Design.
>
> [3] A Single-Timescale Method for Stochastic Bilevel Optimization.
>
> $(\textbf{Q3})$ Overall, the paper tackles an interesting problem and the proposed algorithm is novel and provides the first general framework for solving distributed bilevel optimization problems in an asynchronous setting. The empirical analysis, while limited, also shows promising results.
>
> $\textbf{Reply:}$
>
> We truly appreciate your insightful comments. We have added more experiment results to enrich the empirical analysis. Specifically, more results on CIFAR-10 and Australian datasets are added in Figure 11,12, Appendix D.1, page 35. And results on additional application, i.e., meta-learning, have been added in Figure 9,10, Appendix A, page 17.

---

> ### Author Response · Authors · 2022-11-15
> **We are happy to address any further concerns**
>
> We sincerely thank you for raising the concerns in the initial reviews. We have tried our best efforts to clarify those concerns in the responses. Given the limited time for discussion, we would really appreciate it if you could let us know in case there is any additional concern.

---

### Official Review · Reviewer_KFzJ · 2022-10-30

**Confidence:** 4
**Correctness:** 3
**Technical Novelty And Significance:** 2
**Empirical Novelty And Significance:** 2
**Recommendation:** 5

**Clarity, Quality, Novelty And Reproducibility:**

This paper is original and novel. However the written part needs to be improved.

**Strength And Weaknesses:**

Strength:
1. Strong empirical results with theoretical analysis

Weakness:
1. Redundant formulation compare to other bilevel paper, which makes it hard to read, especially when explaining how cutting plane works.
2. Insufficient references. There are works on asynchronous distributed optimization that are not included in this paper. For example, https://ieeexplore.ieee.org/ielaam/6884276/8350883/7903733-aam.pdf. There are also many works asynchronous federated learning, and probably needed to be included here.
3.  What is the value of $\tau$ in the experiment?
4. In the appendix, I don't quite follow the logic from eqn 57 to eqn 58. Can you please explain more?
5. I don't understand from optimization perspective, why this algorithm can outperform other algorithms in asynchronous setting. From Theorem 2, the larger $S$ is, then the less number of iterations is needed. For experiment, what if you run other synchronous algorithms (e.g. SDBO) in the same setting of ADBO, i.e. that the master problem will update its variable once it receives updates from $S$ workers. Will ADBO still outperform SDBO?

**Summary Of The Paper:**

This paper proposes an algorithm that solves bilevel optimization problem in an asynchronous distributed manner. The iteration complexity for the algorithm to obtain $\epsilon$-stationary point is upper bounded by $\mathcal{O}(\frac{1}{\epsilon^2})$. Empirical results show that under asynchronous setting, the proposed algorithm outperforms state-of-art distributed bilevel optimization algorithm.

**Summary Of The Review:**

I think this paper is marginally below the acceptance threshold. Please see the weakness part for my reasoning. I think the authors need to revise the written to make the problem formulation/algorithm more neat and easier to follow.

---

> ### Author Response · Authors · 2022-11-12
> **Author Response (Part 2)**
>
> $(\textbf{Q4})$ In the appendix, I don't quite follow the logic from eqn 57 to eqn 58. Can you please explain more?
>
> $\textbf{Reply:}$
>
> Thanks for pointing out that. This is a typo in Eqn 58, it should be ${\cal O}(\frac{{{{\hat L}_p}({{{x}}^{{T_1}}},{{{y}}^{{T_1}}}) - {{\hat L}_p}^*}}{d}\frac{1}{\epsilon} + {T_1})$. And this is because the upper bound in Eqn 57 has to satisfy $\frac{{{{\hat{L}_p}}({{\boldsymbol{x}}^{{T_1}}},{{\boldsymbol{y}}^{{T_1}}}) - {{\hat{L}_p}^{*}}}}{{(T - {T_1})d}}\le \epsilon$ according to the definition of $\epsilon$-stationary point, then we have Eqn 58. Per your suggestion, we have corrected this typo.
>
> $(\textbf{Q5})$ I don't understand from optimization perspective, why this algorithm can outperform other algorithms in asynchronous setting. From Theorem 2, the larger $S$ is, then the less number of iterations is needed. For experiment, what if you run other synchronous algorithms (e.g. SDBO) in the same setting of ADBO, i.e. that the master problem will update its variable once it receives updates from $S$ workers. Will ADBO still outperform SDBO?
>
> $\textbf{Reply:}$
>
> Thanks for your comments. Although the less number of iterations is needed with the increase of $S$, the time for convergence will increase since the maximum delay in each iteration increases. In the special case that $S=N$, the ADBO will become SDBO, whose speed is limited by the worker with maximum delay. In the experiment, for SDBO, we have $S = N$ since the master needs to receive the updates from all workers (i.e., $N$ workers) to update its information according to the synchronous setting.
>
> $(\textbf{Q6})$ This paper is original and novel. However the written part needs to be improved.
>
> $\textbf{Reply:}$
>
> We truly appreciate your constructive comments. Per your suggestions, we have carefully revised the manuscript to improve its readability.  Please see Section 2 on page 3, Section 3 on page 3 to 6, Section 5 on page 8 for details.

---

> ### Author Response · Authors · 2022-11-12
> **Author Response (Part 1)**
>
> Thank you for your constructive suggestions. Below please find our point by point reply to the questions you have raised.
>
> $(\textbf{Q1})$ Redundant formulation compare to other bilevel paper, which makes it hard to read, especially when explaining how cutting plane works.
>
> $\textbf{Reply:}$
>
> We truly appreciate your insightful comments. The formulation given in Eq. (2) is a standard bilevel optimization without any redundant variables.  The reformulation given in Eq. (3) is a  consensus problem which allows to develop distributed training algorithms for bilevel optimization based on the parameter server architecture. In parameter server architecture, the
> communication is conducted around a set of master nodes (or servers) that constitute the hubs of a star network, and worker nodes (or clients) pull the shared parameters from and send their updates to the master nodes [1]. Parameter server training is a well-known data-parallel approach for scaling up ML model training on a multitude of machines [2]. We have added more explanations into the revised manuscript to elaborate on the consensus problem, please refer to the beginning of Section 3 on page 3 for details.
>
> The goal of the cutting plane method is to construct a polytope to approximate the feasible region of the upper-level optimization problem [3].  To better explain how cutting plane method works, we divide the contents in Section 3 into four parts, 3.1 estimate of solution to lower-level optimization problem, 3.2 polyhedral approximation, 3.3 asynchronous algorithm, 3.4 updating cutting planes. And we have added an overview of the proposed method at the beginning of Section 3, as ``To better clarify how ADBO works, we sketch the procedure of ADBO. Firstly, ADBO computes the estimate of the solution to lower-level optimization problem. Then, inspired by cutting plane method, a set of cutting planes is utilized to approximate the feasible region of the upper-level bilevel optimization problem. Finally, the asynchronous algorithm for solving the resulting problem and how to update cutting planes are proposed.''
>
> Reference
>
> [1] Mahmoud Assran, Arda Aytekin, Hamid Reza Feyzmahdavian, Mikael Johansson, and Michael G Rabbat. Advances in asynchronous parallel and distributed optimization. Proceedings of the
> IEEE, 108(11):2013–2031, 2020.
>
> [2] Joost Verbraeken, Matthijs Wolting, Jonathan Katzy, Jeroen Kloppenburg, Tim Verbelen, and Jan S Rellermeyer. A survey on distributed machine learning. Acm computing surveys (csur), 53(2):
> 1–33, 2020.
>
> [3] Michalka, Alexander. Cutting planes for convex objective nonconvex optimization. Columbia University, 2013.
>
> $(\textbf{Q2})$ Insufficient references. There are works on asynchronous distributed optimization that are not included in this paper. For example, https://ieeexplore.ieee.org/ielaam/6884276/8350883/7903733-aam.pdf. There are also many works asynchronous federated learning, and probably needed to be included here.
>
> $\textbf{Reply:}$
>
> Thanks for your suggestion. We have added the suggested work [4] and works about the  asynchronous federated learning [6][7] into the Related Work as ``Asynchronous distributed methods [4][5] have been employed for a number of real-world applications, such as Google's DistBelief system, the training of  10 million YouTube videos, federated learning for edge computing [6][7].''.
>
> Reference
>
> [4] Tianyu Wu, Kun Yuan, Qing Ling, Wotao Yin, and Ali H Sayed. Decentralized consensus optimization with asynchrony and delays. IEEE Transactions on Signal and Information Processing over Networks, 4(2):293–307, 2017.
>
> [5] Yaohua Liu, Cameron Nowzari, Zhi Tian, and Qing Ling. Asynchronous periodic event-triggered
> coordination of multi-agent systems. In 2017 IEEE 56th Annual Conference on Decision and
> Control (CDC), pp. 6696–6701. IEEE, 2017.
>
> [6] Yinghui Liu, Youyang Qu, Chenhao Xu, Zhicheng Hao, and Bruce Gu. Blockchain-enabled asynchronous federated learning in edge computing. Sensors, 21(10):3335, 2021c.
>
> [7] Yunlong Lu, Xiaohong Huang, Yueyue Dai, Sabita Maharjan, and Yan Zhang. Differentially private asynchronous federated learning for mobile edge computing in urban informatics. IEEE Transactions on Industrial Informatics, 16(3):2134–2143, 2019.
>
> $(\textbf{Q3})$ What is the value of $\tau$ in the experiment?
>
> $\textbf{Reply:}$
>
> Thanks for your suggestion. We have added the value of $\tau$ in the experiment in page 8 as  ''In MNIST and Fashion MNIST datasets, we set $N=18$, $S=9$ and $\tau=15$.'', and ''In Covertype and IJCNN1 datasets, we set $N=18$, $S=9$, $\tau = 15$ and $N=24$, $S=12$, $\tau = 15$, respectively.''. And more details of the experiments have been added into Appendix D.2, page 36. The code is released in https://github.com/ICLR23Submission6251/adbo.

---

> ### Author Response · Authors · 2022-11-15
> **We are happy to address any further concerns**
>
> We sincerely thank you for raising the concerns in the initial reviews. We have tried our best efforts to clarify those concerns in the responses. Given the limited time for discussion, we would really appreciate it if you could let us know in case there is any additional concern.

---

### Author Response · Authors · 2022-11-17
**Summary of concerns and our responses**

Thanks for the constructive suggestions from all reviewers. We summarize the main concerns from reviewers and our corresponding replies as follows to facilitate further discussions.

$\textbf{(Reviewer KFzj)}$

$(\textbf{Q})$ Redundant formulation, especially when explaining how cutting plane work

$\textbf{Reply:}$

The formulation given in Eq. (2) is a standard bilevel optimization without any redundant variables.  The reformulation given in Eq. (3) is a  consensus problem which allows to develop distributed training algorithms for bilevel optimization based on the parameter server architecture. We have added more explanations into the revised manuscript to elaborate on the consensus problem, please refer to the beginning of Section 3 on page 3 for details. The goal of the cutting plane method is to construct a polytope to approximate the feasible region of the upper-level optimization problem, we have added more explanations into the revised manuscript, please refer to Section 3 for details.

$(\textbf{Q})$ Insufficient references

$\textbf{Reply:}$

We have added the references as suggested.

$\textbf{(Reviewer 9ja2)}$

$(\textbf{Q})$ The presentation could be improved

$\textbf{Reply:}$

We have carefully modified the presentation of the full paper, especially Section 3. And Section 3 has been broken down into four parts, i.e., 3.1 estimate of solution to lower-level optimization problem, 3.2 polyhedral approximation, 3.3 asynchronous algorithm, 3.4 updating cutting planes. And we have removed the paragraph on page 2 regarding the reformulation as a consensus problem to the beginning of Section 3. Moreover, we have added an overview of the proposed method at the beginning of Section 3.

$(\textbf{Q})$ Experiment results and reproducibility

$\textbf{Reply:}$

The datasets utilized in experiments are based on the previous bilevel optimization works. And we have added additional experiment results on CIFAR-10  and Australian datasets, and the proposed ADBO also achieves faster convergence rate. Details of the added experiment results can be found in Figure 11 and Figure 12, page 35 in the revised manuscript. And more details about the experiments setting are given in Appendix D.2, page 36. The code is released in https://github.com/ICLR23Submission6251/adbo.

$\textbf{(Reviewer 4Y6m)}$

$(\textbf{Q})$ The reason for asynchrony and the tradeoff between synchrony and asynchrony should be better explained

$\textbf{Reply:}$

We agree with you, the synchronous and asynchronous algorithms have different application scenarios. When the delay of each worker is not much different, the synchronous algorithm may exhibit better performance than the asynchronous algorithm. While there are stragglers in the distributed system, the asynchronous algorithm is more preferred. And we have added the discussion in Related Work. And the suggested reference [1-5] have also been added in the Related Work to better discuss the tradeoff between synchrony and asynchrony.

The reason for asynchrony has been better explained with real-world cases in the Related Work, as ``The asynchronous distributed algorithm is strongly preferred for large scale distributed systems in practice since it does not suffer from the straggler problem. Asynchronous distributed methods have been employed for many real-world applications, such as Google's DistBelief system, the training of  10 million YouTube videos, federated learning for edge computing.''.

$(\textbf{Q})$ How to resolve the staleness issue

$\textbf{Reply:}$

According to previous works, the staleness issues refer to the slower convergence rate or nonconverging executions issues caused by stale updates. And there are some works observing that only high-stale information could lead to the staleness issues. There are many ways to alleviate the staleness issue in asynchronous algorithm, e.g., setting a bounded delay condition, delay compensation, dampening the step size of stale updates and so on. Following the previous works, the proposed method enforces a bounded delay condition to ensure the sufficient freshness of all the updates in the algorithm, which alleviates the staleness issue. Specifically, every worker has to communicate with the master at least once every $\tau$ iterations, where $\tau$ is a user-defined parameter and can be controlled flexibly.

$(\textbf{Q})$ Adding more applications in the experiments

$\textbf{Reply:}$

Per your suggestion, we have added more experiment results regarding meta-learning. Details of the added experiment results can be found in Figure 9 and Figure 10, page 17 in the revised manuscript.

---

### Author Response · Authors · 2022-11-18
**To AC: Summary of our contributions and modifications**

Thanks for your help on handling our paper, to facilitate further discussions, we have summarized our contributions in this paper.

$\textbf{Contributions}$


* A novel algorithm ADBO has been proposed in this paper, which can solve the bilevel optimization problem in an $\textbf{asynchronous}$ $\textbf{distributed}$ $\textbf{manner}$. To the best of our knowledge, it is the $\textbf{first}$ work that tackles the bilevel optimization problem in an asynchronous distributed manner.



* We demonstrate that the proposed ADBO can be applied to bilevel optimization with $\textbf{non-convex}$ upper-level and lower-level objectives $\textbf{with}$ $\textbf{constraints}$. We also theoretically derive that the iteration complexity for the  proposed ADBO to obtain the $\epsilon$-stationary point is upper bounded by $\mathcal{O}(\frac{1}{{{\epsilon ^2}}})$.

* Our thorough empirical studies justify the superiority of the proposed ADBO over the existing state of the art methods.

According to the suggestions of all reviewers, we have carefully modified the manuscript, the modifications can be summarized as follows.

$\textbf{Modifications}$

* (Presentation) The presentation of the full paper has been carefully modified, especially Section 3.

* (Insufficient reference) Some references have been added as suggested.

* (Experiment results) More experiment results have been added into the revised manuscript. Specifically, experiment results regarding meta-learning are added in Appendix A, please refer to Figure 9 and Figure 10. More experiment results on CIFAR-10 and Australian datasets have been added in Appendix D, please refer to Figure 11 and Figure 12.

* (Reproducibility) More details about the experiment setting are added in Appendix D. And the code is released in https://github.com/ICLR23Submission6251/adbo.

* (Discussion about synchrony and asynchrony) More discussions about the synchrony and asynchrony have been added in Section 2.

* (Staleness issue) We have added more explanations about how the proposed method resolve the staleness issue in Section 3.

---

### Author Response · Authors · 2022-12-07
**We are happy to receive any feedback**

Dear reviewers,

Thanks for the suggestions you provided. Do you have any further concerns and feedback?  Please let us know, thanks.

Best regards

---

### Decision · Program_Chairs · 2023-01-20

**Decision:**

Accept: poster

**Justification For Why Not Higher Score:**

NA

**Justification For Why Not Lower Score:**

NA

**Metareview: Summary, Strengths And Weaknesses:**

This paper proposed an asynchronous algorithm for Distributed Bilevel Optimization (DBO) that theoretically achieves an $O(1/\epsilon^2)$ convergence rate and achieves good empirical performance. The main complaints from the reviewers are (1) poor writing in the main text and (2) outdated applications in the experiments. The authors have sufficiently addressed both concerns in their revision. I believe this is a solid paper, and the revision has improved significantly over the initial draft. For that reason, I would vote for acceptance of this paper, but I won't be upset if this paper undergoes another round of reviews at a future conference, because it is also reasonable that a significant revision must go through another round of peer review, as is common in journal reviewing process.

**Note From Pc:**

if the above contains the word "oral" or "spotlight" please see: "oral" presentation means -> notable-top-5% and "spotlight" means -> notable-top-25%. As stated in our emails, we are disassociating presentation type from AC recommendations